# Tumour-wide RNA splicing aberrations generate actionable public neoantigens

Darwin W. Kwok[1], Nicholas O. Stevers[1], Iñaki Etxeberria[2,3], Takahide Nejo[1], Maggie Colton Cove[1], Lee H. Chen[1], Jangham Jung[1], Kaori Okada[1], Senthilnath Lakshmanachetty[1], Marco Gallus[1,4], Abhilash Barpanda[5], Chibo Hong[1], Gary K. L. Chan[1], Jerry Liu[1], Samuel H. Wu[1], Emilio Ramos[5], Akane Yamamichi[1], Payal B. Watchmaker[1], Hirokazu Ogino[1], Atsuro Saijo[1], Aidan Du[1], Nadia R. Grishanina[1], James Woo[1], Aaron Diaz[1], Shawn L. Hervey-Jumper[1], Susan M. Chang[1], Joanna J. Phillips[1,6], Arun P. Wiita[5,7,8], Christopher A. Klebanoff[2,3,9 ✉], Joseph F. Costello[1 ✉] & Hideho Okada[1,10 ✉]

T cell-based immunotherapies hold promise in treating cancer by leveraging the immune system's recognition of cancer-specific antigens[1]. However, their efficacy is limited in tumours with few somatic mutations and substantial intratumoural heterogeneity[2-4]. Here we introduce a previously uncharacterized class of tumour-wide public neoantigens originating from RNA splicing aberrations in diverse cancer types. We identified T cell receptor clones capable of recognizing and targeting neoantigens derived from aberrant splicing in *GNAS* and *RPL22*. In cases with multi-site biopsies, we detected the tumour-wide expression of the *GNAS* neojunction in glioma, mesothelioma, prostate cancer and liver cancer. These neoantigens are endogenously generated and presented by tumour cells under physiologic conditions and are sufficient to trigger cancer cell eradication by neoantigen-specific CD8+ T cells. Moreover, our study highlights a role for dysregulated splicing factor expression in specific cancer types, leading to recurrent patterns of neojunction upregulation. These findings establish a molecular basis for T cell-based immunotherapies addressing the challenges of intratumoural heterogeneity.

Cell-based immunotherapy offers durable survival benefits across various malignancies[5,6]. However, many tumours evade eradication owing to intratumoural heterogeneity (ITH)[7,8] in their cellular and genetic landscape. Although immunotherapy is beneficial in tumours with high levels of immune infiltration and high mutational loads[9,10], cancers with extensive ITH or lower mutational burdens remain resistant[2-4]. Current immunotherapies targeting tumour-specific antigens (TSAs) derived from nonsynonymous somatic mutations[6,11] provide limited targets in tumours with low mutational burdens[12,13]. To expand immunotherapeutic options, recent studies have explored cancer-specific splicing events (neojunctions (NJs)) as a source of TSAs[14,15]. NJs are prevalent and can generate TSAs that activate CD8+ T cell responses[14,16,17]. Nevertheless, the spatial and temporal conservation of NJs across entire tumours has not been studied, leaving their clonality unclear.

To address this gap, we investigated the clonality of NJs across cancer types to identify 'public', tumour-wide NJ-derived TSAs. Using a comprehensive pipeline, we mapped RNA splicing junctions across distinct intratumour regions to characterize spatially conserved NJs (Extended Data Fig. 1). We identified NJ-derived TSAs that were proteolytically processed and presented on prevalent human leukocyte antigen (HLA) molecules. These TSAs elicited T cell receptor (TCR)

signalling and antigen-dependent tumour cell tumour killing by CD8+ T cells. These findings demonstrate the potential of targeting tumour-wide public NJ-derived TSAs as a new class of 'off-the-shelf' cancer immunotherapies.

## Characterization of public, pan-cancer NJs

We analysed RNA-sequencing (RNA-seq) data from The Cancer Genome Atlas (TCGA) to identify non-annotated junction reads across 12 cancer types with spatially mapped tumour samples (Fig. 1a and Extended Data Fig. 1a). Only samples with tumour purities of ≥60% were included[18,19] (Fig. 1b) when identifying protein-coding, non-annotated junctions (Extended Data Fig. 1b). A junction's positive sample rate (PSR) represents the percentage of samples in a cohort that express the NJ with a read frequency of ≥1% relative to the canonical splicing junction[20]. Public NJs are identified as those with elevated PSRs in each TCGA tumour cohort (PSR$_{TCGA}$ ≥ 10%; Fig. 1c and Extended Data Fig. 1c). Following NJ nomenclature[14], cancer-specific splicing events were defined as a PSR of <1% in normal tissue from the Genotype-Tissue Expression (GTEx) project (n = 9,166; PSR$_{GTEx}$ < 1%; Extended Data Fig. 1d). On average, 94 public NJs were identified per TCGA tumour type (Fig. 1d and

[1]Department of Neurological Surgery, University of California, San Francisco, San Francisco, CA, USA. [2]Human Oncology and Pathogenesis Program, Memorial Sloan Kettering Cancer Center, New York, NY, USA. [3]Parker Institute for Cancer Immunotherapy, New York, NY, USA. [4]Department of Neurosurgery, University Hospital Muenster, Muenster, Germany. [5]Department of Laboratory Medicine, University of California, San Francisco, San Francisco, CA, USA. [6]Department of Pathology, University of California, San Francisco, San Francisco, CA, USA. [7]Department of Bioengineering and Therapeutic Sciences, University of California, San Francisco, San Francisco, CA, USA. [8]Chan Zuckerberg Biohub San Francisco, San Francisco, CA, USA. [9]Department of Medicine, Memorial Sloan Kettering Cancer Center, New York, NY, USA. [10]Parker Institute for Cancer Immunotherapy, San Francisco, CA, USA. ✉e-mail: klebanoc@mskcc.org; joseph.costello@ucsf.edu; hideho.okada@ucsf.edu

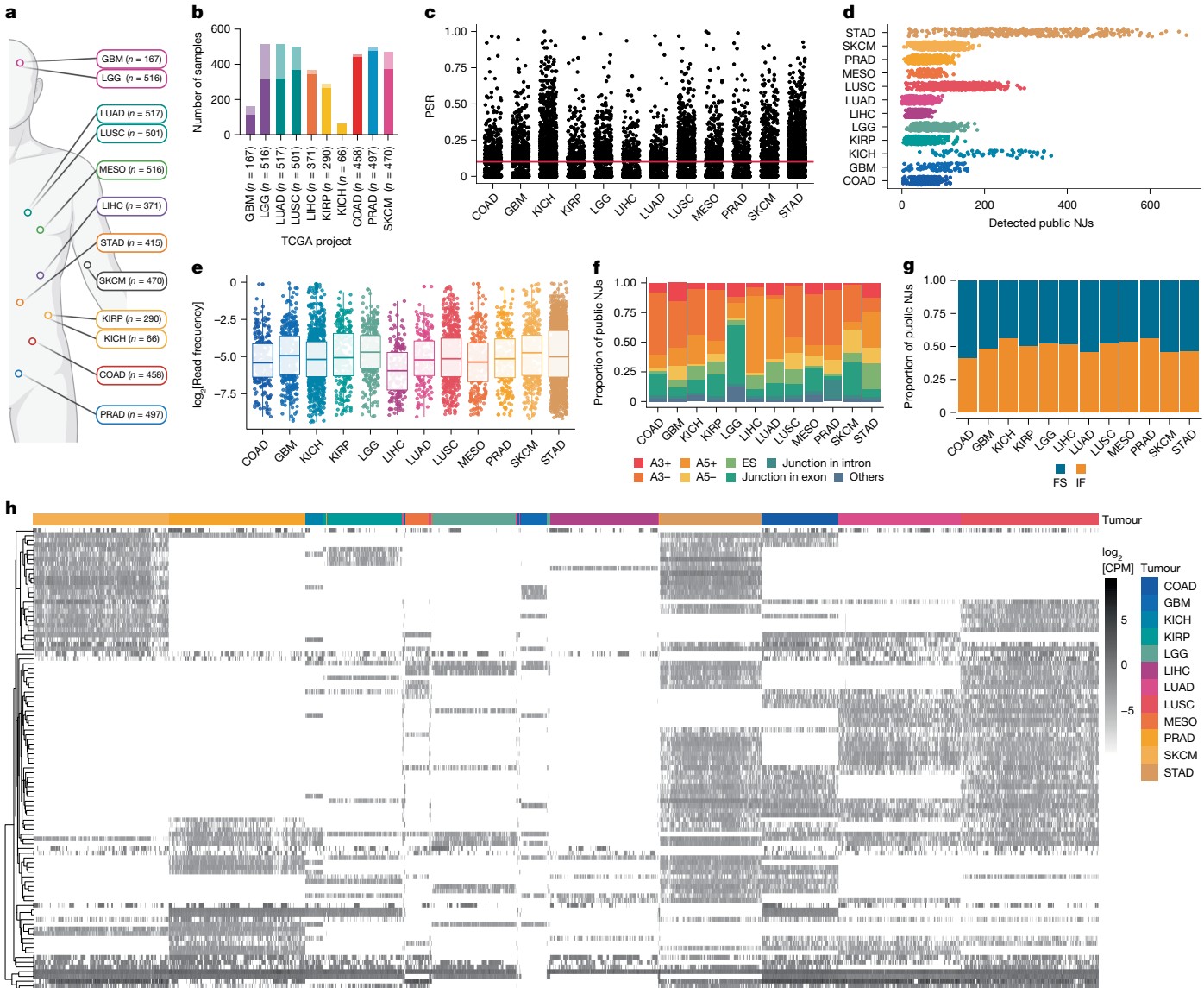

**Fig. 1 | Characterization of public NJs across multiple cancer types. a**, TCGA RNA-seq data were analysed across GBM (*n* = 167 samples), LGG (*n* = 516), LUAD (*n* = 517), lung squamous cell carcinoma (LUSC; *n* = 501), mesothelioma (MESO; *n* = 516), LIHC (*n* = 371), stomach adenocarcinoma (STAD; *n* = 415), SKCM (*n* = 470), kidney renal papillary cell carcinoma (KIRP; *n* = 290), kidney chromophobe (KICH, *n* = 66), colon adenocarcinoma (COAD; *n* = 458) and prostate adenocarcinoma (PRAD; *n* = 497). **b**, Samples with tumour purity ≥60% (solid colour) were selected for analysis, excluding MESO and STAD owing to unavailable purity data. **c**, Interpatient NJ frequency (PSR) was analysed, with public NJs defined as PSR ≥ 10% (red line). **d,e**, Total number (**d**) and log₂[read frequency] (**e**) of public NJs detected per sample across tumour types (COAD, *n* = 265; GBM, *n* = 391; KICH, *n* = 773; KIRP, *n* = 247; LGG, *n* = 327; LIHC, *n* = 173; LUAD, *n* = 175; LUSC, *n* = 555; MESO, *n* = 277; PRAD, *n* = 245; SKCM, *n* = 353; STAD, *n* = 1,433). **f,g**, Public NJs were categorized by splice type: exonic loss at the 3′ or 5′ splice site (A3 or A5 loss (A3−; A5−)), intronic gain at the 3′ or 5′ splice site (A3 or A5 gain (A3+; A5+)), exon skip (ES), junction in exon, junction in intron and others (**f**) and frameshift (FS) status (**g**); IF, in-frame. **h**, Expression of all pan-cancer-spanning NJs (log₂[counts per million (CPM)]) across all studied TCGA tumour types. Further statistical details are provided in Supplementary Table 3. **a**, Created in BioRender (credit: D.W.K., https://BioRender.com/k09l557; 2024).

Supplementary Table 1), with consistent frequencies across samples (Fig. 1e). Public NJs varied by splice type (Fig. 1f) and had consistent proportions of frameshift-inducing splicing events (Fig. 1g). Some NJs were also found in recent splicing studies[17,21] (Extended Data Fig. 1e,f). Unbiased hierarchical clustering revealed that NJ expression grouped by tumour type, suggesting conserved patterns. Additionally, a subset of NJs was expressed across multiple tumour types (Fig. 1h), indicating potential pan-cancer immunotherapy targets arising from aberrant splicing.

## NJs exhibit ITH

To mitigate immune evasion due to antigenic heterogeneity, we need to target multiple neoantigens shared across the entire tumour[1]. NJs,

which can generate immunogenic antigens, present a promising avenue. We analysed intratumoural RNA-seq from prostate[22], liver[23–26], colon[23,27], stomach[23], kidney[23] and lung[28,29] cancers to assess spatial conservation of public NJs (Fig. 2a and Extended Data Fig. 2a). This revealed public NJs consistently expressed across multiple intratumoural samples (Fig. 2b and Extended Data Fig. 2b,c) in many patients (Fig. 2c).

Extensive ITH is common in gliomas, further complicating immunotherapy[30,31]. To examine ITH in depth, we increased the number of intratumoural biopsies analysed across the three main glioma subtypes[32–34]. Approximately 10 maximally distanced, spatially mapped samples were analysed from 51 glioma cases with exome and RNA-seq (Fig. 2d and Extended Data Fig. 2d–h) to detect NJs expressed intratumourally across multiple patients. Iterating from one to ten samples,

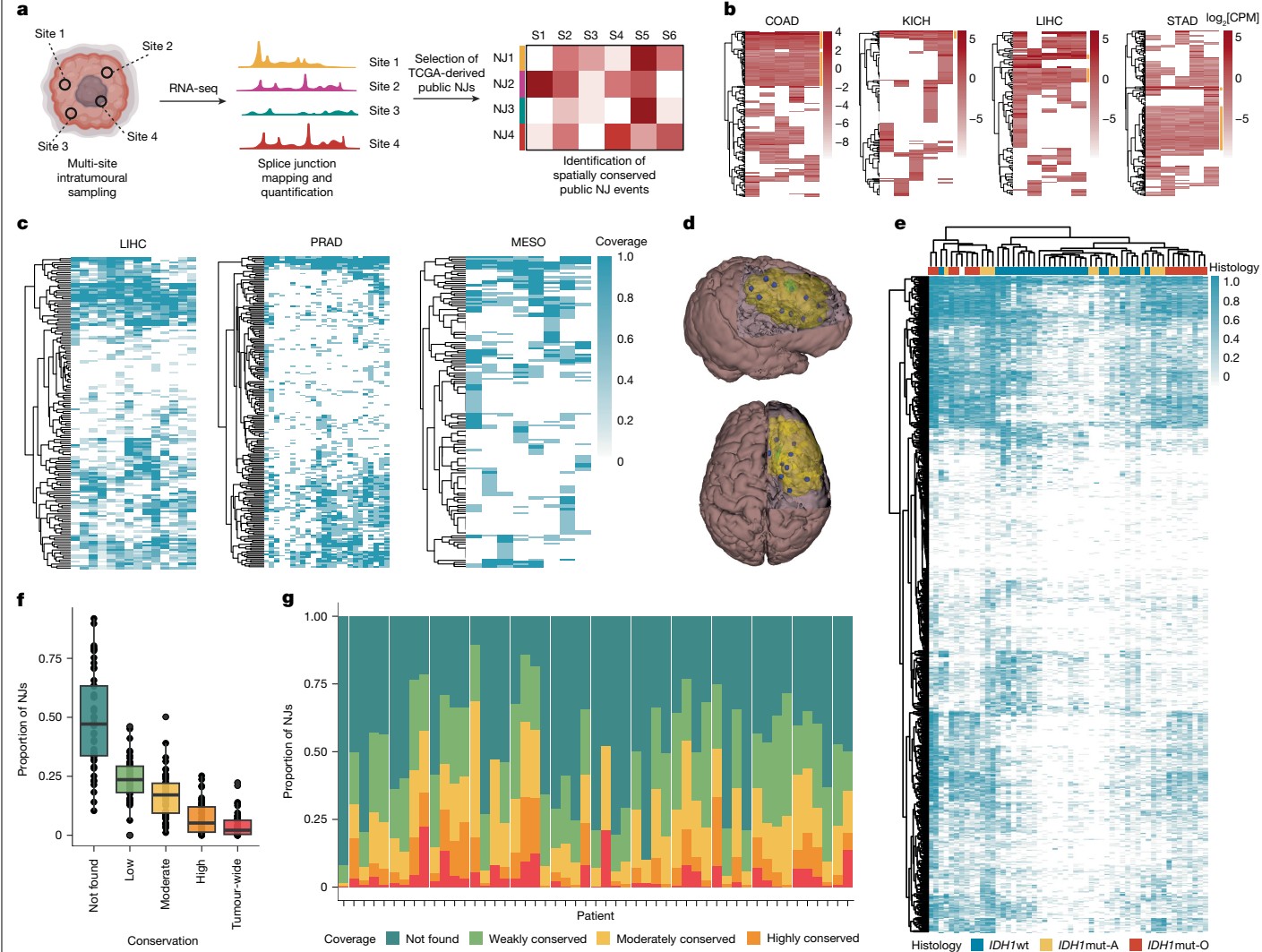

**Fig. 2 | A subset of NJs are expressed tumour-wide. a**, Overview of tumour-wide NJ characterization using RNA-seq data from multiple intratumoural regions in various cancer types. S1–S6 indicate an example numbering of samples isolated per patient. **b**, Heat maps representing $\log_2[\text{CPM}]$ for NJs (rows) across five intratumoural regions in COAD, KICH, LIHC and STAD, with tumour-wide NJs highlighted in yellow. **c**, Heat map illustrating the proportion of intratumoural regions with detectable NJ expression (rows) in LIHC (left), PRAD (centre) and MESO (right). Each column represents a single patient. **d**, Three-dimensional brain and tumour (yellow) models for patient 470. Approximately 10 spatially mapped and maximally distanced biopsies (blue) were taken in each tumour (refer to Supplementary Video 1). **e**, Heat map of

NJ (rows) expression across glioma subtypes: *IDH*wt (blue), *IDH*mut-A (yellow) and *IDH*mut-O (red). Columns represent patients, and cell intensity indicates the percentage of intratumoural regions expressing each NJ. **f**,**g**, NJ ITH in gliomas ($n = 789$) shown using a bar plot (**f**) and parts-of-whole chart (**g**). NJs are classified as: tumour-wide (100% intratumoural regions, red), highly conserved (>70%, orange), moderately conserved (>30% to ≤70%, yellow), or weakly conserved (≥1 region but ≤30%, green). In **f**, the data are represented as box plots, in which the median line represents the 50th percentile. Further statistical details are provided in Supplementary Table 3. **a**, Created in BioRender (credit: D.W.K., https://BioRender.com/h58s281; 2024).

the number of ubiquitously expressed NJs inversely correlated with the number of samples (Extended Data Fig. 2f–h). These findings highlight the critical need for sampling multiple biopsies per tumour to more confidently characterize NJs as tumour-wide.

Hierarchical clustering of our large intratumoural dataset revealed that NJ subsets were associated with either isocitrate dehydrogenase mutant (*IDH*mut) or wild-type (*IDH*wt) subtypes (Fig. 2e). *IDH*mut gliomas exhibited significantly more tumour-wide NJs compared to *IDH*wt gliomas. Although tumour-wide NJs were less common than subclonally expressed NJs (Fig. 2f), at least one tumour-wide NJ was detected in 45 (88.2%) patients (Fig. 2g), with 13 (25.5%) patients expressing more than 50 tumour-wide NJs (Extended Data Fig. 2d). Most TCGA-characterized low-grade glioma (LGG) and glioblastoma (GBM) NJs (774; 98.1%) were detectable in more than 1 tumour region in our dataset, but only 37 (4.7%) NJs were present across all samples in more than 10% of the study

cohort (Extended Data Fig. 2e). These findings indicate that although public NJs are expressed across multiple tumour regions, they are not universally tumour-wide. Combining NJs may allow targeting of the entire tumour landscape.

We next characterized spatially and temporally conserved NJs at metastasis and recurrence, respectively. Analysis of public skin cutaneous melanoma (SKCM) RNA-seq data[35] revealed 13 (9.6%) NJs expressed across metastatic sites in at least 1 patient (Extended Data Fig. 2c). In matched primary–metastasis pairs from TCGA, 43.8% to 72.6% of NJs identified in primary tumours persisted in metastases across colon adenocarcinoma, prostate adenocarcinoma and SKCM cancers (Extended Data Fig. 2i). Similarly, an average of 36.4% of NJs were conserved at recurrence in primary–recurrence pairs from TCGA colon adenocarcinoma, GBM, LGG, liver hepatocellular carcinoma (LIHC) and lung adenocarcinoma (LUAD) cancers (Extended Data Fig. 2j). In our glioma dataset, 79.2% and

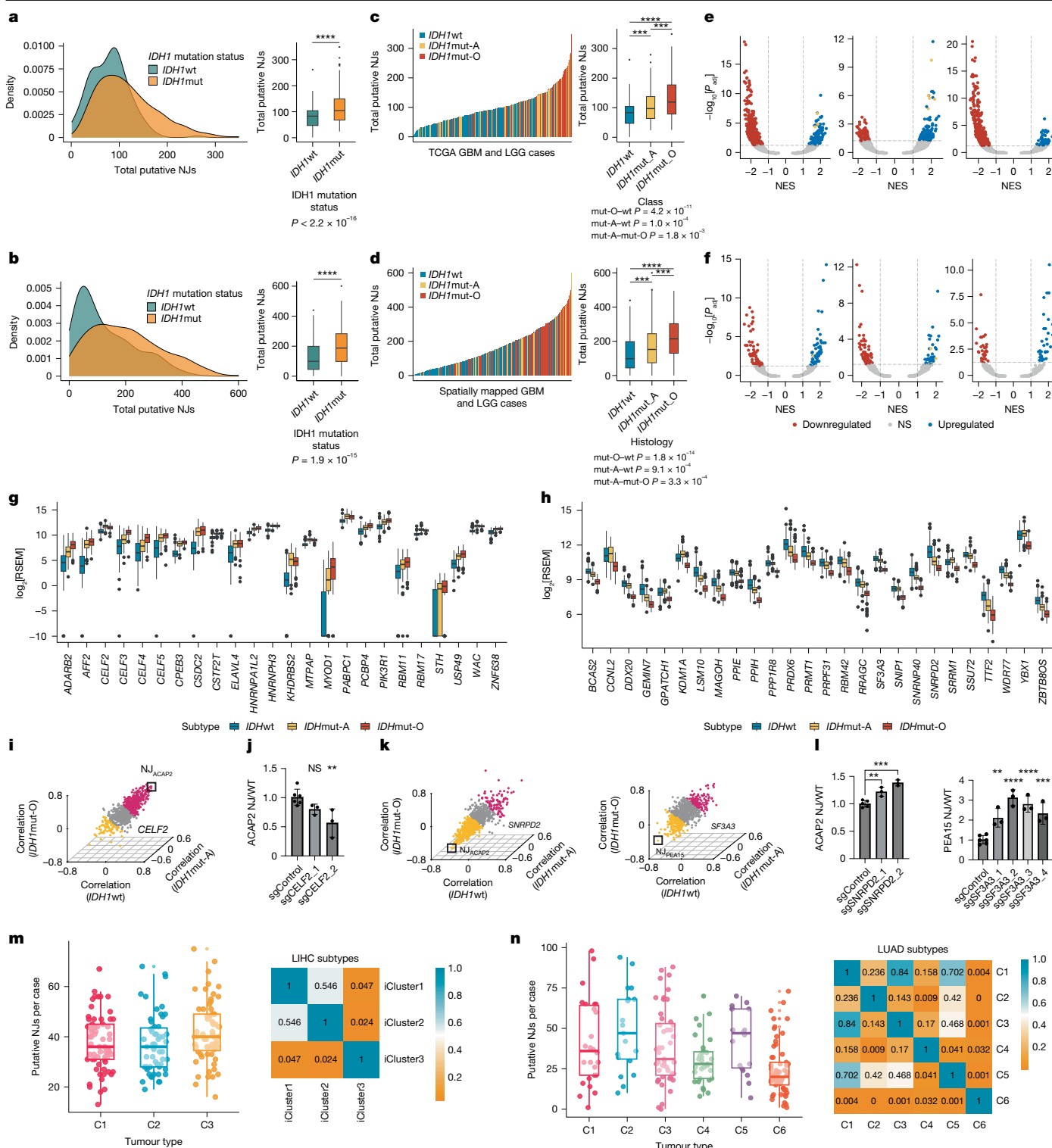

**Fig. 3** | See next page for caption.

82.3% of NJs were conserved in hypermutated and non-hypermutated gliomas, respectively, at recurrence following temozolomide treatment (Extended Data Fig. 2k). Altogether, these findings demonstrate that NJs can persist across both spatial and temporal contexts.

## Tumour subtype factors drive NJ expression

Subtype-specific NJ expression (Fig. 2e) prompted us to investigate splicing machinery dysregulation that may contribute to these

patterns. Although previous investigations suggest that *IDH* mutations may drive splicing aberrations[14], our study revealed additional complexity. *IDH*mut gliomas exhibited significantly more public NJs per case than *IDH*wt gliomas in both TCGA and our spatially mapped datasets (Fig. 3a,b). Among *IDH*mut subtypes, oligodendrogliomas (*IDH*mut-O) had higher NJ expression than astrocytomas (*IDH*mut-A) (Fig. 3c,d). We performed pairwise Pearson correlation analyses to explore whether NJ expression is associated with somatic mutations in commonly mutated RNA splicing factors[36–38] (Extended Data Fig. 3a–c). *FUBP1*, *SF3A1* and

**Fig. 3 | Tumour subtypes demonstrate differential NJ expression.**
**a,b**, Density (left) and box (right) plots showing the total putative NJs expressed in *IDH*mut (orange) and *IDH*wt (green) cases in TCGA GBM and LGG (*IDH*wt, $n = 166$; *IDH*mut, $n = 263$; **a**) and spatially mapped GBM and LGG data (*IDH*wt, $n = 258$; *IDH*mut, $n = 277$; **b**). **c,d**, Histograms and box plots depicting NJ counts in *IDH*wt (blue), *IDH*mut-A (yellow), and *IDH*mut-O (red) in TCGA GBM and LGG (**c**) and in-house GBM and LGG datasets (**d**). **e,f**, Volcano plots illustrating significantly upregulated ($P < 0.05$ and NES $> 1$, blue) and downregulated ($P < 0.05$ and NES $< -1$, red) gene sets comparing *IDH*mut-O versus *IDH*wt (left), *IDH*mut-A versus *IDH*wt (centre), and *IDH*mut-O versus *IDH*mut-A (right). GOBP (**e**) and Gene Ontology Cellular Component (**f**) gene sets were investigated. Splicing-related gene sets are denoted in yellow. NES, normalized enrichment score. **g,h**, Box-and-whisker plots depicting $\log_2$[RNA-seq by expectation–maximization (RSEM)] of splicing-related genes from GOBP sets with significant ($P < 0.05$) $\log_2$[fold expression] differences: increased ($\log_2$[fold increase] $\geq 1.5$) between *IDH*mut-A (yellow) and *IDH*mut-O (red) cases when compared to

*IDH*wt cases (blue) (**g**) and decreased ($\log_2$[fold decrease] $\leq 1.5$) between *IDH*mut-O when compared to *IDH*mut-A and *IDH*wt cases (**h**). **i,k**, Pearson correlation of glioma-specific NJs against *CELF2* (**i**), *SNRPD2* (**k**, left) and *SF3A3* (**k**, right) in *IDH*mut-O ($z$ axis), *IDH*mut-A ($y$ axis) and *IDH*wt ($x$ axis) cases. NJs with correlations of $\geq 0.10$ (purple) or $\leq -0.10$ (yellow) are highlighted, with $NJ_{ACAP2}$ (**i,k** (left)) and $NJ_{PEA15}$ (**k**, right) analysed. **j,l**, Expression of splicing-related genes was assessed in LGG (SF10417; **j**) or GBM (GBM115; **l**) cell lines transduced with dCAS9–KRAB and control single guide RNAs (sgRNAs; $n = 6$), *CELF2* sgRNAs (**j**) *SNRPD2* sgRNAs (**l**, left, $n = 3$) or *SF3A3* sgRNAs (**l**, right, $n = 3$). **m,n**, Box plots (left) and heat maps (right) showing NJ expression per case and Wilcoxon rank-sum test results across iCluster (C) subtypes in TCGA LIHC (iCluster 1, $n = 65$; iCluster 2, $n = 55$; iCluster 3, $n = 63$) (**m**) and LUAD (iCluster 1, $n = 26$; iCluster 2, $n = 19$; iCluster 3, $n = 47$; iCluster 4, $n = 31$; iCluster 5, $n = 18$; iCluster 6, $n = 61$) (**n**). Further statistical details are provided in Supplementary Table 3. NS, not significant; **$P < 0.01$; ***$P < 0.001$; ****$P < 0.0001$.

---

*NIPBL* mutations were highly correlated with the *IDH* mutation, and *FUBP1* mutations were specifically prevalent in *IDH*mut-O gliomas[39]. Despite this, no significant clustering was observed between NJs and *FUBP1*, *SF3A1* or *NIPBL* mutation status (Extended Data Fig. 3d–i).

Dysregulation of individual splicing factors can result in aberrant splicing[37]. To investigate possible drivers of the glioma-subtype-specific NJ expression, we evaluated differentially expressed splicing-related gene sets across three glioma subtypes in TCGA (Extended Data Fig. 4a,b and Supplementary Table 1). Gene set enrichment analysis identified significantly upregulated splicing-related genes in *IDH*mut compared to *IDH*wt gliomas in both Gene Ontology Biological Process (GOBP; Fig. 3e) and Gene Ontology Cellular Component databases (Fig. 3f). When ordered on the basis of NJ expression, splicing-related genes expressed at high levels in both *IDH*mut tumour subtypes largely clustered together, suggesting that they have a role in driving subtype-specific NJ production (Extended Data Fig. 4c–e).

To investigate splicing-related genes driving increased NJ expression in *IDH*mut gliomas (Fig. 3g,h), we selected GOBP splicing-related genes ($n = 24$) with a significant ($P < 0.05$) 1.5-fold increase in expression in *IDH*mut cases compared to the wild type (Fig. 3g). Notably, *CELF2* (ref. 40) was previously reported to generate splice aberrations when overexpressed. Analyses correlating the expression of *CELF2* against the expression of all 789 public NJs identified a greater percentage of NJs whose expression generally increased (average Pearson correlation coefficient of $>0.10$) with the increasing level of *CELF2* expression across all glioma subtypes (Fig. 3i). Of the 789 NJs, 359 (45.5%) increased in expression level with *CELF2* expression, whereas 81 (10.3%) negatively correlated with *CELF2* expression. We performed both CRISPRi-mediated (Extended Data Fig. 5a) and short interfering RNA (siRNA)-mediated (Extended Data Fig. 5b,c) knockdown of *CELF2* in patient-derived *IDH*mut cell lines[41] and investigated the change in expression of $NJ_{ACAP2}$, the NJ most highly correlated with *CELF2* (Fig. 3i). With CRISPRi-mediated knockdown of *CELF2*, we observed a significant decrease in the expression level of $NJ_{ACAP2}$ (Fig. 3j), and siRNA-mediated knockdown demonstrated trends of reduced $NJ_{ACAP2}$ expression (Extended Data Fig. 5d). We characterized 244 NJs significantly upregulated in *IDH*mut compared with *IDH*wt glioma cases ($\log_2$[fold change] $> 1.5$, $P$ value $< 0.05$), a subset of which were detected in other TCGA *IDH*mut cancer types (Extended Data Fig. 5e). RNA-seq analyses demonstrated a decrease in the level of expression of 19 (8.6%) and 28 (12.7%) *IDH*mut-associated NJs, respectively, in oligodendroglioma (SF10417) and astrocytoma (SF10602) cells following *CELF2* knockdown compared to non-treated controls (Extended Data Fig. 5f). A correlative increase was observed in the expression level of a candidate *IDH*mut NJ with increased expression of *IDH*mut-associated splicing-related genes (Extended Data Fig. 5g). These findings suggest that NJ prevalence is regulated by altered expression of RNA-binding proteins in tumour subtypes and modulating these genes alters NJ levels.

Re-examining GOBP splicing-related gene sets (Extended Data Fig. 4c–e) revealed subclusters of genes significantly downregulated in *IDH*mut-O cases. Most of the genes found in these clusters reside on either chromosome 1p or 19q, co-deletion of which is a distinctive diagnostic feature of *IDH*mut-O gliomas. To evaluate whether this downregulation contributes to the characteristic increase in the expression level of putative NJs seen in *IDH*mut-O cases, we selected GOBP splicing-related genes ($n = 26$) with a significant ($P < 0.05$) 1.5-fold decrease in expression in *IDH*mut-O cases compared to both *IDH*mut-A and *IDH*wt cases (Fig. 3h). Of these splicing genes, disruption of normal *SNRPD2* and *SF3A3* expression was previously reported to lead to splicing aberrations[42]. Correlation analysis of *SNRPD2* and *SF3A3* expression against the expression of the 789 NJs across all glioma subtypes supported our hypothesis that decreased *SNRPD2* and *SF3A3* expression levels may contribute to greater NJ expression (Fig. 3k). Of the 789 NJs, 385 (48.8%) showed increased expression with decreasing *SNRPD2* levels, and 93 (11.8%) NJs tended to increase in expression level with increasing levels of *SNRPD2*. Similarly, with decreasing levels of *SF3A3* expression, 178 (22.6%) NJs tended to increase in expression level, and 127 (16.1%) NJs tended to decrease in expression level. We investigated whether the two NJs that showed the strongest inverse correlations with *SNRPD2* and *SF3A3* expression, $NJ_{ACAP2}$ and $NJ_{PEA15}$ (Fig. 3k), respectively, might be causally linked to the expression of these splicing factors. Notably, both CRISPRi and siRNA knockdown of either *SNRPD2* or *SF3A3* in the GBM115 cell line (Extended Data Fig. 5a–c), which contains two copies of chromosomes 1p and 19q, led to significant increases in $NJ_{ACAP2}$ or $NJ_{PEA15}$ levels, respectively (Fig. 3l and Extended Data Fig. 5d). RNA-seq of GBM115 cells treated with siRNA knockdown of *SNRPD2* or *SF3A3* demonstrated similar increases in $NJ_{ACAP2}$ or $NJ_{PEA15}$ expression levels, respectively (Extended Data Fig. 5h). We also characterized 52 *IDH*mut-O-associated NJs significantly upregulated in *IDH*mut-O glioma cases compared to *IDH*mut-A and *IDH*wt gliomas ($\log_2$[fold change] $> 1.5$, $P$ value $< 0.05$). Increased expression levels of 7 (13.5%) and 4 (7.7%) *IDH*mut-O-associated NJs were seen in GBM115 cells treated with *SF3A3* or *SNRPD2* siRNA, respectively (Extended Data Fig. 5i). Although previous studies linked splicing factor mutations to NJs in cancers, our results shed light on a previously undescribed mechanism in which decreased wild-type splicing factor expression can drive NJ formation. These findings suggest that commonly altered components of the RNA splicing machinery in gliomas are mechanistically linked to increased NJ expression.

Finally, we extended our analysis across the remaining TCGA cancer types used in this study to identify tumour subtypes with significantly dysregulated NJ expression. Whereas NJ expression remained relatively consistent across SKCM, kidney renal papillary cell carcinoma, kidney chromophobe and prostate adenocarcinoma cancers (Extended Data Fig. 5j–m and Supplementary Table 2), iCluster 3 in TCGA LIHC and iCluster6 in TCGA LUAD demonstrated significantly differentiated

NJ expression compared with other iCluster subtypes (Fig. 3m,n). Gene set enrichment analysis of the six LUAD iCluster subtypes revealed a decreased level of expression of splicing-related gene pathways. Notably, 23 of these splicing-related gene sets were consistently downregulated in LUAD iCluster 6 compared with all 5 other iCluster subtypes. Together, these results indicate that in addition to splicing factor mutations, dysregulated expression of canonical splicing-related genes can lead to the generation of disease-specific NJs.

## Public NJ-derived RNA and peptides are detectable

We next validated the expression of public NJs and their protein products in cell line transcriptomic and tumour tissue proteomic data, focusing on gliomas owing to their high ITH and poor outcomes. Using RNA-seq data for xenografts derived from patients with GBM ($n = 66$)[43] and LGG ($n = 2$) cell lines[41], we detected 767 (97.2%) and 510 (64.6%) public NJs in GBM and LGG, respectively (Extended Data Fig. 6a,b). To overcome the limitations of bulk RNA-seq, we designed primers spanning a subset of NJs and their flanking exons, performed deep amplicon sequencing, and confirmed mRNA expression of NJ-spanning reads expressed in glioma cell lines (Extended Data Fig. 6c).

To determine whether NJs are translated into proteins, we analysed mass spectrometry (MS) data from patients with glioma ($n = 447$) using publicly available MS datasets[44–46]. This identified neopeptides mapping to 302 (38.3%) unique public NJs (Extended Data Fig. 6d). We confirmed that the peptide sequences span the aberrantly spliced regions with sequence-specific searches in the MS data and subsequent analysis of the resulting MS spectra (Extended Data Fig. 6e,f). Notably, 41.7% of the detected peptides mapped back to NJs that result in frameshifts (Extended Data Fig. 6g), indicating that frameshift-inducing splicing aberrations can lead to detectable translated peptides. Overall, our peptidome analysis determined that NJ-encoding transcripts are actively translated into protein products. Combining RNA-seq and MS results, we selected 192 (24.3%) public NJs expressed across all patient-derived samples for subsequent investigations (Extended Data Fig. 6h). These findings highlight the recurrent nature of public NJs and their role in generating tumour-specific peptides.

## Tumour-wide NJs encode presentable neoantigens

We reasoned that a subset of translated NJs could produce peptides presented as targetable neoantigens[16,17]. To test this, we assessed whether the 789 characterized public NJs can generate peptides loaded onto HLA class I following proteasomal processing. NJ-derived sequences from TCGA were translated in silico to generate a NJ-derived protein dataset. Iterating through all possible $n$-base polypeptides of 8 to 11 amino acids (Extended Data Fig. 6i), we defined tumour-specific $n$-base polypeptides as those absent from a UniProt reference normal human tissue proteome dataset.

Prediction of HLA class I-presented peptides requires incorporating the key aspects of antigen-presentation machinery, including peptide processing and HLA binding. To this end, we integrated two independent prediction algorithms, MHCflurry 2.0 and HLAthena, to identify neoepitope sequences[47,48] (Extended Data Fig. 6j). Candidate $n$-base polypeptides were ranked by their binding potential to the most prevalent HLA-A alleles. Among the 36 predominant HLA-A alleles (Extended Data Fig. 6k,l), our analyses investigated the presentation likelihood of neoantigen candidates by HLA-A*01:01, HLA-A*02:01, HLA-A*03:01, HLA-A*11:01 and HLA-A*24:02. Together, these alleles are expressed by most of the global population[49]. High-binding targets were defined as $n$-base polypeptides scoring in the top 1% with both algorithms (Extended Data Fig. 6m–p) Candidate $n$-base polypeptides yielding these scores ($n = 832$) were retained for downstream analysis (Extended Data Fig. 6q). When these top candidates were mapped to their originating NJs, 315 neopeptide-encoding NJs (NEJs; 39.9% of the

originally characterized public NJs) produced cancer-specific peptides containing these top $n$-base polypeptide candidates. Although a greater number of top-scoring $n$-base polypeptide candidates are generated from frameshifts and alternative exonic 3′ splice sites (Extended Data Fig. 7a,b), presentation scores remained relatively consistent across all frameshift types and mutation types (Extended Data Fig. 7c–f). Cross-referencing the 315 NEJs with the 192 transcriptomically and proteomically validated NJs (Extended Data Fig. 6h) yielded 81 NEJs (Extended Data Fig. 7g), with many encoding multiple strongly predicted candidates. We focused our downstream analyses on 32 candidate NEJs that predicted to bind strongly to HLA-A*02:01 owing to this allele's high prevalence across North American and European populations and the ability to benchmark to other neoantigen studies (Extended Data Fig. 7h). Examining the ITH of these 32 NEJs in spatially mapped samples (Extended Data Fig. 7i) revealed high intratumoural conservation for most of these NEJs, particularly the NEJ located in *GNAS* (NEJ$_{GNAS}$) that encodes an A3 loss of two nucleotides. These findings demonstrate that intratumourally conserved public NEJs may generate HLA-presented neopeptides.

## Identification of NEJ-reactive TCRs

We next sought to determine whether NEJ-derived neopeptides can drive T cell responses. We performed in vitro sensitization (IVS) to identify neoantigen-reactive CD8+ T cell populations from healthy-donor-derived peripheral mononuclear cells (PBMCs; Fig. 4a). We focused our initial analysis on a subset ($n = 4$) of our 32 top NEJ candidates predicted to generate high-affinity binders to HLA-A*02:01 (Extended Data Fig. 6k–m). We therefore performed IVS of naive CD8+ T cells against neopeptide-pulsed autologous monocyte-derived dendritic cells collected from HLA-A*02:01+ healthy donors ($n = 5$) to retrieve TCR gene sequences that confer specificity against these neoantigens. Subsequent interferon-γ (IFNγ) enzyme-linked immunosorbent assay (ELISA) assays on the corresponding antigen-presenting cell (APC) and CD8+ T cell (APC:CD8+) conditions revealed neoantigen-reactivity in two out of four of the public NEJ-derived neoantigens: NeoA$_{RPL22}$ and NeoA$_{GNAS}$ (Fig. 4b). Both neoantigens are detectable in publicly available MS data (Extended Data Fig. 6e,f). NeoA$_{GNAS}$ results in an A3 loss of two nucleotides that generates a frameshift and a premature stop codon. NeoA$_{RPL22}$ encodes an in-frame A3 loss of six nucleotides, resulting in a loss of two amino acids in an α-helix (Extended Data Fig. 7j). These results additionally indicate that NEJ-reactive CD8+ T cells can exist in the naturally occurring human T cell repertoire.

To retrieve TCR gene sequences that confer reactivity to these neoantigens, we repeated the peptide-pulsed-APC:CD8+ T cell co-culture on NeoA$_{RPL22}$- and NeoA$_{GNAS}$-reactive CD8+ T cell populations and performed combined single-cell V(D)J and RNA-seq. Neoantigen-reactive TCR clonotypes were associated with significantly elevated *IFNG*, *TNFA* and *GZMB* transcript levels in neoantigen-peptide-specific manners. Using this method, we identified seven NeoA$_{RPL22}$-reactive TCRs, two from donor 3 (TCR$_{R3.7}$ and TCR$_{R3.9}$) and five from donor 4 (TCR$_{R4.5}$, TCR$_{R4.6}$, TCR$_{R4.7}$, TCR$_{R4.9}$ and TCR$_{R4.11}$), and one NeoA$_{GNAS}$-reactive TCR from donor 4 (TCR$_{G4.1}$; Fig. 4c). Although only one NeoA$_{GNAS}$-reactive TCR clonotype was characterized, this same clonotype was the most proliferated TCR clone, expanding to more than 4% of the TCR repertoire in the CD8+ T cell population (Fig. 4d). The expansion of neoantigen-reactive CD8+ T cell clones suggests a strong immunogenic proponent of these two neoantigens.

## NEJ-reactive TCRs recognize HLA-presented neoantigens

To determine the peptide-specific reactivity of identified TCR$_{R3.9}$- and TCR$_{G4.1}$-reactive T cell clones, we transduced TCR-null triple-reporter (TR) Jurkat76 cells which express the CD8α–CD8β heterodimer

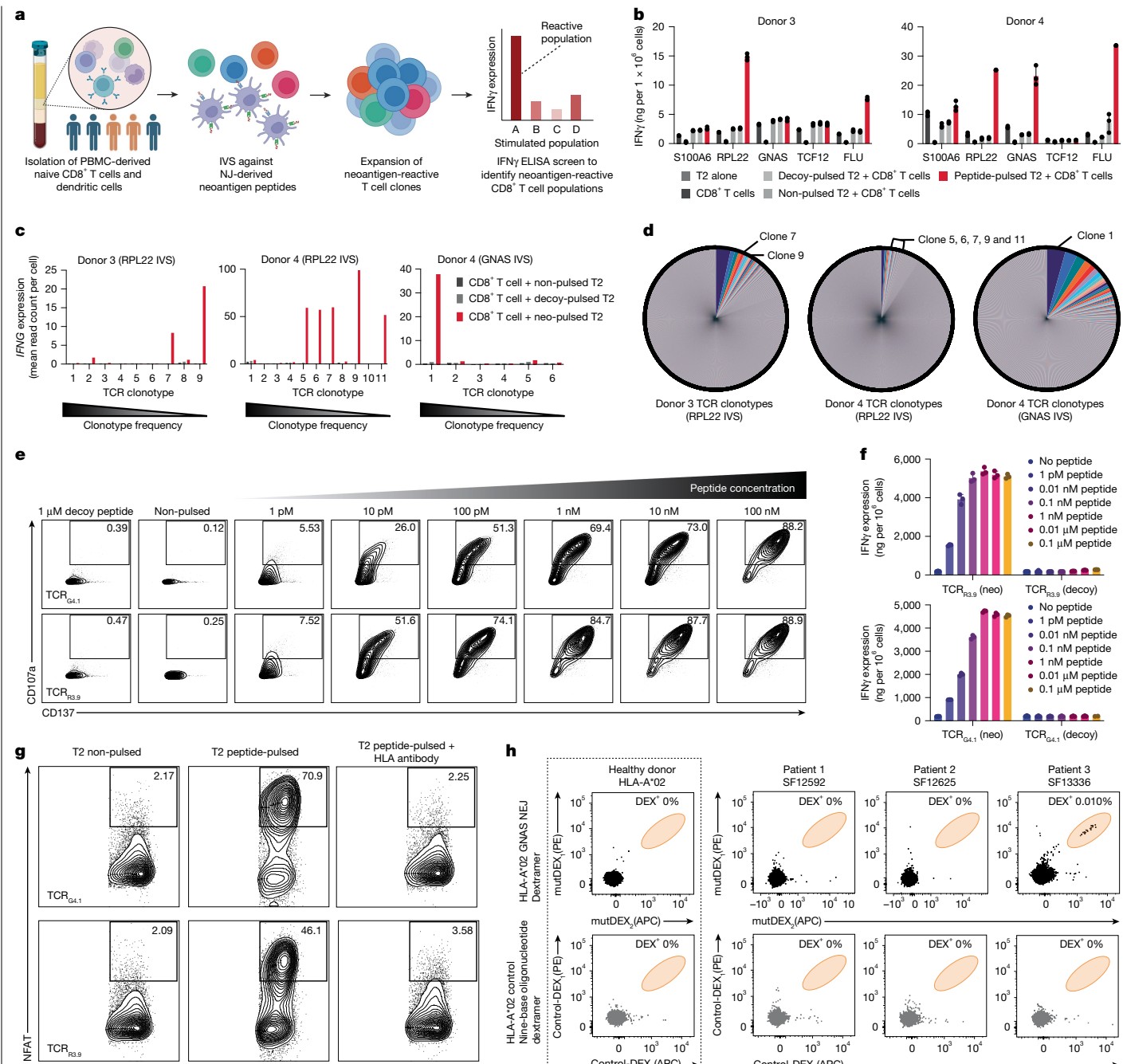

**Fig. 4 | TCRs specifically react to NEJ-derived neoantigens. a**, Pipeline overview for identifying T cell populations reactive to NEJ-derived neoantigen through IVS of CD8[+] T cells derived from PBMCs from healthy donors against APC-presented neopeptides. **b**, IFNγ ELISA of reactive CD8[+] T cell populations (*n* = 3) following IVS with neoantigen. **c**, 10× V(D)J sequencing shows *IFNG* signatures of highly proliferated TCR clonotypes cultured with T2 cells pulsed with neoantigen (coloured), control peptide (light grey) or no peptide (dark grey). Specific TCR clonotypes are highlighted for NeoA_RPL22 and NeoA_GNAS reactivity in donors 3 (left) and 4 (centre and right). **d**, Clonotype frequency analysis of TCR clones in CD8[+] T cells from donors 3 (left) and 4 (centre and right) following IVS with NeoA_RPL22 or NeoA_GNAS. Neoantigen-reactive TCR clones are denoted by text. **e**, NeoA_GNAS-specific (top) and NeoA_RPL22-specific (bottom) TCR-transduced PBMC-derived CD8[+] T cells were activated against neoantigen-pulsed T2 cells in a dose-dependent manner. TCR-transduced cells were also co-cultured with control-peptide-pulsed T2 cells at the highest dose concentration (1 μM). PBMC-derived CD8[+] T cells were stained with CD107a and CD137 antibodies, and surface expression of the TCR coactivation markers was

analysed by flow cytometry. The percentages of activated (CD107a and CD137 antibody-stained) CD8[+] T cells detected in flow analysis are indicated by the numbers within the box. **f**, IFNγ ELISA (*n* = 3) of NeoA_GNAS-reactive (top) and NeoA_RPL22-reactive (bottom) TCR-transduced CD8[+] T cells co-cultured with dose-dependent neoantigen (neo)-pulsed (left) and control-peptide-pulsed T2 cells (right). **g**, NeoA_GNAS-specific (top) and NeoA_RPL22-specific (bottom) TCR-transduced triple-reporter Jurkat76 cells were co-cultured with non-pulsed T2 cells (left), 0.1 μM neoantigen-pulsed T2 cells (centre) or 0.1 μM neoantigen-pulsed T2 cells treated with pan-HLA class I blocking antibody (right). Cells were stained with CD3 antibody, and TCR activation was evaluated by NFAT–GFP activity. The percentages of CD3[+] and NFAT-GFP[+] TR Jurkat76 cells detected in flow analysis are indicated by the numbers within the box. **h**, NeoA_GNAS-dextramer staining of bulk CD8[+] T cells derived from an HLA-A*02:01 healthy donor (left) and patients with glioma (right) following two cycles of NeoA_GNAS IVS. Further statistical details are provided in Supplementary Table 3. **a**, Created in BioRender (credit: D.W.K., https://BioRender.com/z79j394; 2024).

(Jurkat76/CD8) or PBMC-derived CD8[+] T cells with lentiviral vectors encoding the retrieved TCR α- and β-chains. The TR Jurkat76/CD8 cells have response elements for NFAT, NF-κB and AP-1 that drive expression of eGFP, CFP and mCherry, respectively[50] (Extended Data Fig. 8a). TCR-transduced TR Jurkat76 cells cultured with T2 cells pulsed with varying concentrations of neoantigen peptide demonstrated dose-dependent reactivity (Extended Data Fig. 8b–d). Both TCRs demonstrated nanomolar-level neoantigen recognition, illustrating a relatively high functional avidity of the corresponding TCRs. The antigen-specificity of these receptors was supported by negligible TCR activation in the presence of supraphysiologic levels of the control peptide (1 μM). TCR-transduced PBMC-derived CD8[+] T cells exhibited similar dose-dependent neoantigen-specific behaviour (Fig. 4e,f). TCR-transduced CD8[+] T cells were stained for surface expression of the T cell activation and degranulation markers, CD137 and CD107a, to quantify markers of T cell activation and effector function, respectively. T cell activation was observed at neoantigen-peptide concentrations as low as 1 pM (Fig. 4e). Similarly, IFNγ and tumour necrosis factor (TNF) expression levels measured by ELISA suggested strong potency of both TCRs as indicated by their half-maximal effective peptide concentrations ($EC_{50}$ values) of between 0.01 and 0.1 nM (Fig. 4f and Extended Data Fig. 9a). Treatment of neoantigen-pulsed T2 cells with an HLA-blocking antibody before co-culture with the TCR-transduced TR Jurkat76 cells validated that neopeptide T cell activation is HLA dependent (Fig. 4g).

Next we performed alanine scanning mutagenesis to determine whether either NEJ-reactive TCR can recognize peptides derived from off-target normal human proteins. TCR-transduced triple-reporter Jurkat76/CD8 cells were cultured against residue-substituted neoantigen isoforms, and key residues were defined as those that resulted in diminished TCR activation (Extended Data Fig. 9b). Alterations in the recognition of a variant peptide indicate that the substituted residue is critical for TCR recognition. Referencing the peptide recognition motif of each TCR to a normal human proteome library (UniProt Proteome ID: UP000005640) demonstrated that no known human proteins share the key residues required for TCR recognition. Together, our results reveal TCRs that recognize NEJ-derived public neoantigens with robust sensitivity and highlight a potential immunotherapeutic approach utilizing TCR-engineered T cells to target this new class of shared neoantigens.

Finally, using PBMCs from HLA-A*02:01[+] patients with gliomas known to express NEJ$_{GNAS}$ (Extended Data Fig. 6m), we tested whether NEJ-reactive CD8[+] T cells naturally occur. Short-term IVS of bulk PBMC samples with NEJ$_{GNAS}$ led to the detection of a response in one of three patients with glioma with no immunogenicity against an irrelevant HLA-A*02-restricted neoantigen dextramer control (Fig. 4h). These findings further support the immunogenicity and potential clinical application of targeting NEJ-derived neoantigens.

## NEJ-derived neoantigens are processed and HLA-presented

Next we tested whether NEJ-derived transcripts generate peptides that are functionally presented by HLA and recognized by reactive TCRs. We evaluated the presentation of NEJ-derived neoantigens using two approaches: functional TCR recognition and HLA immuno-precipitation followed by liquid-chromatography with tandem MS (Fig. 5a). To determine whether the NEJ transcript expression leads to immune recognition, we co-cultured COS-7 cells transfected with the HLA-A2 and full-length mutated transcript together with either TCR-transduced TR Jurkat76 or CD8[+] T cells. TCR$_{R3.9}$- and TCR$_{G4.1}$-transduced TR Jurkat76 and CD8[+] T cells reacted against COS-7 cells transfected with their respective neoantigen, demonstrating endogenous processing and presentation of the public NEJs (Fig. 5b,c). We then performed affinity-column-based immunopurification of

HLA-I ligands on COS-7 cells co-transfected with HLA and mutant NEJ transcript. The MS analysis identified the same NeoA$_{GNAS}$ peptide as the highly abundant HLA-A2-bound peptide with high-confidence. Likewise, both NeoA$_{RPL22}$ neopeptides were detected with high confidence on COS-7 cells co-transfected with HLA-A*02:01 and NEJ$_{RPL22}$ with the higher scoring NeoA$_{RPL22}$ nine-amino acid polypeptide identified with higher relative abundance (Fig. 5d). Furthermore, we could detect the HLA-A*02:01-restricted NeoA$_{GNAS}$ peptide in an unmodified GBM cell line (GBM115; Fig. 5e). This finding demonstrates that physiologic levels of NEJ expression in tumour cells are sufficient to generate an NEJ-derived neoantigen. Together, these experimental observations confirm our in silico predictions for proteasomal processing and HLA binding (Extended Data Fig. 6l).

## NEJ-specific T cells mediate tumour cytotoxicity

On the basis of the sensitivity of the neoantigen-specific TCRs we identified (Fig. 4e) and the endogenous presentation of NEJ-derived neoantigens (Fig. 5e), we hypothesized that public NEJ-expressing tumour cells would be susceptible to the cytotoxic effects of TCR-transduced T cells. We evaluated the cytotoxicity of TCR-transduced CD8[+] T cells against HLA-A*02:01[+] tumour cells endogenously expressing NEJ$_{RPL22}$ and NEJ$_{GNAS}$. As a positive control, we used neoantigen-peptide-pulsed tumour cells to define maximum cell killing. At a 1:1 effector/target ratio, TCR$_{R3.9}$ and TCR$_{G4.1}$-transduced CD8[+] T cells mediated TCR-dependent cytotoxicity against GBM115 cells (Fig. 5f). TCR$_{G4.1}$-transduced CD8[+] T cells mediated comparable levels of tumour killing against a second GBM cell line, GBM102, and two melanoma cell lines, RPMI-7951 and WM-266-4 (Extended Data Fig. 10a). Adding an HLA-I blocking antibody partially blocked killing compared to an isotype control, verifying that tumour cell killing is initiated by TCR recognition of the HLA–peptide complex (Fig. 5g). Co-culture of TCR$_{G4.1}$-transduced CD8[+] T cell with an HLA-A2[−], NEJ$_{GNAS}$-expressing GBM cell line (Mayo, patient-derived xenograft, GBM39) revealed cytotoxicity only when the gene encoding HLA-A*02:01 was transduced (Fig. 5h). These results illustrate that the recognition and killing of NEJ-expressing tumour cells is mediated by HLA-dependent neoantigen presentation. Relative to that on non-transduced CD8[+] T cells, increased surface expression of CD137 on TCR-transduced CD8[+] T cells co-cultured with tumour cells further confirmed neoantigen-specific T cell activation (Extended Data Fig. 10b–d). Notably, TCR-transduced CD8[+] T cells co-cultured with tumour cells demonstrated significantly elevated levels of secreted granzyme B relative to their non-transduced T cell conditions (Fig. 5i), illustrating the mechanism for our observed neoantigen-specific cytotoxicity. Elevated levels of secreted IFNγ, interleukin-2 (IL-2) and TNF further support neoantigen-specific CD8[+] T cell activation (Extended Data Fig. 10e–g). Together, these data indicate that NEJs are endogenously processed and presented at sufficient levels to enable tumour cytotoxicity by neoantigen-specific CD8[+] T cells.

## Discussion

Our analysis of cases with multi-site samples indicated the expression of NEJ$_{GNAS}$ and NEJ$_{RPL22}$ across multiple samples in the same tumour. Most notably, NEJ$_{GNAS}$ was expressed tumour-wide in diverse tumour types, including glioma, mesothelioma, prostate cancer and hepatocellular carcinoma (Fig. 1h). The discovery of a targetable tumour-wide neoantigen in GBM provides a new potential therapeutic approach for this disease. The higher prevalence of NEJ$_{GNAS}$ detection compared to NEJ$_{RPL22}$ may stem from the higher transcript expression level of GNAS in tumours, enhancing its immunogenicity and tumour-specific killing by TCR$_{G4.1}$ (Fig. 5f,g). Notably, circulating NeoA$_{GNAS}$-reactive CD8[+] T cells were detected in an HLA-A*02:01[+] patient with an NEJ$_{GNAS}$-expressing glioma (Fig. 4h). Naturally presented neoantigens on HLA do not always generate detectable T cell responses in patients with cancer[11,51,52];

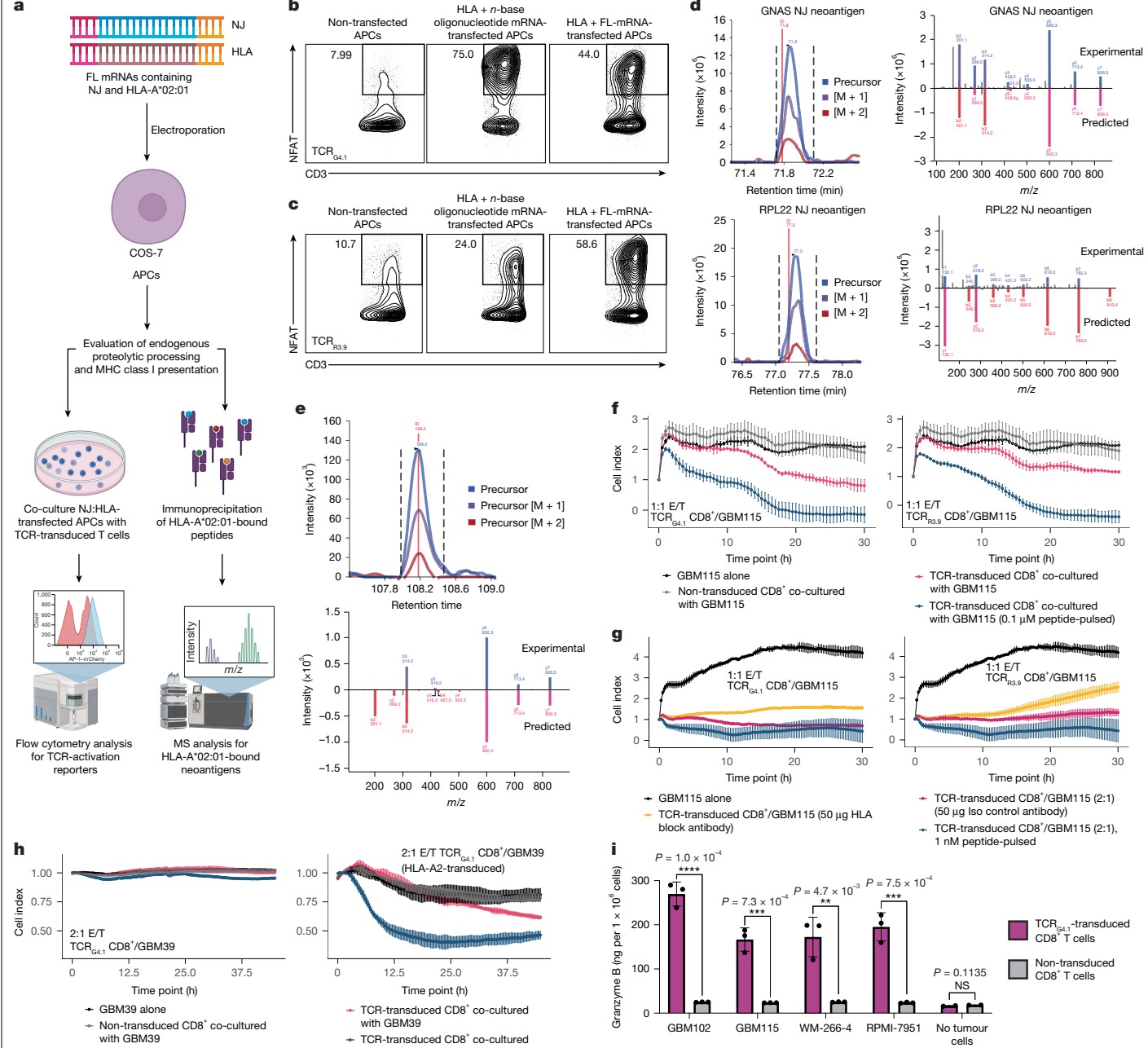

**Fig. 5 | NEJ-derived neoantigens elicit TCR-mediated tumour-specific killing through HLA presentation. a**, Pipeline overview for validating endogenous proteolytic cleavage and subsequent HLA presentation. HLA-null APCs (COS-7) were electroporated with mRNAs encoding full-length (FL) mutant protein or neoantigen *n*-base polypeptides alongside HLA-A*02:01. TCR activation was quantified using neoantigen-specific TCR-transduced triple-reporter Jurkat76 or CD8 cells through flow cytometry. HLA-I-bound peptides were validated by immunoprecipitation with tandem MS. **b,c**, NFAT–GFP flow cytometry results showing TCR activation of NEJ*GNAS*-specific (**b**) and NEJ*RPL22*-specific (**c**) triple-reporter Jurkat76 cells co-cultured with COS-7 cells expressing the mutant *n*-base-polypeptide sequence and HLA-A*02:01 (centre), full-length mutant gene and HLA-A*02:01 (right), or neither (left). The percentages of CD3+ and NFAT-GFP+ TR Jurkat76 cells detected in flow analysis are indicated by the numbers on the plot. **d,e**, MS spectra confirming HLA-A02:01-bound NEJ*GNAS*-derived (**d** (top),**e**) and NEJ*RPL22*-derived (**d**, bottom) neoantigens in transfected COS-7 cells (**d**) and non-transfected GBM115 tumour cells (**e**). **f**, Cytotoxic killing of GBM115 cells by NEJ*GNAS*-derived (left; coloured), NEJ*RPL22*-derived

(right; coloured) neoantigen-specific TCR-transduced, non-transduced (grey) CD8+ T cells, or no CD8+ T cells (black) (*n* = 3) using an xCELLigence assay. Tumour cell death is shown as a reduction in cell index, with T cells killing both untreated and peptide-pulsed tumour cells. TCR-transduced CD8+ T cells were co-cultured with GBM115 tumour cells that were untreated (red) or pulsed with 0.1 μM of the corresponding neoantigen peptide (blue). **g**, xCELLigence live-cytotoxicity assay of CD8+ T cells co-cultured with GBM115 tumour cells incubated with anti-HLA-I antibody (yellow, *n* = 3), isotype control antibody (purple, *n* = 3) or 1 nM of the neoantigen peptide (blue, *n* = 3). NEJ*GNAS*-specific (left) and NEJ*RPL22*-specific (right) CD8+ T cells were cultured against GBM115. **h**, xCELLigence live-cytotoxicity assay of HLA-A*02:01−, parental GBM39 cells (left) or HLA-A*02:01-transduced GBM39 cells (right) co-cultured with non-transduced or NEJ*GNAS*-TCR-transduced CD8+ T cells (*n* = 3). **i**, ELISA readout of secreted granzyme B by NEJ*GNAS*-specific (purple) or non-transduced (grey) CD8+ T cells when cultured with tumour cell lines (*n* = 3). Further statistical details are provided in Supplementary Table 3. **a**, Created in BioRender (credit: D.W.K., https://BioRender.com/x48d520; 2024).

however, once reactive TCRs are cloned, TCR-redirected autologous T cells can effectively recognize the tumour cells harbouring the relevant mutations[11].

We also investigated whether dysregulated-splicing-related gene expression in *IDH*mut gliomas correlate with increased NJ production compared to *IDH*wt gliomas. *IDH* mutations are prevalent in other cancers, including acute myeloid leukaemia, cholangiocarcinoma, chondrosarcoma, sinonasal undifferentiated carcinoma and angioimmunoblastic T cell lymphoma. Our study demonstrated dysregulation in splicing factor expression in different disease types and that these aberrations may contribute to significant changes in NJ production. In the case of *IDH*mut-O, lower *SF3A3* and *SNRPD2* expression levels are probably due to the characteristic co-deletion of chromosomes 1p and 19q, respectively, and targeted knockdown in cells with intact 1p and 19q increased NJ expression. This suggests that components of the RNA splicing machinery are mechanistically linked to the generation of NJs. Future studies identifying and targeting splicing-related genes associated with NEJ$_{GNAS}$ and NEJ$_{RPL22}$ expression could increase their expression level for improved therapeutic response.

Although HLA class II-restricted neoepitopes can drive CD4$^+$ T cell responses, limitations in current HLA-II binding prediction prevent their assessment in our study. Similarly, our study does not investigate surface-bound NJ-derived neoantigens as we found they were difficult to characterize[20]. We focused on neopeptides that bind to HLA-A*02:01 as a proof-of-concept. However, future studies could include candidates predicted to bind to other prevalent HLA class I alleles to expand the repertoire of targetable neoantigens and the diversity of patients who might benefit.

The most comprehensive analysis of ITH (average of 10 intratumourally mapped samples) was conducted using GBM and LGG samples. To fully validate NJs and their corresponding neoantigens as tumour-wide across other cancer types, we will need a greater set of intratumoural sites per patient, including a temporal and wide anatomical distribution to maximally represent the evolving tumour. Finally, we did not assess the biological contribution of the studied NEJs to the malignant phenotype.

In conclusion, our study highlights that RNA splicing aberrations are a robust source of intratumourally conserved public TSAs that the immune system can recognize. The ability to target tumour-wide neoantigens with engineered T cells enables a powerful therapeutic approach that could tackle the substantial clinical challenge of ITH. Ultimately, the results from our study could allow us to design effective vaccine panels comprising tumour-wide neoantigen targets and to engineer T cell-based modalities that target tumour-wide splice-derived antigens across a wide range of cancer types.

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

## Methods

### Human clinical datasets

The intratumoural multi-region sampling cohort for various cancer types utilizes RNA-seq data from the following studies: this paper, for multi-region sampling of GBM and LGG; ref. 24, for multi-region sampling of hepatocellular carcinoma; ref. 23, for multi-region sampling of hepatocellular carcinoma, STAD, renal cell carcinoma and COAD; ref. 22, for multi-region sampling of prostate cancer; and ref. 29, for multi-region sampling of MESO.

Analysis of NJ expression in multi-region samples was conducted immediately with our NJ prediction pipeline if the FASTQ file was available. If RNA-seq data were available only in BAM format, the sequencing file was first converted into FASTQ format utilizing the Picard software (version 2.7.7a). NJ prediction is detailed in the 'Characterization of public NJs' section of the Methods.

### Data download

Bulk RNA-seq data for GBM ($n = 167$), LGG ($n = 516$), LUAD ($n = 517$), LUSC ($n = 501$), MESO ($n = 516$), LIHC ($n = 371$), STAD ($n = 415$), KIRC ($n = 533$), KIRP ($n = 290$), KICH ($n = 66$), COAD ($n = 458$) and PRAD ($n = 497$) samples were downloaded from TCGA in FASTQ format. Download of intratumoural multi-region sampling sequencing data is detailed in the previous section. Similarly, bulk RNA-seq data for 9,166 normal tissue samples in FASTQ format were downloaded from the GTEx repository. Bulk RNA-seq data for 66 patient-derived GBM cell lines were received from the Mayo Clinic Brain Tumor Patient-Derived Xenograft National Resource[43]. Proteomics data for 100 GBM samples were downloaded from the Clinical Proteomic Tumor Analysis Consortium[44].

### RNA-seq alignment

All downloaded RNA-seq datasets were individually aligned using a STAR aligner-based processing pipeline. Using the STAR software (version 2.7.7a), we constructed a genome index containing non-annotated junctions through the initial alignment pass of the input data. The complete set of command line parameters was as follows: --runThreadN 1 \ --outFilterMultimapScoreRange 1 \ --outFilterMultimapNmax 20 \ --outFilterMismatchNmax 10 \ --alignIntronMax 500000 \ --alignMatesGapMax 1000000 \ --sjdbScore 2 \ --alignSJDBoverhangMin 1 \ --genomeLoad NoSharedMemory \ --limitBAMsortRAM 80000000000 \ --readFilesCommand gunzip -c \ --outFilterMatchNminOverLread 0.33 \ --outFilterScoreMinOverLread 0.33 \ --sjdbOverhang 100 \ --outSAMstrandField intronMotif \ --outSAMattributes NH HI NM MD AS XS \ --limitSjdbInsertNsj 2000000 \ --outSAMunmapped None \ --outSAMtype BAM SortedByCoordinate \ --outSAMheaderHD @HD VN1.4 \ --twopassMode Basic \ --outSAMmultNmax 1 \ and aligned using the GRCH37 STAR index file.

### TCGA sample selection and gene expression quantification

TCGA tumour samples with an absolute tumour purity greater than 0.60 were retained for downstream in silico analysis[18,19]. We selected non-mitochondrial, protein-coding transcripts defined by the Ensembl *Homo sapiens* GRCH37.87 gene annotation gene transfer format (GTF) file and utilized this curated list to select and retain protein-coding transcript isoforms in the TCGA RNA-seq data. Transcript-level expression data ($\log_2$[RSEM transcripts per million + 0.001]) for all TCGA samples were downloaded from the University of California, Santa Cruz Xena Toil pipeline and transformed into standard TPM values. Protein-coding transcript isoforms with a median TPM ≥ 10 were retained for downstream analysis. In the case of glioma TCGA cases, subsequent expression data in TPM were subset into six disease-type categories: all cases ($n = 429$), GBM cases ($n = 115$), LGG cases ($n = 314$), *IDH*wt cases ($n = 166$), *IDH*mut-A cases ($n = 140$) and *IDH*mut-O cases ($n = 123$). Protein-coding transcript isoforms with a median TPM ≥ 10 in at least one of the six disease types were retained for further analysis.

### Characterization of public NJs

For public cancer-specific splicing event counting, we designed a custom R script that detected and quantified non-annotated, cancer-specific splicing events found across each corresponding patient cohort. From the output files derived from STAR aligner in the previous step, alternative splicing events were quantified in detected junction counts in the corresponding sj.out.tab file. We removed splicing events detected in the GRCh37.87 GTF sj.out.tab (GENCODE v33) file to define non-annotated splicing junctions. Non-annotated splicing junctions that overlap non-mitochondrial, protein-coding genes identified in the previous step were retained for continued analytical processing. We removed all splicing junctions with fewer than 10 of their target spliced reads (count) or fewer than 20 total spliced reads (depth) over the whole cohort. Similarly to previous studies[14], we computed spliced frequency as the sum of the total number of target spliced reads divided by the collective sum of spliced reads from the target and canonical junctions. Splicing junctions with a read frequency greater than 1% were retained for downstream analyses. We defined public splicing junctions as ones that were putatively expressed with the aforementioned criteria of total read count, read depth and read frequency across at least 10% of the studied patient cohort and retained those for further analysis. To characterize cancer-specific splicing events, otherwise known as NJs, we removed all junctions that were putatively expressed with the same parameters in more than 1% of GTEx normal samples.

### Detection of cancer-specific intron retention events

Intronic splicing events were detected and characterized using IRFinder v1.2.3. RNA-seq data from TCGA (GBM and LGG) and GTEx (central nervous system) aligned to GRCh37 (hg19) were imported into the software for the detection of intron retention events. General linear model-based analysis was used for differential intron retention assessment. The intron retention ratio is calculated as (intronic reads)/sum(intronic reads, normal spliced reads). Significant intron retention changes are defined as: no less than 10% in both directions; and adjusted *P* values less than 0.05. An intron retention event's PSR in TCGA or GTEx is defined as the number of cases that fulfil these criteria divided by the total number of cases in the cohort. Putative cancer-specific intron retention NJs are characterized as intron retention events with a TCGA PSR ≥ 0.10 and a GTEx PSR < 0.01.

### Transcriptomic validation of expressed NJs

**Detection of expressed NJs in patient-derived GBM and LGG cell lines.** RNA-seq data for cell lines derived from xenografts from patients with GBM were downloaded from the Mayo Clinic Brain Tumor Patient-Derived Xenograft National Resource. Patient-derived LGG cell lines were generated from surgically resected specimens in the Neurological Surgery Brain Tumor Center at the University of California, San Francisco (UCSF)[41]. RNA-seq data from GBM and LGG cell lines were aligned and processed as described above. Public NJs with splice junction CPM of >0 are considered detectable in cell line-derived RNA-seq data.

**Detection of expressed NJs in multi-region cases.** In our cohort of spatially mapped glioma cases, approximately ten or more maximally distanced anatomical biopsies were collected from each patient, allowing for intratumoural assessment of genetic heterogeneity through bulk RNA-seq and whole-exome sequencing. Multi-region sequencing data of various other cancer types vary in the number of sampled regions per tumour and are detailed in the corresponding references (Extended Data Fig. 2). RNA-seq data collected from each multi-region sample were processed and aligned as described above. We searched for putative NJs previously characterized from TCGA in each multi-region sampling dataset. Public NJs with CPM > 0 were considered detectable. Public NJs with putative expression (≥10 spliced reads) in two or more

mapped samples in the same case are considered spatial-conserved NJs. NJs detected in all multi-region samples in the same tumour are considered tumour-wide NJs.

## Proteomic validation of expressed NJ-derived peptides

From the putative NJs detected in the above pipeline, we generated a database of all plausible polypeptides derived from all NJs. NJ-encoding transcripts were generated by mapping the junction coordinates to an hg19 human genome assembly in the Ensembl annotation database (AH13964, EnsDb.Hsapiens.v75). Prediction of NJ-derived amino acid sequences was subsequently performed, and appropriately translated sequences (methionine starting residue, removal of sequences following first stop codon) were retained for downstream $n$-base-polypeptide iteration. To detect NJ-derived polypeptides in GBM cases, we analysed RAW files of GBM and LGG MS data housed in the Clinical Proteomic Tumor Analysis Consortium ($n = 99$), ref. 45 ($n = 99$), ref. 53 ($n = 92$) and ref. 54 ($n = 84$). MaxQuant (v1.6.17.0) was used to identify tryptic sequences from the corresponding MS datasets. Predicted NJ-derived peptides, decoy sequences and a human reference proteome (UniProt Proteome ID: UP000005640) were input as a FASTA file into MaxQuant, and tryptic sequences derived from the input file were matched against the publicly available MS databases. Cancer-specific peptides spanning NJ-derived protein sequences were considered MS-confirmed. The relative detection levels of the NJ-derived peptides and normal-tissue-derived peptides were evaluated by their $\log_2$[peak intensities]. Aside from the default settings, the following commands and parameters were modified and used for MS analysis in MaxQuant: Digestion mode = Trypsin/P; Max missed = 3; Minimum peptide length = 5; Minimum peptide length for unspecific search = 5.

## Peptide processing and HLA binding and presentation predictions

Cancer-specific transcripts with associated NJs were translated in silico into their corresponding amino acid sequences. A library of all possible peptides of 8 to 11 amino acids in length was then generated, and cancer-specific sequences were selected by removing those detectable in normal-tissue peptide isoforms in a reference human proteome dataset (UniProt Proteome ID: UP000005640). All cancer-specific peptides with their upstream and downstream flanking sequences (maximum flanking length of 30 amino acids) were independently analysed and ranked by MHCflurry 2.0 and HLAthena MSiC. HLA-I binding affinity was assessed against HLA-A*01:01, HLA-A*02:01, HLA-A*03:01, HLA-A*11:01 and HLA-A*24:02 in both cases. In the HLAthena evaluation of antigen binding and presentation to the corresponding HLA haplotypes, peptides were assigned to alleles by rank with a threshold of 0.1. Contexts of up to 30 flanking amino acids on both amino and carboxy termini were utilized with aggregation by peptide and no log-transformed expression. Baseline MHCflurry 2.0 models with both peptide–HLA binding affinity predictor and antigen-processing predictor were used. Overall, peptide–HLA presentation scores were characterized by mhcflurry_presentation_score and MSiC_HLA scores in MHCflurry 2.0 and HLAthena, respectively. To select for high-binders, we curated lists of peptide–HLA complexes in the top 10 percentile of scores from both prediction algorithms.

## Cell culture

### Culture of cells derived from xenografts from patients with GBM.
GBM, GBM34, GBM43, GBM108, GBM115, GBM118, GBM102, GBM137, GBM148, GBM164 and GBM195, were obtained from the Mayo Clinic Brain Tumor PDX national resource. Xenograft lines were cultured according to recommended conditions in previous literature[55] and passaged a maximum of 20 times before restoration to earlier passages. Cells were cultured in Dulbecco's modified Eagle's medium (DMEM) supplemented with 10% fetal bovine serum and 1% penicillin and streptomycin. Cell culture plates were treated overnight at 4 °C with DPBS (with calcium and magnesium) and 10% laminin (Gibco catalogue number 23017015) before use.

### Primary patient-derived GBM and LGG cell culture.
Primary patient-derived *IDH*wt GBM (SF7996), *IDH*mut-A (SF10602) and *IDH*mut-O (SF10417) cell lines were previously internally generated from dissociated glioma biopsies and cultured as previously described[41]. Cells were cultured in serum-free, glioma neural stem cell medium, which comprises Neurocult NS-A (STEMCELL Technologies catalogue no. 05751) supplemented with N-2 supplement (Invitrogen catalogue no. 17502048), B-27 supplement minus vitamin A (Invitrogen catalogue no. 12587010), 1% penicillin and streptomycin, 1% glutamine and 1% sodium pyruvate. Before immediate use in culture, glioma neural stem medium was supplemented with 20 ng ml⁻¹ EGF (Peprotech catalogue no. AF-100-15), bFGF (Peprotech catalogue no. AF-100-18B) and PDGF-AA (Peprotech catalogue no. AF-100-13A). As for cell lines derived from xenografts from patients with GBM, cell culture plates were incubated overnight at 4 °C with DPBS (with calcium and magnesium) and 10% laminin (Gibco catalogue no. 23017015) before use.

### Jurkat76 cell culture.
Jurkat76 cells were used as the TCR α- and β-negative human T cell derivative that allowed for non-competing introduction of exogenous TCRs. CD8⁺ Jurkat76 cells were cultured in RPMI supplemented with 10% fetal bovine serum and 1% penicillin and streptomycin.

### T2 cell culture.
T2 cells were used in the study to monitor immune cell response to the exogenous antigen of interest in a non-competitive environment. T2 cells are deficient in a peptide transporter involved in antigen processing (TAP), and as such, induction of these cells with exogenously administered peptides allows for their association and presentation by HLA molecules, HLA-A*02:01 in particular. We cultured T2 cells in IMDM medium supplemented with 20% FBS.

### COS-7 cell culture.
We opted to use COS-7 (ATCC catalogue no. CRL-1651) cell lines as our respective primate and human artificial APC models[52]. These cell lines do not express HLA molecules, which allows for the introduction of the HLA allele of interest. COS-7 cells were cultured in DMEM medium supplemented with 10% FBS and 1% penicillin and streptomycin.

### THP-1 cell culture.
THP-1 cells (ATCC catalogue no. TIB-202) were used to investigate immune reactivity against neoantigen presentation by dendritic cells (DCs). THP-1 cells were cultured in RPMI-1640 supplemented with 10% FBS. All cell lines have been tested for forms of mycoplasma contamination. Cell lines were obtained from trusted sources and have not been authenticated.

## siRNA-mediated knockdowns of splicing-related genes

Cells were seeded in 2 ml of antibiotic-free medium in a 6-well plate at the following densities: GBM115, 45,000 cells per well; SF10417, 100,000 cells per well; and SF10602, 100,000 cells per well. At 24 h post-seeding, cells were transfected by adding 400 µl reaction containing serum-free medium, 2.0 µl DharmaFECT 1 reagent (Horizon, no. T-2001-02), and their respective siRNA pools (four-siRNA equimolar mix) at a final concentration of 30 nM. At 24 h post-transfection, the medium was changed to complete medium. At 72 h post-transfection, RNAs were isolated and purified using the Zymo Quick-RNA microprep kit (Zymo Research, no. R1058).

## CRISPRi

sgRNAs were designed using the Broad CRISPick webportal[56,57]. Top-ranked sgRNAs were ordered from IDT: top strands were appended with 'CACCG' on their 5′ end, bottom strands were appended with 'AAAC' on their 5′ end and 'C' on their 3′ end. The oligonucleotide names and

sequences ordered from IDT are: sgSF3A3_CRISPRi_1_TopStrand, 5′-CACCGGAATTGAGAAGCCGCGACTA-3′; sgSF3A3_CRISPRi_1_BottomStrand, 5′-AAACTAGTCGCGGCTTCTCAATTCC-3′; sgSF3A3_CRISPRi_2_TopStrand, 5′-CACCGAAGCCGCGACTAAGGGAAGA-3′; sgSF3A3_CRISPRi_2_BottomStrand, 5′-AAACTCTTCCCTTAGTCGCG GCTTC-3′; sgSF3A3_CRISPRi_3_TopStrand, 5′-CACCGAGGGAAGAT GGAGACAATAC-3′; sgSF3A3_CRISPRi_3_BottomStrand, 5′-AAACGTATT GTCTCCATCTTCCCTC-3′; sgSF3A3_CRISPRi_4_TopStrand, 5′-CAC CGATTCAGACCACCAACACGGC-3′; sgSF3A3_CRISPRi_4_Bottom-Strand, 5′-AAACGCCGTGTTGGTGGTCTGAATC-3′; sgCELF2_CRISPRi_1_TopStrand, 5′-CACCGTCCCCTCCGAAATCCAGCGC-3′; sgCELF2_CRIS-PRi_1_BottomStrand, 5′-AAACGCGCTGGATTTCGGAGGGGAC-3′; sgCELF2_CRISPRi_2_TopStrand, 5′-CACCGGCCCCGGCGCTGGATT TCGG-3′; sgCELF2_CRISPRi_2_BottomStrand, 5′-AAACCCGAAATCC AGCGCCGGGGCC-3′; sgSNRPD2_CRISPRi_1_TopStrand, 5′-CACCGA GCGTAGTGACCATCATGTG-3′; sgSNRPD2_CRISPRi_1_BottomStrand, 5′-AAACCACATGATGGTCACTACGCTC-3′; sgSNRPD2_CRISPRi_2_TopStrand, 5′-CACCGCCTAGCCCGGCCTCACATGA-3′; sgSNRPD2_CRISPRi_2_BottomStrand, 5′-AAACTCATGTGAGGCCGGGCTA GGC-3′; sgROSA26_CRISPRi_TopStrand, 5′-CACCGACAGCAAGTT GTCTAACCCG-3′; sgROSA26_CRISPRi_BottomStrand, 5′-AAA CCGGGTTAGACAACTTGCTGTC-3′; sgAAVS1_CRISPRi_TopStrand, 5′-CACCGGGGCCACTAGGGACAGGAT-3′; sgAAVS1_CRISPRi_BottomStrand, 5′-AAACATCCTGTCCCTAGTGGCCCC-3′.

sgROSA26 (ref. 58) and sgAAVS1 (ref. 59) were from previous literature. Top and bottom strands of each sgRNA were then annealed and ligated into the CRISPRi vector pLV hU6-sgRNA hUbC-dCas9-KRAB-T2a-Puro (Addgene plasmid no. 71236)[60]. Lentivirus was produced as described in the section of the Methods entitled Lentiviral transduction. For transduction, SF10417 was plated at 20,000 cells per well in 24-well plates, and GBM115 cells were plated at 60,000 cells per well in 6-well plates. At 24 h post-seeding, cells were transduced by addition of virus in complete medium supplemented with 4 µg ml⁻¹ Polybrene. At 24 h post-transduction, medium was replaced with complete medium with 1 µg ml⁻¹ puromycin, and cells were selected for 72 h and then allowed to recover in complete medium. Each sgRNA was assessed by three separate transductions.

## Quantitative PCR with reverse transcription

A 1,000 ng quantity of DNAse-treated RNA was converted to cDNA using the iScript cDNA synthesis kit (BioRad, no. 1708891). This cDNA was then diluted 1:3 using ultrapure, nuclease-free water, and 2 µl was used per quantitative PCR (qPCR) reaction. qPCR with reverse transcription was performed using the Applied Biosystems POWER SYBR Green Master Mix (Applied Biosystems, no. 4367659). All samples were run in biological triplicates, with technical triplicates for each biological triplicate using the Quantstudio 5 (Thermo Scientific), and all gene expression data were normalized to the housekeeping gene *GUSB*. The cycling protocol was as follows: 2 min at 50 °C, 10 min at 95 °C, followed by 40 cycles at 95 °C for 15 s, and 60 °C for 60 s. Dissociation curves were plotted to confirm specific product amplification. Primer sequences corresponding to each gene for the mRNA expression analysis were designed using NCBI Primer.

## Amplicon sequencing for validation of NJ expression

RNAs from respective cell lines were isolated and purified using the Zymo Quick-RNA microprep kit (Zymo Research, no. R1058). A 1,000 ng quantity of DNAse-treated RNA was converted to cDNA using the iScript cDNA synthesis kit (BioRad, no. 1708891). This cDNA was then diluted 1:3 using ultrapure, nuclease-free water, and 2 µl was used per PCR reaction. Sixteen reactions were carried out per amplicon per cell line using Q5 High-Fidelity 2× master mix (NEB, no. M0492L) with primers containing partial Illumina adaptors. Reaction mixtures were set up according to the manufacturer's guidelines. These products were then purified by separation on a 1.0% agarose gel at 100 V (constant)

for 1 h and were then purified using the Monarch DNA gel extraction kit (NEB, no. T1020L). Purified products were quantified with a qubit high-sensitivity dsDNA kit (Invitrogen, no. Q32851) and prepared and submitted according to Azenta (Genewiz) guidelines for amplicon sequencing.

## IVS of healthy-donor PBMCs

HLA-A*02:01:01⁺ PBMCs were purchased from StemExpress in either fresh or cryopreserved format. Approximately $1 \times 10^9$ fresh PBMCs (StemExpress catalogue no. LE001F) were immediately proportioned into aliquots of $3 \times 10^8$ cells and cryopreserved in liquid nitrogen, with one aliquot actively used for downstream IVS. Cryopreserved PBMCs (StemExpress catalogue no. PBMNC300C) totalling approximately $3 \times 10^8$ cells per cryovial were used in one vial per IVS procedure. PBMCs were thawed with 1:1,000 Benzonase/RPMI (Sigma Aldrich catalogue no. E8263). The CD14⁺ population was isolated from the PBMCs using CD14⁺ Miltenyi microbeads (Miltenyi Biotec catalogue no. 130-050-201) as per the manufacturer's instructions. The CD14⁻ flowthrough was cryopreserved for 6 days before naive CD8⁺ T cell isolation. Isolated CD14⁺ cells were cultured in CellGenix GMP DC medium (CellGenix catalogue no. 20801-0500) supplemented with 1% human serum (Sigma Aldrich catalogue no. H6914), 1% penicillin and streptomycin, 1,000 U ml⁻¹ recombinant human IL-4 (Peprotech catalogue no. 200-04) and GM-CSF (Peprotech catalogue no. 300-03) in non-treated 24-well plates at a seeding density of $5 \times 10^5$ cells per well. On day 3, recombinant human IL-4 and GM-CSF (1,000 U ml⁻¹ each) were added to the DC culture. On day 5, the DC culture was matured with 250 ng ml⁻¹ LPS (Sigma Aldrich catalogue no. L6529) in addition to supplementation of recombinant human IL-4 and GM-CSF (1,000 U ml⁻¹ each). Naive CD8⁺ T cells were isolated from the thawed CD14⁻ population on day 6 using the EasySep Human Naive CD8⁺ T Cell Isolation Kit (STEMCELL Technologies catalogue no. 19258) as per the manufacturer's instructions. Isolated naive CD8⁺ T cells were cultured in X-Vivo 15 medium (Lonza catalogue no. 04-418Q) supplemented with 5% human serum, 1% penicillin and streptomycin and 10 ng ml⁻¹ of recombinant human IL-7 (Peprotech catalogue no. 200-07) in 48-well plates at a seeding density of $5 \times 10^5$ cells per well. On day 8, adherent matured DCs were collected from the plate using cold PBS. The collected DCs ($1 \times 10^6$ cells ml⁻¹) were exogenously pulsed with 1 µM of the neoantigen peptide, influenza peptide or no peptide for 1 h at 37 °C. The peptide-pulsed or non-pulsed DCs were then co-cultured with naive CD8⁺ T cells at an optimal DC/T cell ratio of 1:4 in 48-well plates. The co-culture was maintained with X-Vivo 15 medium supplemented with 10 ng ml⁻¹ of recombinant human IL-7, 10 ng ml⁻¹ recombinant human IL-15 (Peprotech catalogue no. 200-15) and 60 ng ml⁻¹ of recombinant human IL-21 (Peprotech catalogue no. 200-21) for 10 days with IL-7 and IL-15 restimulation every 2 days. Cells were reseeded into subsequent 24-well, 12-well and 6-well plates on the basis of confluency. This concluded the first cycle of IVS of the neoantigens and influenza peptides. On days 19 and 29, sensitized CD8⁺ T cells were reintroduced to a second and third round of stimulation with newly pulsed DCs, and the co-culture was maintained for 10 additional days until the end of the second and third cycle of IVS. Cytokine assays were performed at the end of the second and third cycles of IVS to determine whether a peptide-reactive T cell population has expanded.

## Mutation-specific ELISA screen

Aliquots containing CD8⁺ T cells from individual parent IVS wells were collected and split equally into 96-well plate daughter wells containing $1 \times 10^5$ cells per well. Daughter wells in triplicate were stimulated with T2 cells pulsed with the neoantigen peptide of interest, control peptide, no peptide or no T2 cells at all for 16 h at an effector-to-target (E/T) ratio of 1:1. T2 cells were pulsed with 1 pM to 1 µM of the neoantigen peptide of interest, control peptide or no peptides for 1 h at 37 °C. Influenza-reactive T cells were co-cultured against influenza

peptide-pulsed T2 cells as a positive control. Co-culture supernatant was collected and diluted for use in IFNγ (BD Biosciences catalogue no. 555142) and TNF (BD Biosciences catalogue no. 555212) ELISAs as per the manufacturer's instructions. ELISA readouts were performed on the Epoch Microplate Spectrophotometer (BioTek Instruments) using the BioTek Gen5 Data Analysis software (version 1.11). Wells with significantly increased expression levels of IFNγ and TNF were selected for downstream single-cell immune profiling using single-cell RNA and V(D)J sequencing.

## Single-cell immune profiling

Once an expanded neoantigen-reactive CD8+ T cell population from IVS was identified, single-cell RNA and V(D)J sequencing were performed using the 10x Genomics platform. Before sequencing, CD8+ T cells from the expanded neoantigen-reactive (ELISA screen-positive) wells were collected and co-cultured with T2 cells pulsed with 1 μM of the neoantigen peptide of interest, a control peptide or no peptides at an E/T ratio of 1:1. One co-culture replicate was performed for 3 h for single-cell RNA-seq analysis, and another was performed for 16 h for IFNγ and TNF ELISA confirmation. The final cell concentration was adjusted to approximately $1 \times 10^4$ cells per microlitre with an initial cell viability of at least 90% to maximize the likelihood of achieving the desired cell recovery target. Independent CD8+ T cell and non-pulsed T2 single cultures were sequenced alongside the co-culture conditions for differentiating cell types in the downstream single-cell sequencing analysis. The Chromium Next GEM Single Cell 5′ Reagent Kit v2 (Dual Index) (10x Genomics, catalogue no. CG000331) was used for preparation for single-cell sequencing analysis. Gel beads in emulsions (GEMs) were generated by combining the single-cell 5′ gel beads, partitioning oil and the master mix containing the cells onto the Chromium Next GEM Chip K. Cell lysis and barcoded reverse transcription of RNAs in all single cells were finished inside their corresponding GEM. Barcoded cDNA product was recovered through post-GEM-RT cleanup and PCR amplification. cDNA quality control and quantification were performed on the Fragment Analyzer System (Agilent Technologies). A 50 ng quantity of cDNA was used for the construction of the 5′ gene expression library, and each sample was indexed by a Chromium i7 Sample Index Kit. This process was performed on an Illumina NovaSeq 6000 sequencer at the UCSF Institute of Human Genetics (IHG) with a minimum of 20,000 read pairs per cell for the 5′ Gene Expression library. The enriched product was measured by the Fragment Analyzer System. A 50 ng quantity of enrichment TCR product was used for library construction. Single-cell V(D)J-enriched libraries were subsequently sequenced on the Illumina NovaSeq 6000 with a minimum of 5,000 read pairs per cell for the V(D)J library. Cell Ranger 7.0.0 (10x Genomics Cloud Analysis) was used to pre-process raw single-cell RNA-seq and identify V(D)J clonotypes. The annotation files vdj_GRCh38_alts_ensembl-3.1.0-3.1.0 and GRCh38-3.0.0 were used for demultiplexing cellular barcodes, performing read alignments and generating feature–barcode matrices. Only cells for which clonotype information was available were retained for downstream analysis. Single-cell gene expression and corresponding V(D)J sequences of candidate T cell clonotypes were analysed on the Loupe V(D)J browser. Single cells with detectable *CD8A* expression were specifically isolated and characterized as the CD8+ T cell population and subsequently grouped according to their TCR clonotypes. To identify T cell clonotypes associated with a neoantigen-specific response, we selected expanded TCR clonotypes with significantly increased levels of *IFNG*, *TNF* and *GZMB* expression in the T cell:neoantigen-pulsed T2 condition compared to the T cell:control-pulsed T2 and T cell:non-pulsed T2 conditions.

## HLA typing

OptiType 1.3.1 was used for genotyping HLA alleles from available whole-exome sequencing data available for glioma cell lines with default parameters.

## Plasmids and peptides

HLA-A*02:01 and NEJ-derived gene sequences were all synthesized and cloned into the pTwist Lenti SFFV Puro WPRE vector (Twist Biosciences). Constructs encoding full-length and truncated multi-base-polypeptide versions of the wild-type and mutant *GNAS* and *RPL22* sequences were generated. TCR α and β was synthesized and cloned into the pTwist Lenti SFFV vector (Twist Biosciences). HPLC-grade NEJ-derived neoantigen peptides (>95%) were manufactured by TC Laboratories.

## Lentiviral transduction

HEK293T cells were plated in 6-well culture plates at a density of $1 \times 10^6$ cells per well with 2 ml DMEM supplemented with 10% FBS without antibiotics. After approximately 18 to 24 h or at 90% confluency, HEK293T cells were transfected with the expression construct, see above, and lentiviral packaging plasmids, pMD2.G (Addgene, no. 12259) and psPAX2 (Addgene, catalogue no. 12260).

**TCR α/β transduction.** A 1.0 μg quantity of TCR α/β transfer plasmid, 0.75 μg psPAX2 and 0.25 μg pMD2.G were combined with 200 μl Opti-MEM (Thermo Fischer Scientific catalogue no. 31985062). A 6 μl volume of Xtremegene HP was added to this mixture, and complex formation was allowed to occur for 15 min at room temperature, at which point this reaction mixture was added to the corresponding HEK293T cells. Transfection medium was replaced with fresh DMEM after 24 h. Viral supernatant was collected after 48 h, and the functional virus titre was measured on 6-well plates seeded with Jurkat76/CD8 cells or PBMC-derived CD8+ T cells at 60–70% confluency. Viral transduction was performed with threefold serial dilutions of the virus stock supplemented with Polybrene at a final concentration of 4 μg ml⁻¹. Medium was changed 24 h following viral transduction. Cells were assessed for transduction efficiency after 3–4 days by measuring surface expression of TCR α/β and CD3 by fluorescence-activated cell sorting (FACS) analysis. Cells demonstrating a high level of double-positive expression of TCR α/β and CD3 were flow-sorted and maintained for downstream co-culture and reactivity assays.

**HLA and neoantigen transduction.** Constructs expressing HLA-A*02:01 were linearized and restricted with BamHI and XhoI (New England Biolabs) and purified using the Zymoclean Gel DNA Recovery Kit (Zymo Research catalogue no. D4007). The HLA-A*0201 sequence was then ligated into a lentiviral construct downstream of an EF1A-core promoter and upstream of an IRES followed by a blasticidin resistance gene. A 1.0 μg quantity of either HLA-A*02:01 or neoantigen transfer plasmid, 0.75 μg psPAX2 and 0.25 μg pMD2.G were combined with 200 μl Opti-MEM (Thermo Fischer Scientific catalogue no. 31985062). A 6 μl volume of Xtremegene HP was added to this mixture, and complex formation was allowed to occur for 15 min at room temperature, at which point this reaction mixture was added to corresponding HEK293T cells. As stated above, neoantigen constructs encode either the full-length or truncated version of the NJ-derived peptide. The transfection medium was replaced with fresh DMEM medium after 24 h. HLA-A*02:01 lentiviral transduction and screening were performed first before neoantigen lentiviral transduction and screening for streamlined drug selection. Viral supernatant was collected after a subsequent 48 h, and the functional virus titre was measured on 6-well plates seeded with COS-7 cells at 60–70% confluency. Viral transduction was performed with threefold serial dilutions of the virus stock supplemented with 4 μg ml⁻¹ Polybrene. Medium was changed 24 h following viral transduction and replaced with complete medium supplemented with blasticidin. Cells were assessed for transduction efficiency after 3–4 days by drug screening. HLA-A*02:01-transduced APCs were cultured in medium treated with 10 μg ml⁻¹ blasticidin for approximately 7 days before assessing for cell viability across titres. Neoantigen-lentiviral transduction was subsequently performed, and APCs transduced with

both HLA-A*02:01 and neoantigen-expressing constructs were then cultured in medium treated with 3 μg ml⁻¹ puromycin for approximately 7 days. Cell viability was assessed afterwards across all titre conditions. Cells were assessed for transduction efficiency after 3–4 days by measuring surface expression of HLA-A2 FACS analysis.

## Dose-dependent assessment of TCR reactivity against neoantigen

Specificity of neoantigen-reactive CD8⁺ T cells and TCR-transduced T cells was assessed by human IFNγ (BD Biosciences catalogue no. 555142), IL-2 (BD Biosciences catalogue no. 555190) and TNF (BD Biosciences catalogue no. 555212) ELISA. Assessment of TCR recognition against exogenously introduced neoantigen peptides presented by HLA molecules was conducted by co-culturing T cells with peptide-pulsed T2 cell conditions. T2 cells were pulsed with neoantigen peptide of interest at a concentration between 1 pM and 1 μM, decoy peptide or no peptides for 1 h at 37 °C. Influenza-reactive T cells were co-cultured against influenza peptide-pulsed T2 cells as a positive control. T cells and T2 cells were co-cultured in a 96-well round-bottom plate at a concentration of 1 × 10⁵ of each cell type in 200 μl of medium for 16 h. Supernatant was collected and diluted for cytokine release assays per the manufacturer's instructions. ELISA assay readouts were performed on an Epoch Microplate Spectrophotometer (input wavelength 450 nm and output wavelength 570 nm) using the BioTek Gen5 Data Analysis software. To characterize the dose-dependent activation of the TCRs in transduced triple-reporter Jurkat76/CD8 cells, we performed flow analysis to assess the level of expression of NFAT–GFP, NF-κB–CFP and AP-1–mCherry following 16 h of co-culture. Similarly, the reactivity of TCR-transduced PBMC-derived CD8⁺ T cells was evaluated by flow analysis following anti-CD107a (BioLegend, catalogue no. 328620) and anti-CD137 antibody (4-1BB; BioLegend catalogue no. 309804) staining.

## In vitro transcription synthesis of mRNA

All constructs were subcloned into pcDNA3.1 (Invitrogen, 2520855) and linearized by XhoI restriction enzyme with the plasmid DNA template transcribed downstream from the bacteriophage T7 promoter sequence. For long (>0.5 kilobase (kb)) and short (<0.5 kb) transcripts, 1 μg and 0.5 μg of template were used, respectively. Reactions were assembled at room temperature using the mMESSAGE mMACHINE T7 Transcription Kit as per the manufacturer's instructions (Invitrogen, 2582905) and incubated at 37 °C for 1 h for long transcripts and 16 h for short transcripts. Following DNase treatment, a poly(A) tailing reaction was performed for 1 h according to the HiScribe T7 ARCA manual (NEB, E2060S). Subsequently, the synthesized mRNA was purified by LiCl precipitation using 70% DEPC-based ethanol. Synthesized mRNA was heat-shocked (70 °C, 5 min) with the formaldehyde loading dye to verify quality through gel electrophoresis.

## mRNA transfection of HLA-A*02:01, truncated neoantigen and full-length NEJ-encoding mRNA

Transfection of in vitro transcription-synthesized mRNA into COS-7 cells was performed with electroporation using the Neon Transfection System 100 μl Kit (Invitrogen, MPK10096) as per the manufacturer's instructions. A total of 1 × 10⁶ COS-7 cells were washed and resuspended with 100 μl of Neon Resuspension Buffer. A 5 μg quantity of HLA-A2 and 5 μg of candidate (either the truncated neoantigen sequence or the full-length NEJ sequence) mRNA were added into the cell solution. Electroporation was performed on the Neon NxT Electroporation System (Invitrogen, NEON1). Electroporation of COS-7 cells was performed with the following optimized conditions: pulse voltage of 1,200 V, width of 30 ms and 2 pulses. Transfected cells were immediately transferred into warm RPMI with no antibiotics. Aliquots of transfected cells were retained for validation of HLA-A2 expression by staining with HLA-A2 monoclonal antibody (BB7.2, Thermo Scientific, 17-9876-42) and subsequent flow cytometry analysis.

## Evaluation of TCR specificity against endogenously processed and HLA-presented neoantigen

Characterization of neoantigens that are endogenously processed and presented by surface HLA was conducted by co-culturing HLA-A*02:01/neoantigen-transfected COS-7 cells with TCR-transduced T cells. Similarly, T cells and COS-7 cells were co-cultured in a 96-well flat-bottom plate at a concentration of 1 × 10⁵ of each cell type in 200 μl of medium for 16 h. Supernatant was collected and diluted for cytokine release assays as per the manufacturer's instructions, and cytokine release levels were assessed with the Epoch Microplate Spectrophotometer and BioTek Gen5 Data Analysis software. In all cytokine release assay experiments, maximum cellular cytokine release per well was determined by the addition of 0.2 μl Cell Activation Cocktail (without brefeldin A) (BioLegend catalogue no. 423302) per 100 μl cell solution. Evaluation of endogenously processed and presented neoantigens in glioma cell lines was performed by co-culturing TCR-transduced triple-reporter Jurkat76 cells with glioma cells at a 1:1 E/T ratio (1 × 10⁵ per well in a 96-well plate). Flow analysis was performed to assess the level of expression of NFAT–GFP, NF-κB–CFP and AP-1–mCherry following 16 h of co-culture.

## HLA immunoprecipitation and liquid chromatography with tandem MS

COS-7 cells were co-electroporated with 10 μg of each mRNA encoding the HLA-A*02:01 allele and the full-length coding sequence of the mutated GNAS or RPL22 using the Neon Transfection system (100-μl tip, setting: 1,050 V, 10 ms and 2 pulses). A total of 20 × 10⁶ cells were electroporated per condition and plated in 6-well non-TC plates overnight. For the GMB115 cell line sample, approximately 100 × 10⁶ cells were used. Cells were collected by incubating with 1 mM EDTA (Millipore Sigma) for 10 min at 37 °C. For the immunoprecipitation experiments, cells were lysed in 8 ml of 1% CHAPS (Millipore Sigma) for 1 h at 4 °C; the lysates were then spun down for 1 h at 20,000g and 4 °C, and supernatant was collected. For the affinity-column-based immunopurification of HLA-I ligands, 40 mg of cyanogen bromide-activated Sepharose 4B (MilliporeSigma) was activated with 1 mM hydrochloric acid (MilliporeSigma) for 30 min. Subsequently, 1 mg of W6/32 antibody (Bio X Cell) was coupled to Sepharose in the presence of binding buffer (150 mM sodium chloride, 50 mM sodium bicarbonate, pH 8.3; sodium chloride) for 2 h at room temperature. Sepharose was blocked for 1 h with glycine and washed three times with PBS. Supernatants of cell lysates were run through an affinity column using peristaltic pumps at 6 ml min⁻¹ flow rate overnight at 4 °C. HLA complexes and binding peptides were eluted from the column five times using 1% TFA. Peptides and HLA-I complexes were separated using C18 columns (Sep-Pak C18 1 cc Vac Cartridge, 50 mg of sorbent per cartridge, 37–55-μm particle size, Waters). C18 columns were pre-conditioned with 80% ACN (Millipore Sigma) in 0.1% TFA and equilibrated with two washes of 0.1% TFA. Samples were loaded, washed twice with 0.1% TFA and eluted in 300 μl of 30%, 40% and 50% acetonitrile in 0.1% TFA. All three fractions were pooled, dried down using vacuum centrifugation and stored at −80 °C until further processing. HLA-I ligands were isolated by solid-phase extractions using in-house C18 mini-columns. Samples were analysed by high-resolution, high-accuracy liquid chromatography with tandem MS (Lumos Fusion, Thermo Fisher Scientific). COS-7 samples were run in DDA mode, and GMB115 samples were run in DIA mode. MS and tandem MS were operated at resolutions of 60,000 and 30,000, respectively. Only charge states 1, 2 and 3 were allowed. The isolation window was chosen as 1.6 Th, and collision energy was set at 30%. For tandem MS, maximum injection time was 100 ms with an automatic gain control of 50,000. MS data were processed using FragPipe. Protein false discovery rate was set at 1%. Oxidization of methionine, phosphorylation of serine, threonine and tyrosine, and N-terminal acetylation were set as variable modifications for all samples. Samples were searched against a database comprising

UniProt *Cercopithecus aethiops* or UniProt human-reviewed proteins supplemented with the human HLA-A*02:01 allele sequence, mutRPL22 and mutGNAS, as well as common contaminants.

## Characterization of CD8+ T cell-mediated anti-tumour reactivity

To determine whether TCR-transduced T cells were capable of mounting an anti-tumour response, TCR-transduced Jurkat76/CD8 or PBMC-derived CD8+ T cells were co-cultured with patient-derived GBM or LGG cell lines. CD8+ T cells were isolated from healthy-donor-derived PBMCs using the EasySep Human CD8+ T Cell Isolation Kit (STEMCELL Technologies, catalogue no. 17953). CD8+ T cells were then activated with Dynabeads Human T-Activator CD3/CD28 for T Cell Expansion and Activation (Thermo Scientific, catalogue no. 11161D) at a concentration of 25 µl per $1 \times 10^6$ cells. CD8+ T cells were cultured for 7 days with IL-7 (30 µl per $1 \times 10^6$ cells) supplemented every 2 days. CD8+ T cells were then lentivirus-transduced with neoantigen-specific TCRs with a hybridized mouse TCR constant region using the above transduction procedure. This additional step removes the likelihood of TCR α-chain and β-chain mispairing and allows us to evaluate TCR-transduction efficiency by staining with anti-mouse TCR constant region antibody (clone H57-597; BioLegend catalogue no. 109208). Flow sorting was performed to isolate highly transduced CD8+ T cells by selecting for cells stained strongly with anti-CD3 and anti-mouse TCR constant region antibody. Sorted transduced CD8+ T cells were expanded for 7 days before use in co-culture assays. Killing assays were performed using an xCELLigence RTCA S16 Real-Time Cell Analyzer. Tumour cells were cultured in medium pre-treated with 100 ng ml⁻¹ IFNγ (Peprotech, catalogue no. 300-02) for 48 h and washed twice with PBS before seeding. A total of $1 \times 10^4$ tumour cells were plated per well in a 96-well E-plate (Agilent), and impedance was read for 16 h during incubation. TCR-transduced CD8+ T cells were introduced to each well at an E/T ratio of either 1:1 or 2:1, and tumour-specific killing was measured by changes in cell index over 24–48 h.

## Identification of HLA-restricted CD8+ T cell-mediated reactivity against neoantigens

Evaluation of HLA-restricted T cell reactivity was performed by perturbing TCR and HLA–peptide interactions with the introduction of anti-HLA antibodies. In dose-dependent reactivity assays, T2 cells at a concentration of $1 \times 10^5$ tumour cells per well of a 96-well plate were washed twice with PBS and incubated for 30 min with blocking anti-HLA antibody (50 µg per well; clone W6/32, Bio X Cell, catalogue no. BE0079) or isotype control (50 µg per well; Bio X Cell, catalogue no. BE0085) at a total volume of 100 µl. Without any additional washes, T cells were added to achieve a final volume of 200 µl. In tumour-killing assays, tumour cells were added to each well of a 96-well E-plate in a total volume of 50 µl for initial seeding. Anti-HLA antibody or isotype control (50 µg per well) was added to each well 30 min before the addition of T cells to reach a total volume of 100 µl. T cells were added to each well to achieve a final volume of 200 µl, and impedance was measured for the following 24–48 h.

## Immune monitoring of patients with cancer expressing mut*GNAS*-NEJ

PBMCs obtained from HLA-A*02:01+ patients with cancers expressing *GNAS* NEJs were tested for the presence of mutGNAS-specific CD8+ T cells by FACS using dual-colour HLA-A*02:01 dextramers loaded with the MS-identified mutGNAS peptide. Patient CD8+ T cells were stimulated in vitro with NEJ-expressing HLA-A*02:01-matched monocyte-derived DCs (moDCs) for 2 weeks before the FACS staining. To generate moDCs, HLA-A*02:01 healthy-donor PBMCs were plated in tissue culture flasks at $1 \times 10^6$ cells per square centimetre in complete medium without cytokines for 2 h at 37 °C to separate the adherent (monocyte-containing) and non-adherent (T cell-containing) fractions. The adherent fraction was washed with PBS, and fresh human

A/B serum-containing medium supplemented with recombinant human IL-4 and GM-CSF (400 IU ml⁻¹) was provided every 3 days. On day 6, moDCs were matured with LPS (Invitrogen) and IFNγ (Miltenyi Biotec) for 24 h before transfection. moDCs were electroporated with 100 µg ml⁻¹ of mRNA encoding full-length mutGNAS using the Neon Transfection system (10-µl tip, setting: 1,325 V, 10 ms and 3 pulses). A bulk population of patient CD8+ cells were enriched from PBMCs by negative selection (STEMCELL Technologies) and co-cultured with mut*GNAS*-NEJ-expressing HLA-A*02:01-matched moDCs at a 2:1 ratio in non-tissue-treated 24-well plates (FALCON) in the presence of 300 IU ml⁻¹ IL-2 and 50 ng ml⁻¹ of IL-7, IL-15 and IL-21. Cytokines were replenished every 3 days. As a control, similarly isolated and co-cultured HLA-A*02:01-matched CD8+ cells from a healthy donor were used. For dextramer labelling, HLA-A*02:01 multimers bound to mutGNAS and conjugated to PE or APC were purchased from Immudex. As a specificity control, HLA-A*02:01 multimers bound to the nine-amino acid polypeptide from P53(R175H) (HMTEVVRHC) were used. Cells were labelled with dual-fluorophore-conjugated dextramers for 15 min at room temperature, followed by surface antibodies against CD3–BV785, CD4–BV421 and CD8–BV650 (BioLegend) for an additional 15 min at 4 °C. Cells were washed twice, stained with the viability dye 7-AAD (BioLegend) and acquired on a BD Fortessa X20 flow cytometer.

## FACS analysis and antibodies

TCR-transduced cell lines were stained with anti-human TCR α/β (clone IP26, BioLegend catalogue no. 306717) and anti-human CD3 antibody (clone HIT3a, BioLegend catalogue no. 300307) to assess the surface-level expression of the transduced TCR. CD8+ T cells were stained with anti-CD107a (BioLegend, catalogue no. 328620) and anti-CD137 antibody (4-1BB; BioLegend catalogue no. 309804) to assess CD8+ T cell degranulation and TCR activation, respectively. The viability of cells was assessed with the Zombie Green Fixable Viability Kit (BioLegend, catalogue no. 423111) APCs and patient-derived glioma cell lines were stained with HLA-A2 monoclonal antibody (clone BB7.2, Thermo Fisher Scientific catalogue no. 17-9876-42). Approximately $1 \times 10^6$ cells per 100 µl FACS buffer (PBS supplemented with 1% BSA (Sigma Aldrich catalogue no. L6529)) were incubated with one test volume of antibody for 20 min as indicated by the manufacturer. Stained cells were washed once with FACS buffer before resuspension to a concentration of $4 \times 10^5$ cells per 100 µl FACS buffer. Cells were then analysed with the Attune NxT flow cytometer (Thermo Fischer Scientific). Unless otherwise stated, the concentration of antibody used was the one recommended by the manufacturer.

## Gene set enrichment analysis

Differential gene expression of TCGA, GTEx and UCSF GBM and LGG RNA-seq was performed and quantified using DESeq2 (ref. 61). Only genes with an absolute fold change of >1.5 and a Benjamini–Hochberg-adjusted *P* value < 0.05 called by DESeq2 were considered to be differentially expressed[62]. Pre-ranked gene set enrichment analysis (GSEA) was carried out by ranking genes with the product of their fold-change sign and the −log₁₀[adjusted *P* value].

**Disease subtype-specific differential gene analysis.** GSEA comparison was performed between *IDH* mutation subtypes (*IDH*wt and *IDH*mut) as well as glioma disease subtypes (*IDH*wt, *IDH*mut-A and *IDH*mut-O). Splicing-related gene sets were selected on the basis of keyword search, and gene sets with an adjusted *P* value of <0.05 when comparing two groups are considered differentially enriched. Unbiased hierarchical clustering of differentially enriched gene sets allows the characterization of subgroup-specific upregulated genes.

**NJ-load-specific differential gene analysis.** TCGA LGG and GBM samples were ranked according to the total putative NJs expressed per sample. High (NJ$_{HI}$) and low (NJ$_{LO}$) NJ load samples in each

disease subtype were characterized as the upper and lower 0.10 percentile of ranked samples, respectively. GSEA was carried out between the $NJ_{HI}$ and $NJ_{LO}$ samples of each disease subgroup. Gene sets with a unidirectional fold-change and adjusted $P$ value of <0.05 were considered to be enriched gene sets associated with NJ load. Splicing-related gene sets were selected on the basis of keyword searches. Leading-edge genes shared across all disease subgroups in the same gene set are defined as enriched genes associated with NJ load.

## NJ and splicing-related gene correlation analysis

Selection of *IDH*mut upregulated genes was determined by splicing-related genes expressed with a significant ($P < 0.05$) log$_2$[fold increase] of 1.5 in *IDH*mut cases when compared to their wild-type counterpart. Selection of splicing genes affected by oligodendroglioma-specific loss of chromosomes 1p and 19q was determined by chromosome 1p and 19q splicing-related genes expressed with a significant ($P < 0.05$) log$_2$[fold decrease] of 1.5 in *IDH*mut-O cases compared to both *IDH*mut-A and *IDH*wt cases. Splicing-related genes that were selected for in vitro validation were chosen on the basis of previously reported confirmation of aberrant splicing based on their dysregulated expression[40,42]. To determine correlation factors between each of the identified public NJs with each splicing gene of interest, we performed a Pearson correlation analysis against each NJ and splicing-related gene pair. NJs with the highest positive correlation score against the select *IDH*mut upregulated genes (*CELF2* and *ELAVL4*) averaged across all three glioma subtypes were tested in downstream qPCR assays. Similarly, NJs with the most negative correlation score against select chromosome 1p or 19q splicing-related genes downregulated in *IDH*mut-O cases (*SNRPD2* and *SF3A3*) averaged across all three glioma subtypes were also tested in downstream qPCR assays.

## AlphaFold2 structure predictions

AlphaFold v2.3.2 and its reference databases were installed. AlphaFold was run in multimer mode with default options and the highest rank resulting pdb file was visualized using Pymol. The image was exported with the settings ray 5000,5000 and png image,dpi=2400.

## Quantification and statistical analysis

All statistical analysis was performed in R statistical software (v.4.3.3) or GraphPad Prism (v.9.2.0). Data shown in column graphs represent mean ± standard error of the mean (s.e.m.) or mean ± standard deviation (s.d.), as indicated in the figure legends. Individual data points are plotted. Details of statistical testing can be found in the figure legends. Significance values: *$P < 0.05$; **$P < 0.01$; ***$P < 0.001$; ****$P < 0.0001$; NS, not significant. Statistical information for individual figures is provided in Supplementary Table 3.

## Materials availability

We have cloned TCR cDNAs that have anti-tumour properties. We have filed an invention disclosure (UCSF-743PRV) and will share these with academic investigators as per the material transfer agreement.

## Reporting summary

Further information on research design is available in the Nature Portfolio Reporting Summary linked to this article.

## Data availability

Spatially mapped glioma biopsy RNA-seq datasets are deposited in the European Genome-Phenome Archive (EGA) under the accession codes EGAS00001007986, EGAS00001006785, EGAD00001005221, EGAD00001005222, EGAD00001009496 and EGAD00001009497. Spatially mapped biopsy RNA-seq data for other tumour types were retrieved from their corresponding publications. Through the NIH Sequence Read Archive at https://www.ncbi.nlm.nih.gov/sra, RNA-seq data can be accessed with the accession code PRJNA579899 for ref. 22 and with the accession code SRP066596 for ref. 23. Through the National Omics Data Encyclopedia, RNA-seq data can be accessed with the accession code OEP002956 for ref. 24. Through the EGA, RNA-seq data can be accessed with the accession codes EGAD00001009042 for ref. 25, EGAS00001003813 for ref. 26 and EGAS00001005328 for ref. 29. TRACERx data were requested and received from the Cancer Research UK & University College London Cancer Trials Centre. Glioma MS data were retrieved from the Clinical Proteomic Tumor Analysis Consortium as well as the Proteomics Identifications Database. The Proteomics Identifications Database accession code for ref. 45 is PXD024427. Proteomic data were retrieved from the supplementary files of ref. 46.

## Code availability

All original codes for the identification of tumour-wide public NJs have been deposited in GitHub at https://github.com/dakwok/SSNIP and are publicly available. Any additional information required to reanalyse the data reported in this paper is available from the corresponding authors upon request.

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

**Acknowledgements** This study was supported, in part, by the National Institutes of Health (NIH) grants R35NS105068 (H. Okada), R01CA222965 (H. Okada), NCI 2P50CA097257 (J.F.C.), NCI P50CA097257 (J.F.C.), NCI P01CA118816 (J.F.C.), T32GM008568 (D.W.K.), 5T32CA151022 (L.H.C.), R37 CA259177 (C.A.K.), R01CA269733 (C.A.K.), R01 CA286507 (C.A.K.), P30 CA008748 (C.A.K.), P50 CA217694 (C.A.K.), Cycle for Survival (C.A.K.), and NCI 1R50CA274229 (C.H.), the Gianna Rae Meadows Grant for the Oligodendroglioma Cure (H. Okada and J.F.C.), the Achievement Rewards for College Scientists Scholarship (D.W.K.), the AACR Scholar-in-Training Award (D.W.K.), the Glioblastoma Precision Medicine Project (J.F.C.), the Hana Jabsheh Research Initiative (J.F.C.), the Brain Tumor Funders' Collaborative (J.F.C. and H. Okada), the Dabbiere family (J.F.C. and H. Okada), the Metropoulos Family Foundation (C.A.K.), the Parker Institute for Cancer Immunotherapy (H. Okada, C.A.K. and I.E.) and the Cancer Research Institute (I.E.). The UCSF Glioblastoma Precision Medicine Program is sponsored by the Sandler Foundation (J.F.C.). Sequencing was performed at the UCSF CAT, supported by UCSF PBBR, RRP IMIA and NIH 1S10OD028511-01 grants.

**Author contributions** D.W.K. conceived the work and designed the experimental setup and data analysis with input from H. Okada, J.F.C. and C.A.K. D.W.K. and T.N. jointly designed and implemented the RNA-seq analysis pipeline and performed RNA-seq analyses, characterization of TCGA-derived NJs and quantitative alternative splicing analysis. Pan-cancer analyses were the result of discussion among D.W.K., N.O.S. and J.F.C. D.W.K. analysed publicly available MS data with guidance from A.B., E.R. and A.P.W. Spatially mapped samples were obtained by S.M.C. and J.J.P. Corresponding sequencing data were generated by C.H. and J.F.C. D.W.K. and K.O. performed IVS against NEJ-derived neoantigens with input from P.B.W., H. Ogino., A.S. and C.A.K. N.O.S. and C.H. provided help with 10× V(D)J scRNA-seq preparation, and L.H.C. assisted with scRNA-seq analysis and the identification of neoantigen-reactive TCR clones. M.C.C. generated lentiviral constructs, transduced CD8⁺ T cells with neoantigen-specific TCRs, and performed sorting for TCR-expressing cells. D.W.K. performed all ELISA assays with assistance from G.K.L.C., J.L., J.W., A.Du and N.R.G. J.J., mentored by A. Diaz, performed in vitro transcription to generate HLA- and NEJ-encoding mRNA with experimental design input from I.E. In vitro transcription, electroporation and subsequent COS-7/TR Jurkat76 co-cultures were performed by D.W.K. with input from I.E. Tumour killing assays using xCELLigence were conducted by D.W.K. with guidance from S.L. and A.Y. All flow cytometry data were obtained by D.W.K. with

experimental setup assistance from A.Y. and S.L. Flow cytometry analyses and gating were performed by M.G. and D.W.K. N.O.S. designed the primers for qPCR quantification, and S.H.W. performed the subsequent qPCR experiments to quantify NJ reads. N.O.S. performed siRNA knockdown experiments and CRISPRi knockdown experiments against splicing factors. D.W.K. wrote the manuscript, and all authors provided feedback on manuscript drafts.

**Competing interests** C.A.K. and I.E. are inventors on patents related to public neoantigen-specific TCRs unrelated to the present manuscript and are recipients of licensing revenue shared according to Memorial Sloan Kettering Cancer Center institutional policies. C.A.K. has consulted for or is on the scientific advisory boards for Achilles Therapeutics, Affini-T Therapeutics, Aleta BioTherapeutics, Bellicum Pharmaceuticals, Bristol Myers Squibb, Catamaran Bio, Cell Design Labs, Decheng Capital, G1 Therapeutics, Klus Pharma, Obsidian Therapeutics, PACT Pharma, Roche/Genentech, Royalty Pharma and T-knife. C.A.K. is a scientific co-founder and equity holder in Affini-T Therapeutics.

**Additional information**
**Correspondence and requests for materials** should be addressed to Christopher A. Klebanoff, Joseph F. Costello or Hideho Okada.

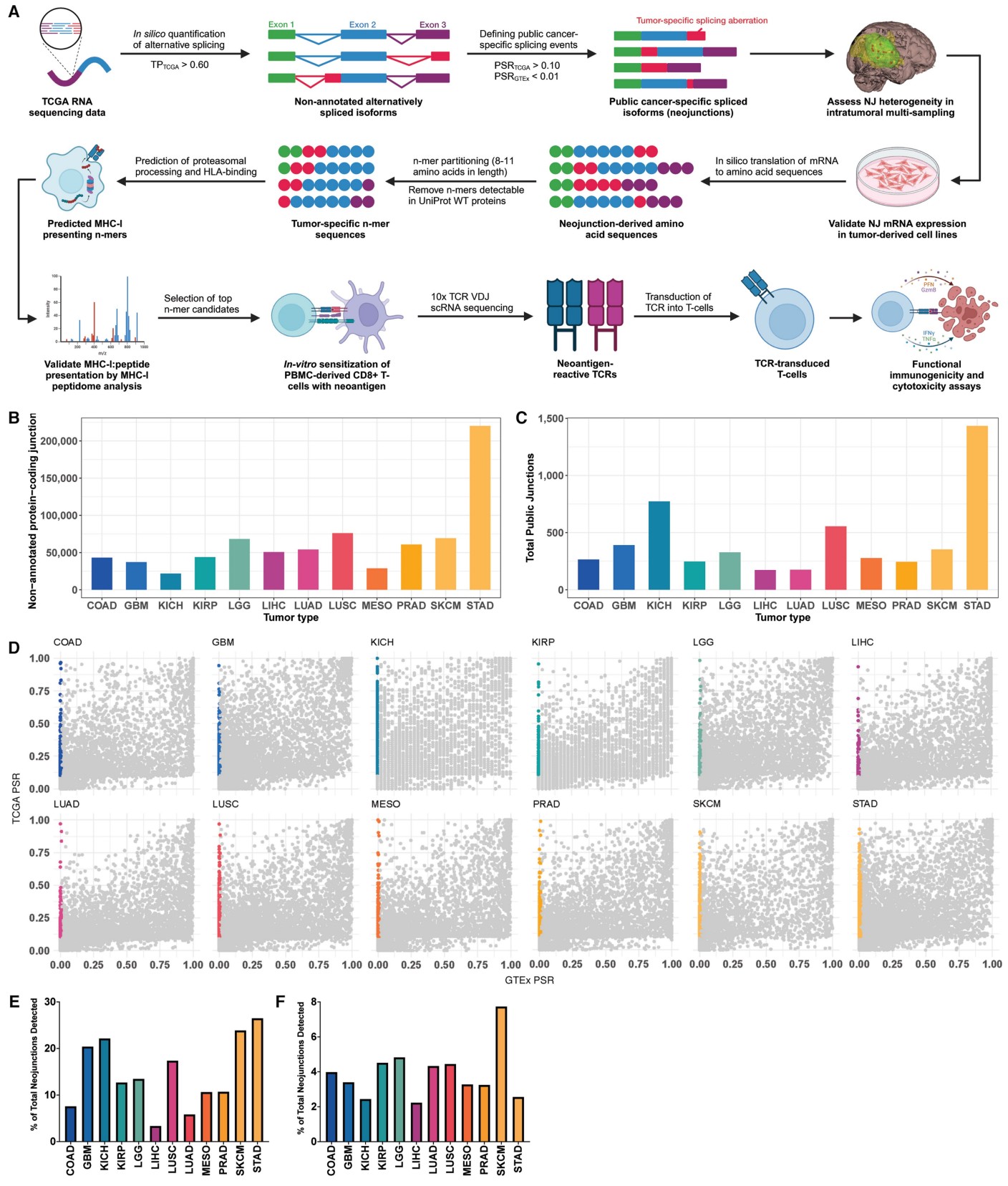

**Extended Data Fig. 1** | See next page for caption.

**Extended Data Fig. 1 | Pan-cancer public NJs are characterized from TCGA.**
**A**. TCGA RNA-seq data across multiple cancers ($n$ = 12) were analyzed for non-annotated, protein-coding, and cancer-specific splicing junctions (GTEx positive sample rate <1%; NJs). Interpatiently conserved (TCGA positive sample rate ≥10%; public NJs) were retained for downstream analysis of ITH. Tumors with sequencing data extracted from multiple intratumoral regions were used to evaluate each public NJ's ITH. Independent prediction algorithms were used to assess proteasomal processing and MHC-I binding of peptide sequences translated from public, intratumorally conserved NJs. The expression of these NJs and their peptide derivatives were validated by RNA-seq and MS analysis of patient-derived tumor samples and cell lines. T-cell receptors (TCRs) were cloned and characterized for top predicted candidates through in vitro sensitization of PBMC-derived CD8⁺ T-cells against the corresponding neoantigen-pulsed antigen-presenting cells and subsequent 10x V(D)J single-cell sequencing. Transduction of these neoantigen-reactive TCR sequences in TCR-null Jurkat76/CD8 cells and PBMC-derived CD8⁺ T-cells allowed the demonstration of neoantigen-specific reactivity and tumor-specific killing. **B**. Total number of non-annotated, protein-coding junctions detected pan-cancer. **C**. Total number of public (PSR_TCGA ≥10%), non-annotated, protein-coding junctions detected pan-cancer. **D**. Dot plots representing the positive sample rate percentage of non-annotated, protein-coding junctions in all studied cancer types (COAD, GBM, KICH, KIRP, LGG, LIHC, LUAD, LUSC, MESO, PRAD, SKCM, STAD). NJs (PSR_TCGA ≥ 10% and PSR_GTEx < 1%) are denoted by colored dots. **E-F.** Bar plots illustrating the proportion of SNIPP-characterized NJs found in the IRIS **(E)** and MAJIQlopedia **(F)** databases. **a**, Created in BioRender (credit: D.W.K., https://BioRender.com/p64h129; 2024).

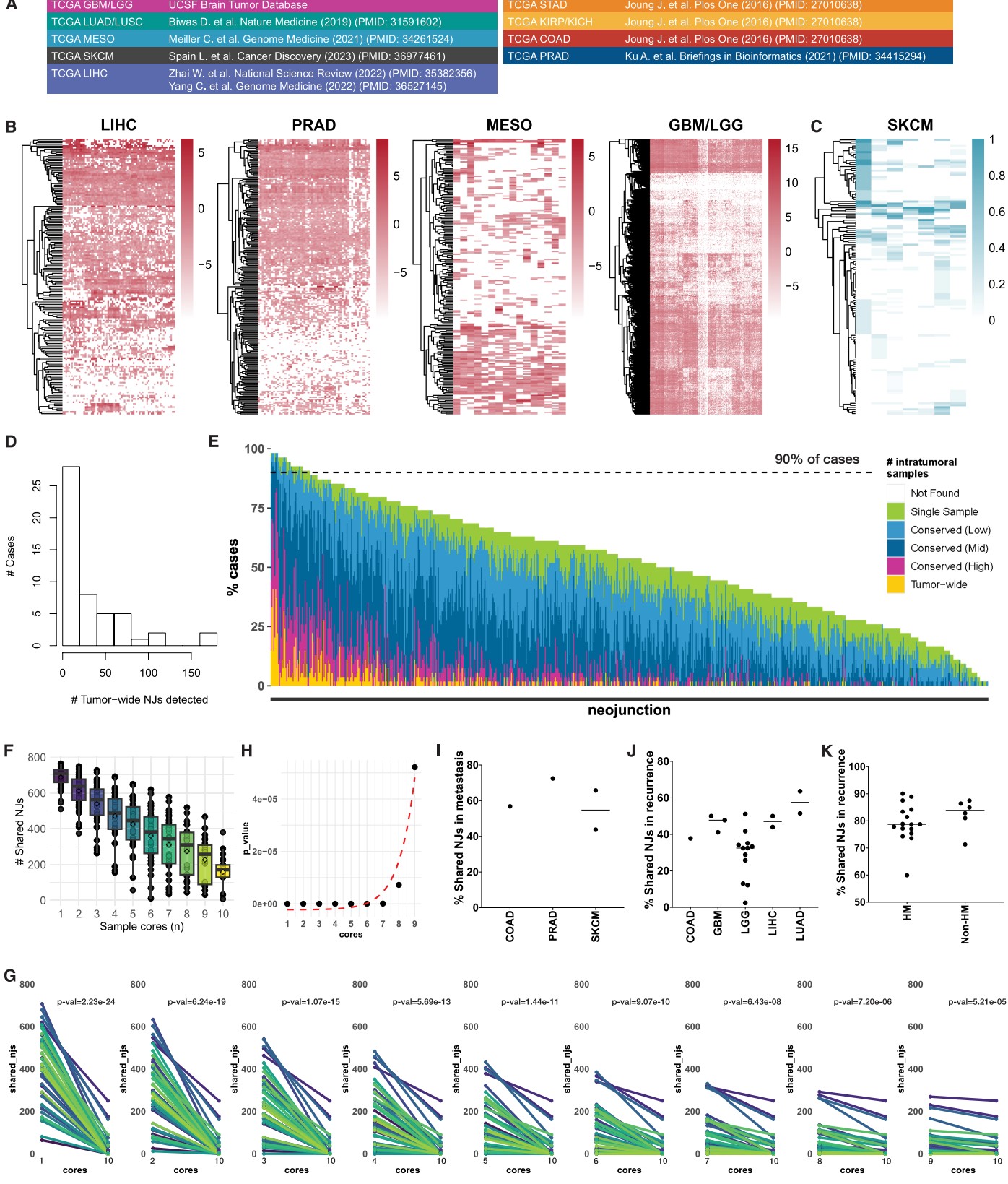

**Extended Data Fig. 2** | See next page for caption.

**Extended Data Fig. 2 | Intratumoral heterogeneity and interpatient characteristics of NJs across various cancer types. A**. Multi-region RNA-seq data of multiple cancer types were collected across various studies. Multi-region sampling is defined in studies in which multiple biopsies were isolated from the same tumor for downstream sequencing analyses. **B**. Counts per million (CPM) of non-annotated, protein-coding NJs across multi-region samples in LIHC, PRAD, MESO, GBM, and LGG cases. **C**. Heatmap illustrating the number of metastases within an SKCM patient (columns) that have a detectable expression of NJs (rows). The intensity of each cell indicates the proportion of regions within the same tumor that have putative expression of each NJ, with the intensity of 1 representing an NJ expressed in all metastases within a corresponding patient. **D**. Histogram of the number of multi-region sampled glioma cases with the corresponding number of tumor-wide NJs. **E**. Distribution of glioma-specific NJs ($n = 789$, columns) based on their ITH across patients. **F**. Total number of NJs found in $n$ cores per patient (n = 52). **G**. Slope charts demonstrating patient-matched pairs (n = 52) of the number of NJs found in $n$ cores compared to 10 cores. Paired t-test analysis was performed on all matched values, and the corresponding p-value is displayed above each slope chart iteration. **H**. Dot-plot with best-fitting curve mapping the p-values of all iterations of paired $n$ core and 10 core comparisons. **I-K**. Percentage of NJs identified in primary tumors that were conserved in paired metastases (COAD, n = 1; PRAD, n = 1; SKCM, n = 2) **(I)**, recurrence (COAD, n = 1; GBM, n = 3; LGG, n = 12; LIHC, n = 2; LUAD, n = 2) **(J)**, and recurrence following temozolomide treatment with or without hypermutation (HM) (hypermutated, n = 16, non-hypermutated, n = 6) **(K)**. Further statistical details are found in Supplementary Table 3.

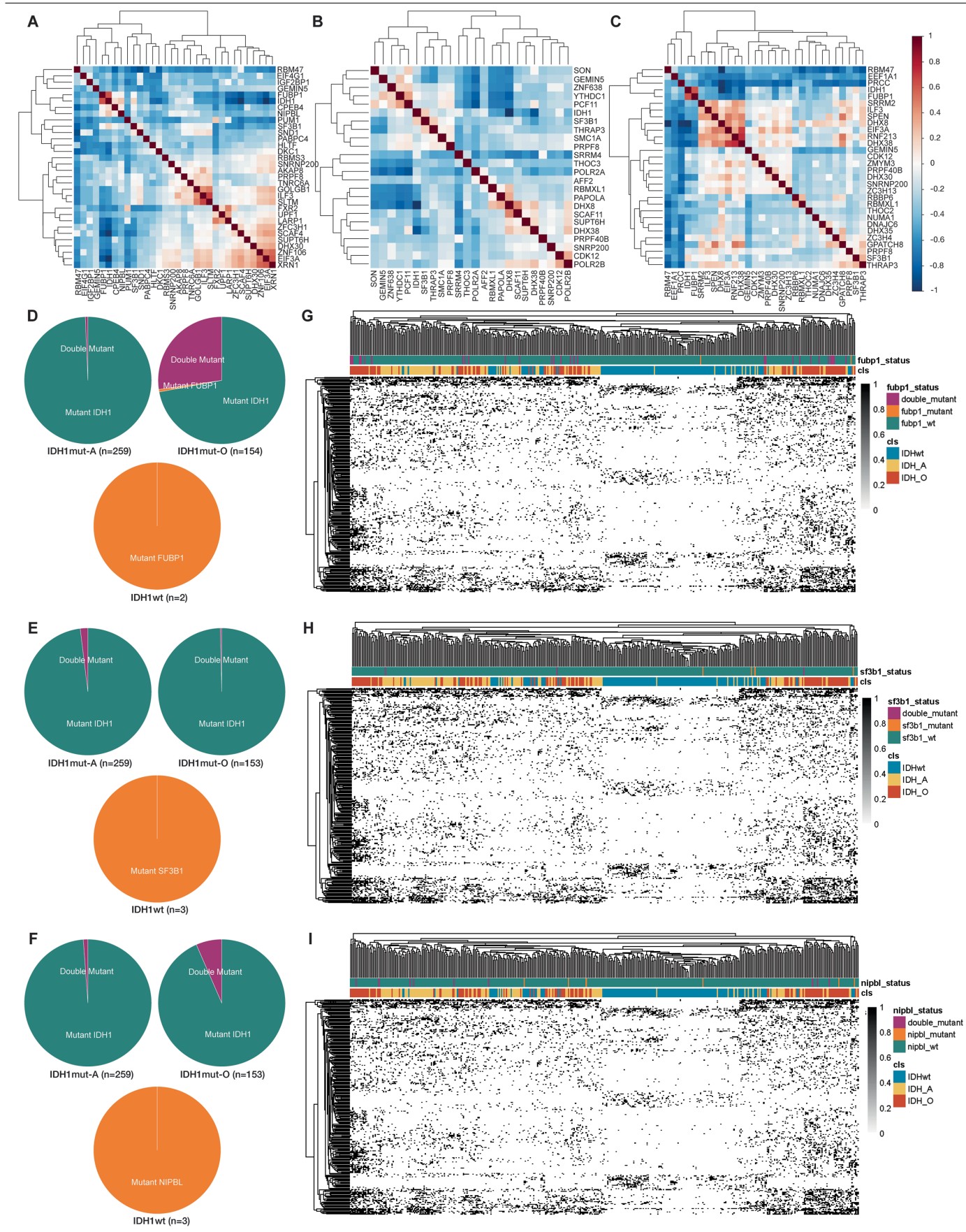

**Extended Data Fig. 3** | See next page for caption.

**Extended Data Fig. 3 | Co-occurrence of somatic mutations in splicing-related genes. A-C**. Heatmaps showing the pairwise Pearson correlation matrix between gene expression of TCGA GBM and LGG samples computed for each gene pair. Splicing-related gene lists were defined by **A**. Nostrand et al.[36], **B**. Sveen et al.[37], and **C**. Seiler et al.[38] **D-F**. Pie charts illustrating the proportions of *IDH*mut samples that also contain mutations in *FUBP1* (**D**), *SF3B1* (**E**), and *NIPBL* (**F**) in *IDH*wt GBM samples (bottom), *IDH*mut-A samples (top-left), and *IDH*mut-O samples (top-right). **G-I**. Binary heatmap demonstrating the putative expression of NJs in relation to glioma subtypes and mutation status of *FUBP1* (**G**), *SF3B1* (**H**), and *NIPBL* (**I**).

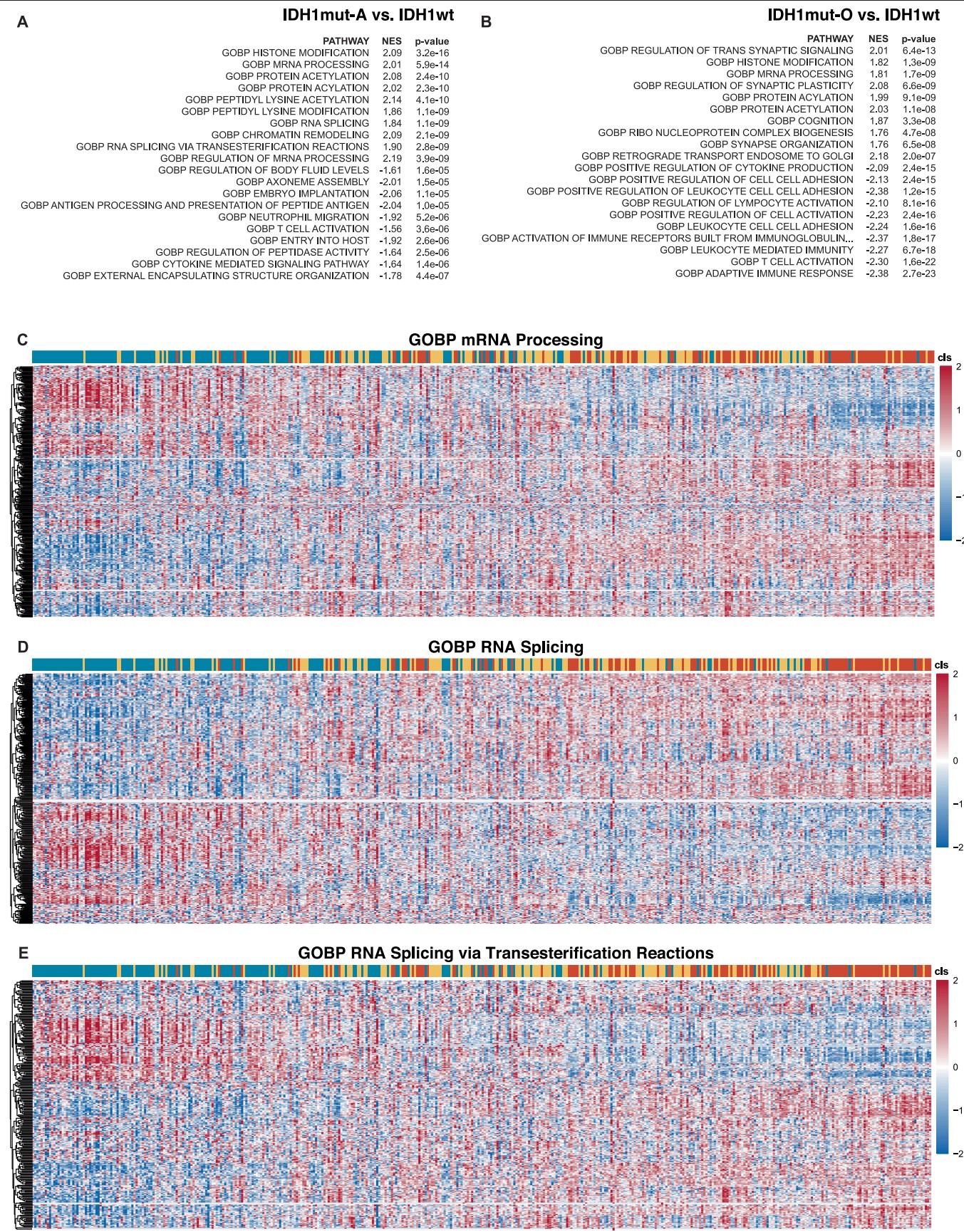

**A** IDH1mut-A vs. IDH1wt

| PATHWAY | NES | p-value |
|---|---|---|
| GOBP HISTONE MODIFICATION | 2.09 | 3.2e-16 |
| GOBP MRNA PROCESSING | 2.01 | 5.9e-14 |
| GOBP PROTEIN ACETYLATION | 2.08 | 2.4e-10 |
| GOBP PROTEIN ACYLATION | 2.02 | 2.3e-10 |
| GOBP PEPTIDYL LYSINE ACETYLATION | 2.14 | 4.1e-10 |
| GOBP PEPTIDYL LYSINE MODIFICATION | 1.86 | 1.1e-09 |
| GOBP RNA SPLICING | 1.84 | 1.1e-09 |
| GOBP CHROMATIN REMODELING | 2.09 | 2.1e-09 |
| GOBP RNA SPLICING VIA TRANSESTERIFICATION REACTIONS | 1.90 | 2.8e-09 |
| GOBP REGULATION OF MRNA PROCESSING | 2.19 | 3.9e-09 |
| GOBP REGULATION OF BODY FLUID LEVELS | -1.61 | 1.6e-05 |
| GOBP AXONEME ASSEMBLY | -2.01 | 1.5e-05 |
| GOBP EMBRYO IMPLANTATION | -2.06 | 1.1e-05 |
| GOBP ANTIGEN PROCESSING AND PRESENTATION OF PEPTIDE ANTIGEN | -2.04 | 1.0e-05 |
| GOBP NEUTROPHIL MIGRATION | -1.92 | 5.2e-06 |
| GOBP T CELL ACTIVATION | -1.56 | 3.6e-06 |
| GOBP ENTRY INTO HOST | -1.92 | 2.6e-06 |
| GOBP REGULATION OF PEPTIDASE ACTIVITY | -1.64 | 2.5e-06 |
| GOBP CYTOKINE MEDIATED SIGNALING PATHWAY | -1.64 | 1.4e-06 |
| GOBP EXTERNAL ENCAPSULATING STRUCTURE ORGANIZATION | -1.78 | 4.4e-07 |

**B** IDH1mut-O vs. IDH1wt

| PATHWAY | NES | p-value |
|---|---|---|
| GOBP REGULATION OF TRANS SYNAPTIC SIGNALING | 2.01 | 6.4e-13 |
| GOBP HISTONE MODIFICATION | 1.82 | 1.3e-09 |
| GOBP MRNA PROCESSING | 1.81 | 1.7e-09 |
| GOBP REGULATION OF SYNAPTIC PLASTICITY | 2.08 | 6.6e-09 |
| GOBP PROTEIN ACYLATION | 1.99 | 9.1e-09 |
| GOBP PROTEIN ACETYLATION | 2.03 | 1.1e-08 |
| GOBP COGNITION | 1.87 | 3.3e-08 |
| GOBP RIBO NUCLEOPROTEIN COMPLEX BIOGENESIS | 1.76 | 4.7e-08 |
| GOBP SYNAPSE ORGANIZATION | 1.76 | 6.5e-08 |
| GOBP RETROGRADE TRANSPORT ENDOSOME TO GOLGI | 2.18 | 2.0e-07 |
| GOBP POSITIVE REGULATION OF CYTOKINE PRODUCTION | -2.09 | 2.4e-15 |
| GOBP POSITIVE REGULATION OF CELL CELL ADHESION | -2.13 | 2.4e-15 |
| GOBP POSITIVE REGULATION OF LEUKOCYTE CELL CELL ADHESION | -2.38 | 1.2e-15 |
| GOBP REGULATION OF LYMPOCYTE ACTIVATION | -2.10 | 8.1e-16 |
| GOBP POSITIVE REGULATION OF CELL ACTIVATION | -2.23 | 2.4e-16 |
| GOBP LEUKOCYTE CELL CELL ADHESION | -2.24 | 1.6e-16 |
| GOBP ACTIVATION OF IMMUNE RECEPTORS BUILT FROM IMMUNOGLOBULIN... | -2.37 | 1.8e-17 |
| GOBP LEUKOCYTE MEDIATED IMMUNITY | -2.27 | 6.7e-18 |
| GOBP T CELL ACTIVATION | -2.30 | 1.6e-22 |
| GOBP ADAPTIVE IMMUNE RESPONSE | -2.38 | 2.7e-23 |

**C** GOBP mRNA Processing

**D** GOBP RNA Splicing

**E** GOBP RNA Splicing via Transesterification Reactions

Histology IDH1wt IDH1mutA IDH1mutO

**Extended Data Fig. 4** | See next page for caption.

**Extended Data Fig. 4 | Glioma subtype-specific aberrations in splicing factor expression lead to differential levels of NJ expression. A**. Ranked log2 fold change of genes within the top enriched pathways within GOBP when comparing *IDH*mut-A samples against *IDH*wt samples. **B**. Ranked log2 fold change of genes within the top enriched pathways within GOBP when comparing *IDH*mut-O samples against *IDH*wt samples. **C-E**. Heatmap demonstrating hierarchal clustering of splicing-related genes (rows) in GOBP mRNA Processing **(C)**, GOBP RNA Splicing **(D)**, and GOBP RNA Splicing via Transesterification Reactions **(E)** in TCGA samples (columns) ordered by the total number of expressed putative NJs. Further statistical details are found in Supplementary Table 3.

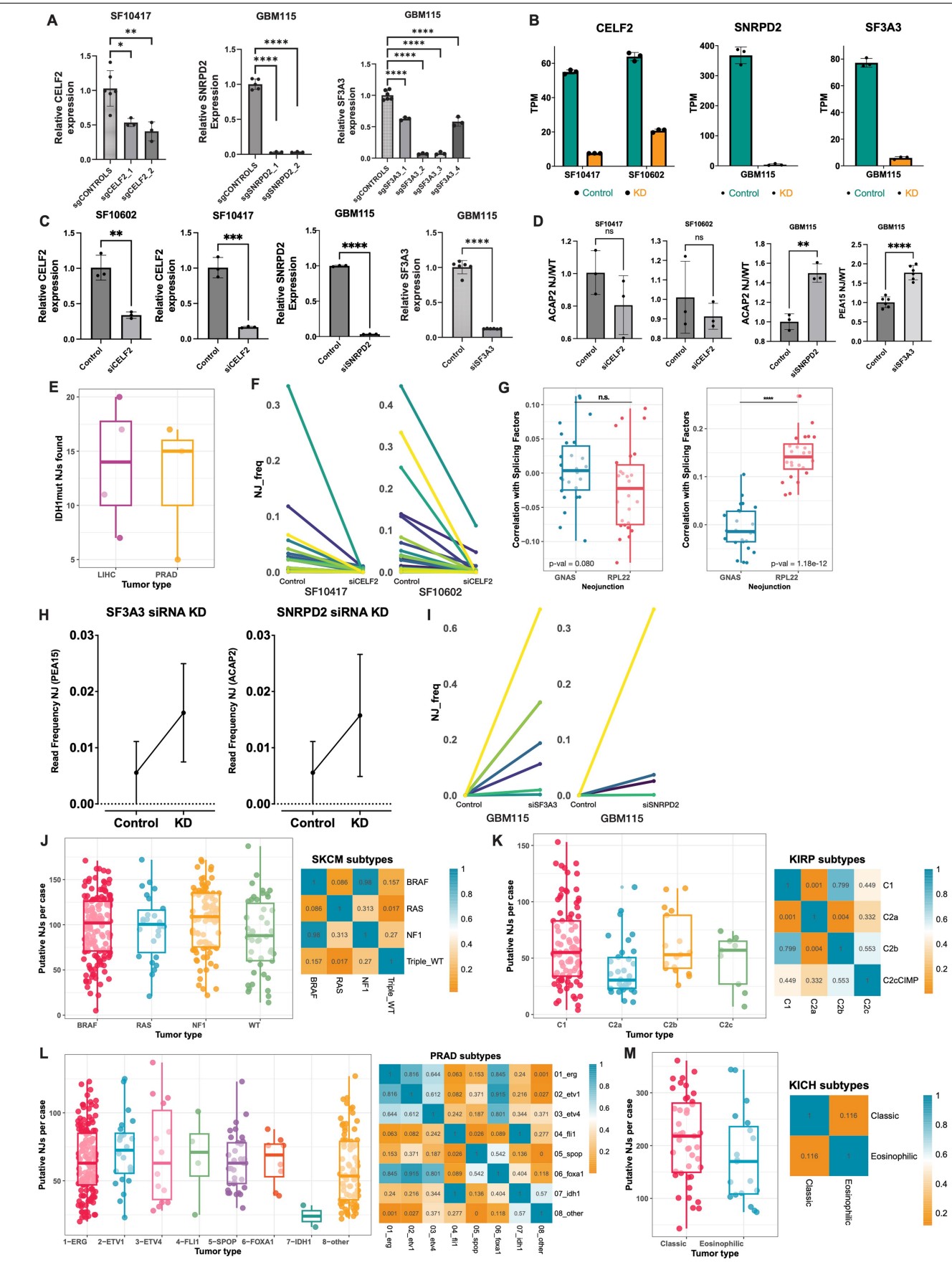

**Extended Data Fig. 5** | See next page for caption.

**Extended Data Fig. 5 | Dysregulated expression of canonical splicing-related genes is associated with disease subtype-specific expression of NJs.**
**A**. qPCR validation of target gene knockdown by CRISPRi was performed on glioma cell lines (n = 3). **B-C**. RNA-seq-derived **(B)** and qPCR **(C)** TPM expression of *CELF2*, *SNRPD2*, and *SF3A3* following siRNA knockdown (n = 3). **D**. (Left) Expression of NJ$_{ACAP2}$ in LGG (SF10417 and SF10602) cell line treated with control siRNA or si*CELF2* (n = 3). (Right) Expression of NJ$_{ACAP2}$ or NJ$_{PEAIS}$ in GBM (GBM115) cell line treated with control siRNA or si*SNRPD2* or si*SF3A3*, respectively (n = 3). **E**. Detection of glioma-derived *IDH*mut NJs in *IDH*mut TCGA LIHC (n = 4) and PRAD (n = 3) samples. **F**. Slope plots demonstrating a decrease in the frequency of *IDH*mut-specific NJs in *CELF2* siRNA-treated SF10417 (left) and SF10602 cells (right). **G**. Correlation of *IDH*mut-specific splicing-related genes (right, n = 26) and chromosome 1p and 19q splicing-related genes (left, n = 25) with NJ$_{GNAS}$

and NJ$_{RPL22}$. **H**. RNA-seq-derived read frequency of NJ$_{PEAIS}$ and NJ$_{ACAP2}$ in GBM115 cells (n = 3) treated with control siRNA and si*SF3A3* (left) or control siRNA and si*SNRPD2* (right), respectively. **I**. Slope plots demonstrating an increase in the frequency of *IDH*mut-O-specific NJs in *SF3A3* siRNA- (left) and *SNRPD2* siRNA-treated (right) GBM115 cells. **J-M**. Bar plot (left) showing the total NJs expressed per case across all disease subtypes and heatmap (right) displaying the Wilcoxon rank-sum test of NJ expression between each subtype within TCGA SKCM (BRAF, n = 150; RAS, n = 91; NF1, n = 26; Triple WT, n = 46) **(J)**, KIRP (C1, n = 89; C2a, n = 34, C2b, n = 17; C2cCIMP, n = 9) **(K)**, PRAD (ERG, n = 145; ETV1, n = 24; ETV4, n = 14; FLI1, n = 4; SPOP, n = 33; FOXA1, n = 8; IDH1, n = 2; other, n = 80) **(L)**, KICH (Eosinophilic, n = 19; Classic, n = 43) **(M)**. Further statistical details are found in Supplementary Table 3.

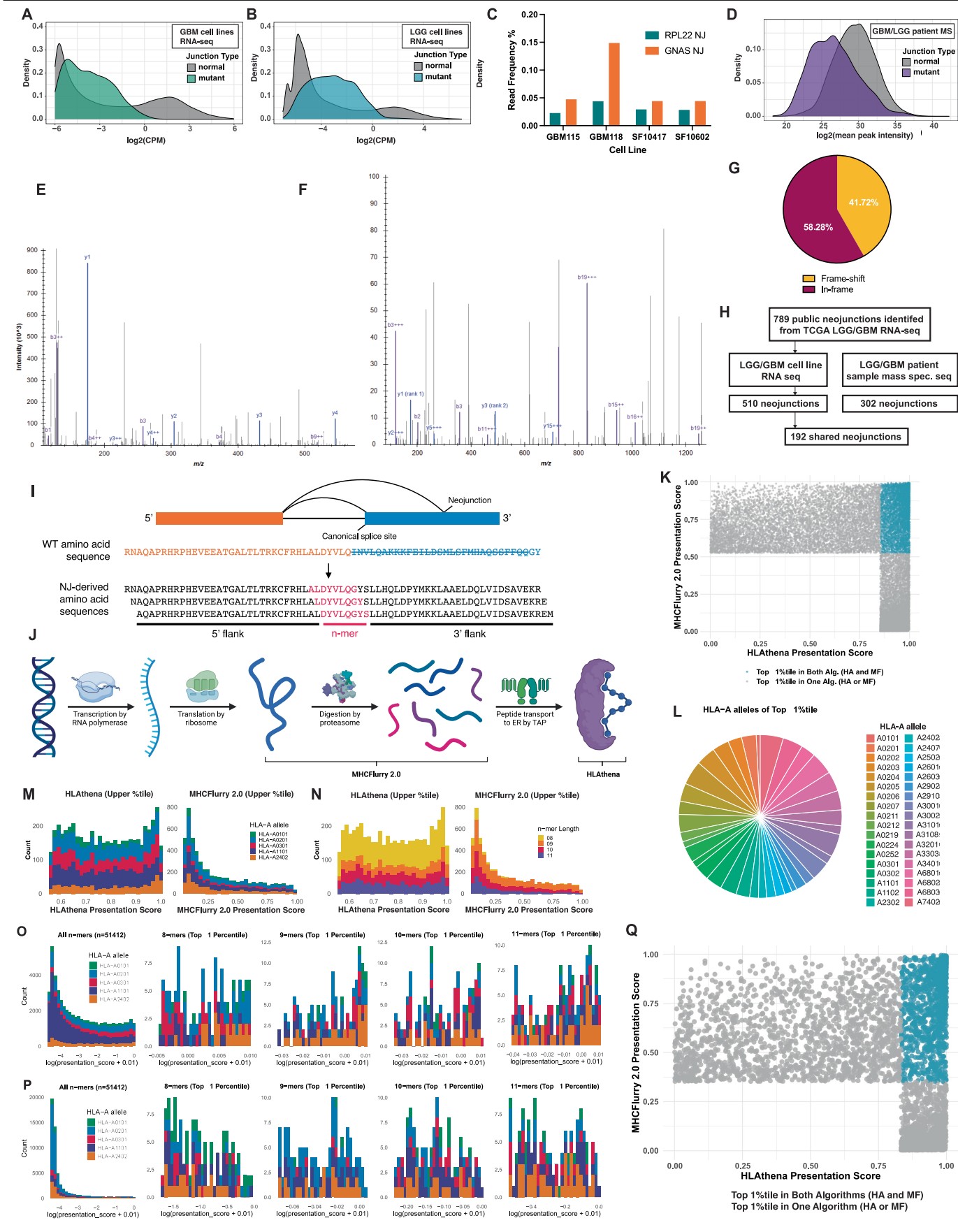

**Extended Data Fig. 6** | See next page for caption.

**Extended Data Fig. 6 | NJ-derived neoepitopes are predicted to be processed and presented by HLA. A-B**. Density plots depicting $\log_2$(CPM) of junction reads from RNA-seq in patient-derived GBM (**A**) and LGG (**B**) cell lines. Detectable NJ expression (colored) is validated against canonical splicing (gray). **C**. Read frequency spanning NJs in *RPL22* and *GNAS* compared to the canonical junction spanning reads in glioma cell lines ($n = 1$). **D**. Density plot depicting MS analysis of publicly available LGG and GBM data sets ($n = 447$) reveals comparable log2(peak intensity) for NJ-derived peptides (purple) and endogenous peptides (gray). **E-F**. Mass spectra of peptide sequences spanning the aberrantly spliced regions in (**E**) RPL22 and (**F**) GNAS detected in publicly available glioma MS data. **G**. Proportion of MS-detected NJs that encode for frame-shift or in-frame mutations. **H**. Schematic of the selection of 192 high-confidence NJs based on RNA-seq and MS detection. **I**. Diagram illustrating a mechanism of neoantigen production and peptide bank generation for prediction analysis. **J**. Schematic depicting biological steps leading to the generation of HLA class I-presented antigens. SSNIP considers the pre-presentation steps of proteasomal processing and HLA-binding. **K**. Dot plot showing the overlay of the top scoring 1-percentile of HLAthena and MHCflurry 2.0 algorithms against neopeptide candidates presented by all demographically predominant HLA-A haplotypes. Top-scoring final candidates are indicated in blue as the candidates that scored in the top 1 percentile in both algorithms. **L**. Pie chart illustrating the distribution of neopeptide candidates found in the overlapping top 1 percentile based on composite HLA-A haplotype score. **M-N**. Histogram of peptide presentation likelihood scores for all top 1-percentile *n*-base polypeptides categorized by HLA-allele (**M**) or *n*-base polypeptides length (**N**), and varying *n*-base polypeptides lengths in HLAthena (**O**) and MHCflurry 2.0 (**P**). **Q**. Dot plot overlaying top 1-percentile candidates from two algorithms, highlighting final candidates (blue) scoring highly in both. **j**, Created in BioRender (credit: D.W.K., https://BioRender.com/r01o115; 2024).

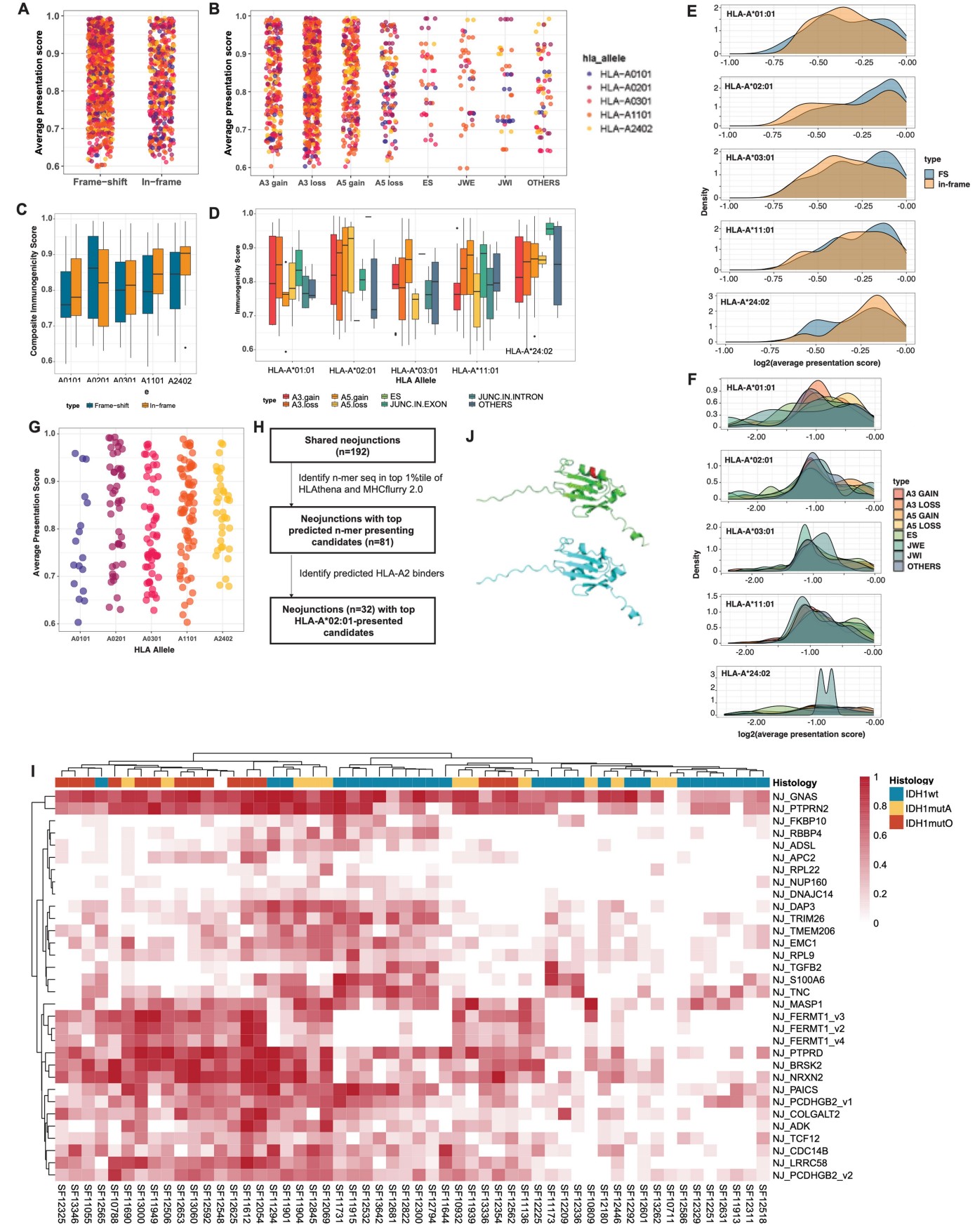

**Extended Data Fig. 7 |** See next page for caption.

**Extended Data Fig. 7 | NEJs generate a diverse portfolio of splicing aberrations capable of generating presentable neoepitopes. A**. Jitter plot corresponding to the average presentation scores of peptides derived from NEJs generating frameshifts or in-frame mutations. **B**. Jitter plot corresponding to the average presentation scores of peptides derived from NEJs derived from various splice types. **C-D**. Box-and-whisker plots illustrating composite presentation scores of candidates by HLA-allele (HLA-A*01:01, n = 117; HLA-A*02:01, n = 276; HLA-A*03:01, n = 294; HLA-A*11:01, n = 440; HLA-A*24:02, n = 230) based on frame-shift status (frame-shift, n = 893; in-frame, n = 464) (**C**) or alternative splicing category (alternative 3' splice site gain, n = 267; alternative 3' splice site loss, n = 518; alternative 5' splice site gain, n = 275; alternative 5' splice site loss, n = 119; exon skip, n = 37; junction within exon, n = 43; junction within intron, n = 38; others, n = 60) (**D**). **E-F**. Density plots depicting the average presentation scores of neoantigens derived from NJs generating (**E**) frame shifts (FS) or (**F**) various splice types presented by HLA-A*01:01, HLA-A*02:01, HLA-A*03:01, HLA-A*11:01, and HLA-A*24:02. **G**. Composite presentation scores for validated *n*-base polypeptides detected in RNA-seq and MS data. **H**. Schematic of final NEJ derivation, focusing on top HLA-A*02:01-presented candidates. **I**. Heatmap illustrating intratumoral heterogeneity of final candidate HLA-A*02:01-presented NJs across all spatially-mapped glioma samples. Glioma subtypes analyzed in this study include *IDH*wt (blue), *IDH*mut-A (yellow), *IDH*mut-O (red). **J**. AlphaFold2 protein structure prediction of wildtype RPL22 (top) and NEJ variant of RPL22 (bottom).

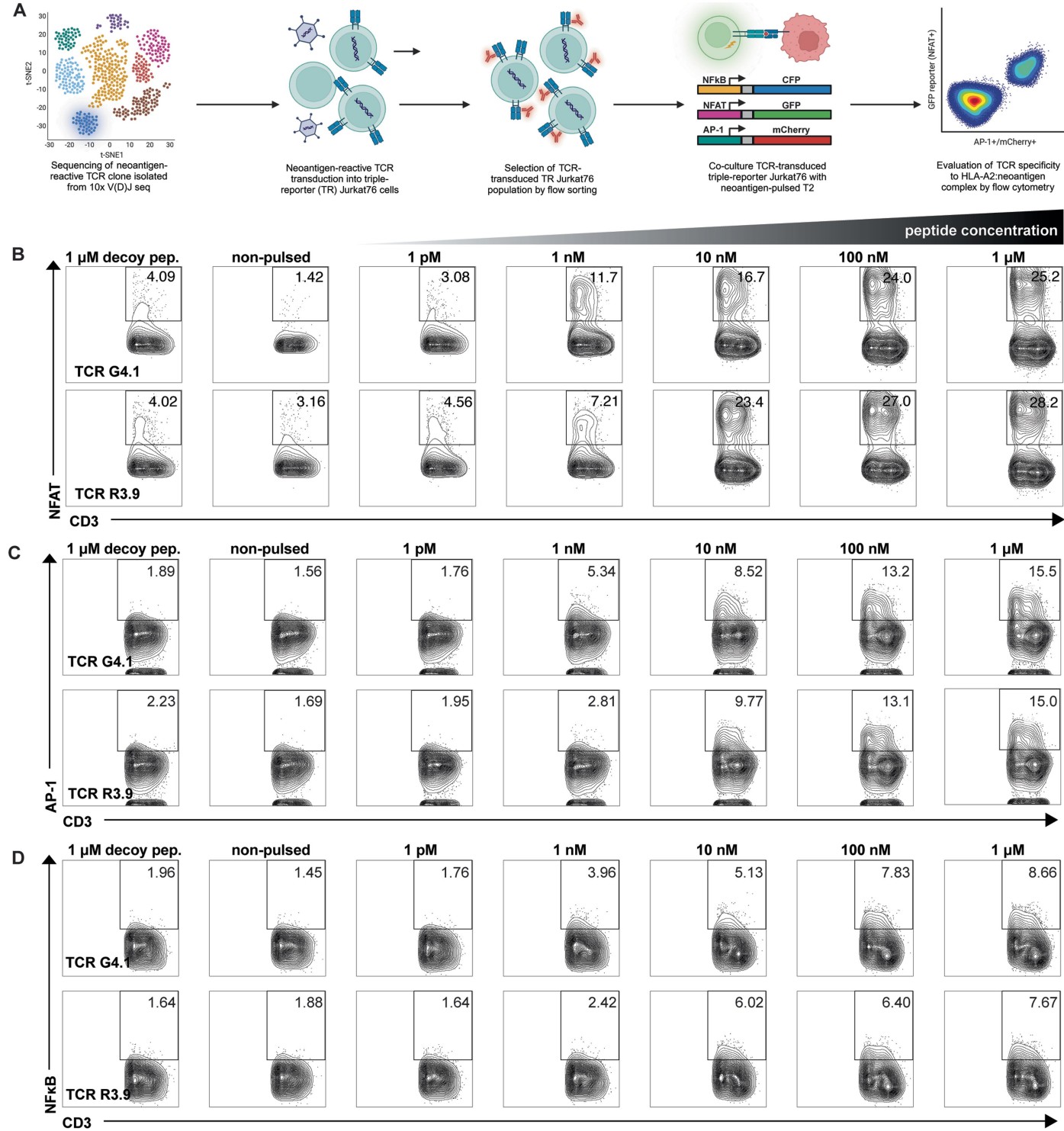

**Extended Data Fig. 8 | Neoantigen-reactive T-cell clones are isolated from PBMC and elicit an immune response upon neoantigen recognition.**
**A.** Pipeline for validating the specificity of neoantigen-reactive TCR clonotypes found in 10x V(D)J single-cell RNA-seq (scRNA-seq) against NJ-derived neoantigen candidates utilizing a TCR-transduced triple-reporter Jurkat76/CD8 system followed by flow cytometry analysis. **B-D**. NEJ$_{GNAS}$-derived and NEJ$_{RPL22}$-derived

neoantigen-specific TCR-transduced triple-reporter Jurkat76 cells activated against dose-dependent neoantigen-pulsed T2 cells. TCR activation of triple-reporter TCR-transduced triple-reporter Jurkat76 is measured by flow cytometry analysis of (**B**) NFAT-GFP, (**C**) AP-1-mCherry, and (**D**) NFκB-CFP. **a**, Created in BioRender (credit: D.W.K., https://BioRender.com/z56q380; 2024).

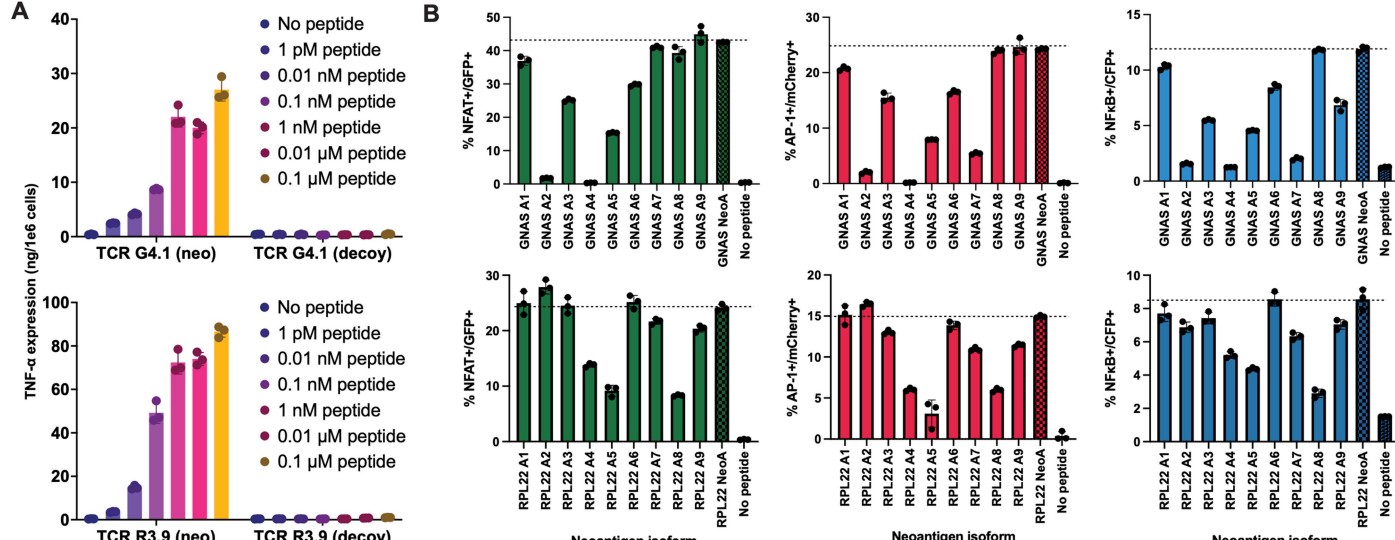

**Extended Data Fig. 9 | High-affinity neoantigen-reactive T-cell receptors recognize cancer-specific neoantigen sequences. A**. TNFα expression of PBMC-derived CD8⁺ T-cells (*n* = 3) transduced with TCR$_{G4.1}$ (top) and TCR$_{R3.9}$ (bottom) against T2 cells pulsed with varying concentrations of corresponding neoantigen or decoy antigen. **B**. Alanine scanning mutagenesis of NeoA$_{GNAS}$ (top) and NeoA$_{RPL22}$-reactive (bottom) TCR-transduced triple-reporter Jurkat76/CD8 cells co-cultured with alanine-substituted neoantigen-pulsed T2 cells (n = 3), neoantigen-pulsed T2 cells (n = 3), or non-pulsed T2 cells (n = 3). Flow analysis was performed to evaluate TCR activity through NFAT-GFP (left), AP-1-mCherry (center), and NFκB-CFP (right) activity.

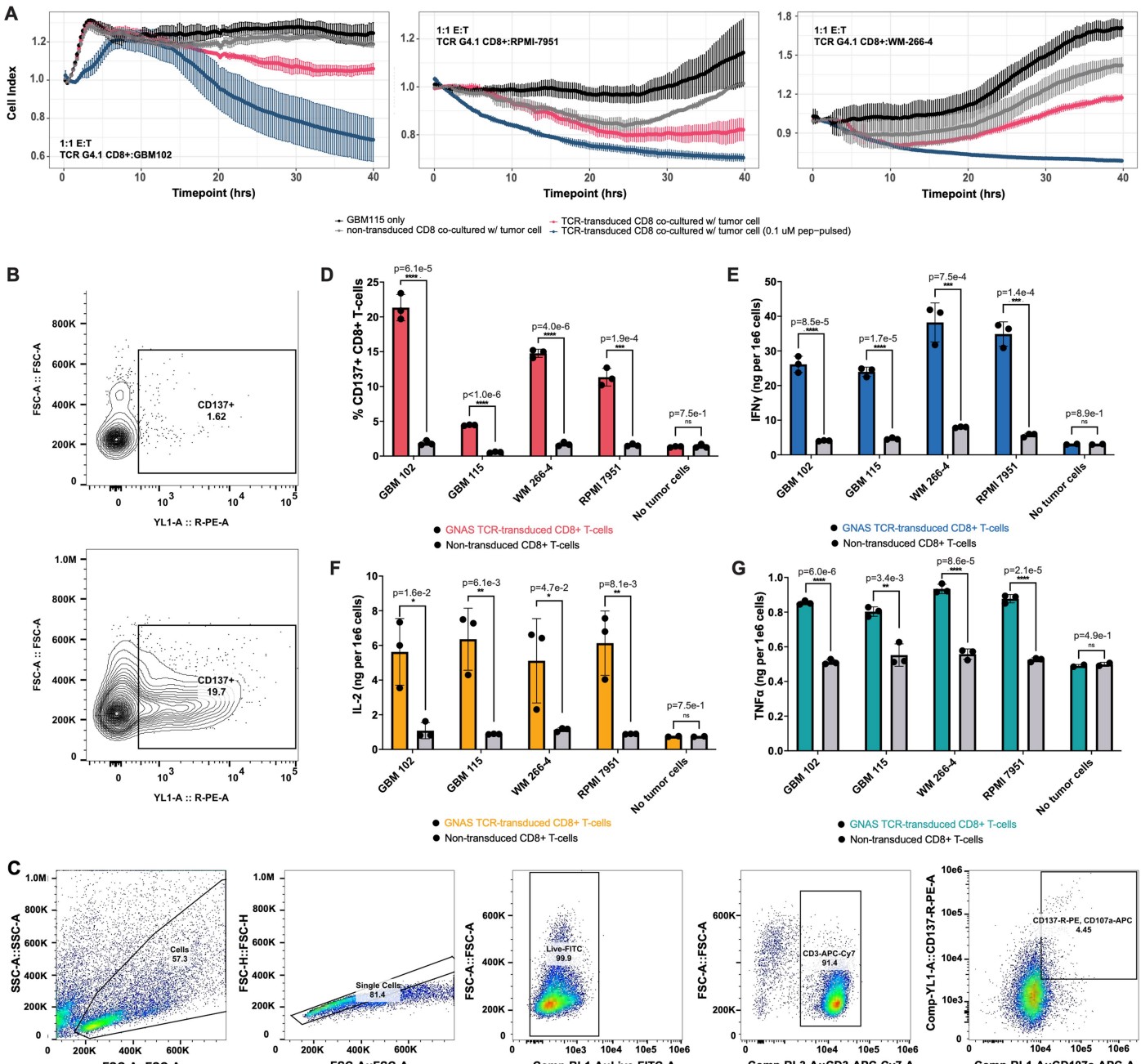

**Extended Data Fig. 10 | NeoA_GNAS-specific cytotoxicity by NEJ_GNAS-specific TCR-transduced CD8+ T-cells. A**. NeoA_GNAS-specific TCR-transduced (colored), or non-transduced (gray) CD8+ T-cells cultured against GBM102 (left), RPMI-7951 (center), and WM-266-4 (right) on the xCELLigence plate platform. The assay was performed at an E:T ratio of 2:1. Cytotoxic killing was determined as the reduction of cell index compared to the control co-cultures with non-transduced CD8+ T-cells (gray) or no CD8+ T-cell introduction (black) at a given time point. (n = 3) **B**. Representative flow gating of surface CD137 expression in non-transduced (top) and TCR-transduced (bottom) CD8+ T-cells co-cultured against cancer cell lines. **C**. Gating strategy for flow cytometry experiments. Further statistical details are found in Supplementary Table 3. **D**. Bar plot of the surface expression of CD137 on NeoA_GNAS-TCR-transduced (red) or non-transduced (gray) CD8+ T-cells when cultured with tumor cell lines (n = 3). **E-F**. ELISA readout of secreted IFNγ **(E)**, IL-2 **(F)**, and TNFα **(G)** by NeoA_GNAS-TCR-transduced (colored) or non-transduced (gray) CD8+ T-cells when cultured with tumor cell lines (n = 3). Further statistical details are found in Supplementary Table 3.

# Reporting Summary

## Statistics

For all statistical analyses, confirm that the following items are present in the figure legend, table legend, main text, or Methods section.

| n/a | Confirmed | |
|---|---|---|
| ☐ | ☒ | The exact sample size (*n*) for each experimental group/condition, given as a discrete number and unit of measurement |
| ☒ | ☐ | A statement on whether measurements were taken from distinct samples or whether the same sample was measured repeatedly |
| ☐ | ☒ | The statistical test(s) used AND whether they are one- or two-sided *Only common tests should be described solely by name; describe more complex techniques in the Methods section.* |
| ☒ | ☐ | A description of all covariates tested |
| ☒ | ☐ | A description of any assumptions or corrections, such as tests of normality and adjustment for multiple comparisons |
| ☐ | ☒ | A full description of the statistical parameters including central tendency (e.g. means) or other basic estimates (e.g. regression coefficient) AND variation (e.g. standard deviation) or associated estimates of uncertainty (e.g. confidence intervals) |
| ☐ | ☒ | For null hypothesis testing, the test statistic (e.g. *F*, *t*, *r*) with confidence intervals, effect sizes, degrees of freedom and *P* value noted *Give P values as exact values whenever suitable.* |
| ☒ | ☐ | For Bayesian analysis, information on the choice of priors and Markov chain Monte Carlo settings |
| ☒ | ☐ | For hierarchical and complex designs, identification of the appropriate level for tests and full reporting of outcomes |
| ☐ | ☒ | Estimates of effect sizes (e.g. Cohen's *d*, Pearson's *r*), indicating how they were calculated |

*Our web collection on statistics for biologists contains articles on many of the points above.*

## Software and code

Policy information about availability of computer code

| Data collection | Softwares utilized for data collection are detailed below. These softwares include Picard and STAR aligner for RNA-sequencing analysis, MHCFlurry 2.0 and HLAthena for peptide processing and presentation prediction, Cell Ranger for single-cell RNA-sequencing analysis, and MaxQuant for mass spectrometry peptide detection. |
|---|---|
| Data analysis | Analysis of neojunction expression within multi-region samples across all cancer types were conducted with our neojunction prediction pipeline if the FASTQ file is available. If RNA-sequencing data is only available in BAM format, the sequencing file is converted into FASTQ format utilizing the Picard software (version 2.7.7a). All downloaded RNA-sequencing data sets were individually aligned using a STAR aligner-based processing pipeline (version 2.7.7a). For public cancer-specific splicing event counting, we designed a custom R script that detected and quantified non-annotated, cancer-specific splicing events found across each corresponding patient cohort. Splicing events detected in the GRCh37.87 GTF sj.out.tab (GENCODE v33) file were removed to define non-annotated splicing junctions. A library of all cancer-specific peptides were selected by removing those detectable in normal tissue peptide isoforms in a reference human proteome dataset (UniProt Proteome ID #UP000005640). All cancer-specific peptides with their upstream and downstream flanking sequences (maximum flanking length of 30 amino acids) were independently analyzed and ranked by MHCFlurry 2.0 (v2.1.3) and HLAthena MSiC (v1.0.0). HLA-I binding affinity was assessed against HLA-A*01:01, HLA-A*02:01, HLA-A*03:01, HLA-A*11:01, and HLA-A*24:02 in both cases. In the HLAthena evaluation of antigen binding and presentation to the corresponding HLA haplotypes, peptides were assigned to alleles by rank with a threshold of 0.1. Context of up to 30 flanking amino acids on both N and C terminus were utilized with aggregation by peptide and no log-transformed expression. Baseline MHCFlurry 2.0 models with both peptide:MHC-I binding affinity (BA) predictor and antigen processing (AP) predictor was used. Cell Ranger 7.0.0 (10x Genomics Cloud Analysis) was used to pre-process raw single-cell RNA sequencing and identifying V(D)J clonotypes. Differential gene expression of TCGA, GTEx, and UCSF GBM/LGG RNA-sequencing was performed and quantified using DESeq2. Visualizations of protein folding following splicing aberrations were generated using AlphaFold2. Protein detection from mass spectrometry data was performed utilizing MaxQuant (v1.6.17.0). Flow analysis was performed using FlowJo (v10.9.0). |

Custom codes used for analysis are available through the following GitHub link: https://github.com/dakwok/SSNIP

For manuscripts utilizing custom algorithms or software that are central to the research but not yet described in published literature, software must be made available to editors and reviewers. We strongly encourage code deposition in a community repository (e.g. GitHub). See the Nature Portfolio guidelines for submitting code & software for further information.

## Data

Policy information about availability of data

All manuscripts must include a data availability statement. This statement should provide the following information, where applicable:
- Accession codes, unique identifiers, or web links for publicly available datasets
- A description of any restrictions on data availability
- For clinical datasets or third party data, please ensure that the statement adheres to our policy

Spatially-mapped glioma biopsy RNA-seq datasets are deposited in the European Genome-Phenome Archive under accession numbers EGAS00001007986, EGAS00001006785, EGAD00001005221/2, EGAD00001009496/7. Spatially-mapped biopsy RNA-seq data for other tumor types were retrieved from their corresponding publications: Through the NIH SRA at https://www.ncbi.nlm.nih.gov/sra, RNA-seq data can be accessed with accession ID PRJNA579899 for Ku et al. 2018, SRP066596 for Joung et al. 2016. Through the National Omics Data Encyclopedia (NODE), RNA-seq data can be accessed with accession code OEP002956 for Yang et al. 2022. Through EGA, RNA-seq data can be accessed with accession code EGAD00001009042 for Jeon et al. 2023, EGAS00001003813 for Zhai et al. 2023, EGAS00001005328 for Meiller et al. 2021. TRACERx data was requested and received from the Cancer Research UK & University College London Cancer Trials Centre. Glioma MS data was retrieved from the CPTAC Consortium as well as the Proteomics Identifications Database (PRIDE). PRIDE accession code for Bader et al. 2021 is PXD024427. Proteomic data from Wong et al. 2022 was retrieved from their supplementary files.

## Research involving human participants, their data, or biological material

Policy information about studies with human participants or human data. See also policy information about sex, gender (identity/presentation), and sexual orientation and race, ethnicity and racism.

| | |
|---|---|
| Reporting on sex and gender | N/A |
| Reporting on race, ethnicity, or other socially relevant groupings | N/A |
| Population characteristics | N/A |
| Recruitment | N/A |
| Ethics oversight | N/A |

Note that full information on the approval of the study protocol must also be provided in the manuscript.

# Field-specific reporting

Please select the one below that is the best fit for your research. If you are not sure, read the appropriate sections before making your selection.

☒ Life sciences   ☐ Behavioural & social sciences   ☐ Ecological, evolutionary & environmental sciences

For a reference copy of the document with all sections, see nature.com/documents/nr-reporting-summary-flat.pdf

# Life sciences study design

All studies must disclose on these points even when the disclosure is negative.

| | |
|---|---|
| Sample size | Experimental sample size was determined based on significance, expectations of variability for specific experiment types, and by feasibility in large scale undertakings. Biological and technical triplication was carried out for all experiments to ensure statistical robustness. No formal sample size calculation was performed for this study. The sample sizes were chosen based on practical considerations, including the typical variability observed in similar experiments and the feasibility of generating or analyzing samples within the constraints of available resources, such as time, budget, and experimental materials. |
| Data exclusions | No data were excluded. |
| Replication | Data were reproduced at technical, biological, and experimental levels where relevant replication information is provided. Repeats agreed with one another, and all findings described in this manuscript were confirmed in at least two independent experimental repeats, demonstrating reliable reproducibility. To ensure robustness, flow analysis experiments were performed in most cases using two independent target antibodies. |
| Randomization | Randomization was not relevant to our study because all experiments were conducted in a controlled benchwork setting where external sources of variability were minimized. The experimental procedures involved standardized reagents, equipment, and protocols, ensuring |

uniform conditions across all replicates.

Blinding | Blinding is not applicable to this study.

# Reporting for specific materials, systems and methods

We require information from authors about some types of materials, experimental systems and methods used in many studies. Here, indicate whether each material, system or method listed is relevant to your study. If you are not sure if a list item applies to your research, read the appropriate section before selecting a response.

## Materials & experimental systems

| n/a | Involved in the study |
|---|---|
| ☐ | ☒ Antibodies |
| ☐ | ☒ Eukaryotic cell lines |
| ☒ | ☐ Palaeontology and archaeology |
| ☒ | ☐ Animals and other organisms |
| ☒ | ☐ Clinical data |
| ☒ | ☐ Dual use research of concern |
| ☒ | ☐ Plants |

## Methods

| n/a | Involved in the study |
|---|---|
| ☒ | ☐ ChIP-seq |
| ☐ | ☒ Flow cytometry |
| ☒ | ☐ MRI-based neuroimaging |

## Antibodies

| Antibodies used | APC anti-human HLA-A2 Antibody [clone BB7.2] Invitrogen Ref #17-9876-42<br>APC anti-human CD8a Antibody [clone SK1] BioLegend Cat #344721<br>APC anti-human CD107a Antibody (LAMP-1) [clone H4A3] BioLegend Cat #328620<br>APC anti-human TCR α/β Antibody [clone IP26] BioLegend Cat #306718<br>APC/Cyanine7 anti-human TCR α/β [clone IP26] BioLegend Cat #306728<br>APC/Cyanine7 anti-human CD3 Antibody [SK7] BioLegend Cat #344818<br>FITC anti-human HLA-A3 Antibody [clone BB7.2] BioLegend Cat #343304<br>FITC anti-human CD3 Antibody [clone UCHT1] BioLegend Cat #300440<br>PE anti-human TCR α/β Antibody [clone EP26] BioLegend Cat # 306708<br>PE anti-human HLA-A2 Antibody [clone BB7.2] BioLegend Cat #343306<br>PE anti-mouse TCR β chain Antibody [clone H57-597] BioLegend Cat #109208<br>PE anti-human CD137 (4-1BB) Antibody [clone 4B4-1] BioLegend Cat #309804<br>PE anti-human CD3 Antibody [clone HIT3a] BioLegend Cat #300308<br>Zombie GreenTM Fixable Viability Kit BioLegend Cat #423112<br>InVivoMAb anti-human MHC Class I Antibody W6/32, Bio X Cell, Cat. #BE0079<br>Unless otherwise stated, the concentration of antibody used is the one recommended by the manufacturer.<br>Lot number information was not recorded. |
|---|---|
| Validation | All antibodies used in this study are commercially available and have been validated by the manufacturer or in previous reports. In this specific study, the antibodies were not validated as we used trusted commercial sources and followed the manufacturer's recommended protocols and applications:<br><br>APC anti-human HLA-A2 Antibody [clone BB7.2] Invitrogen Ref #17-9876-42 (https://www.thermofisher.com/antibody/product/HLA-A2-Antibody-clone-BB7-2-Monoclonal/17-9876-42)<br><br>APC anti-human CD8a Antibody [clone SK1] BioLegend Cat #344721 (https://www.biolegend.com/en-ie/products/apc-anti-human-cd8-antibody-6531)<br><br>APC anti-human CD107a Antibody (LAMP-1) [clone H4A3] BioLegend Cat #328620 (https://www.biolegend.com/de-at/products/apc-anti-human-cd107a-lamp-1-antibody-5428)<br><br>APC anti-human TCR α/β Antibody [clone IP26] BioLegend Cat #306718 (https://www.biolegend.com/fr-ch/products/apc-anti-human-tcr-alpha-beta-antibody-6704)<br><br>APC/Cyanine7 anti-human TCR α/β [clone IP26] BioLegend Cat #306728 (https://www.biolegend.com/en-gb/products/apc-cyanine7-anti-human-tcr-alpha-beta-antibody-12516)<br><br>APC/Cyanine7 anti-human CD3 Antibody [SK7] BioLegend Cat #344818 (https://www.biolegend.com/en-gb/products/apc-cyanine7-anti-human-cd3-antibody-6940?GroupID=BLG5900)<br><br>FITC anti-human HLA-A3 Antibody [clone BB7.2] BioLegend Cat #343304 (https://www.biolegend.com/nl-nl/products/fitc-anti-human-hla-a2-antibody-6018?GroupID=BLG7410)<br><br>FITC anti-human CD3 Antibody [clone UCHT1] BioLegend Cat #300440 (https://www.biolegend.com/en-gb/clone-search/fitc-anti-human-cd3-antibody-863?GroupID=BLG5900)<br><br>PE anti-human TCR α/β Antibody [clone EP26] BioLegend Cat # 306708 (https://www.biolegend.com/en-ie/products/pe-anti-human- |

alpha-beta-t-cell-receptor-antibody-773?GroupID=GROUP28)

PE anti-human HLA-A2 Antibody [clone BB7.2] BioLegend Cat #343306 (https://www.biolegend.com/fr-fr/products/pe-anti-human-hla-a2-antibody-6175)

PE anti-mouse TCR β chain Antibody [clone H57-597] BioLegend Cat #109208 (https://www.biolegend.com/de-at/products/pe-anti-mouse-tcr-beta-chain-antibody-272)

PE anti-human CD137 (4-1BB) Antibody [clone 4B4-1] BioLegend Cat #309804 (https://www.biolegend.com/en-gb/products/pe-anti-human-cd137-4-1bb-antibody-1510?GroupID=BLG2203)

PE anti-human CD3 Antibody [clone HIT3a] BioLegend Cat #300308 (https://www.biolegend.com/fr-lu/products/pe-anti-human-cd3-antibody-753)

Zombie GreenTM Fixable Viability Kit BioLegend Cat #423112 (https://www.biolegend.com/fr-ch/products/zombie-green-fixable-viability-kit-9340)

InVivoMAb anti-human MHC Class I Antibody W6/32, Bio X Cell, Cat. #BE0079 (https://bioxcell.com/invivomab-anti-human-mhc-class-i-hla-a-hla-b-hla-c-be0079)

# Eukaryotic cell lines

Policy information about cell lines and Sex and Gender in Research

| Cell line source(s) | GBM cell lines were obtained from the Mayo Clinic Brain Tumor Patient Derived Xenograft National Resource (Vaubel et al., 2020). LGG cell lines were obtained from an in-house consortium (Jones et al., 2020). Triple-reporter Jurkat76 cells were obtained by MTA through the Robert Prins Lab at UCLA. COS7 and T2 cells were obtained through ATCC. Pan-cancer cell lines were obtained through a various sources as referenced in Stevers et al., 2023. |
|---|---|
| Authentication | Cell lines were authenticated by short tandem repeat (STR) analysis at the University of California Berkeley Sequencing Facility and by the commercial providers where applicable. |
| Mycoplasma contamination | Cell lines were confirmed to be Mycoplasma free by PCR using previously published methods. |
| Commonly misidentified lines (See ICLAC register) | No commonly misidentified cell lines were used in this study. |

# Plants

| Seed stocks | N/A |
|---|---|
| Novel plant genotypes | N/A |
| Authentication | N/A |

# Flow Cytometry

## Plots

Confirm that:

☒ The axis labels state the marker and fluorochrome used (e.g. CD4-FITC).

☒ The axis scales are clearly visible. Include numbers along axes only for bottom left plot of group (a 'group' is an analysis of identical markers).

☒ All plots are contour plots with outliers or pseudocolor plots.

☒ A numerical value for number of cells or percentage (with statistics) is provided.

## Methodology

| Sample preparation | PBMCs, T-cells (CD8+ T-cell and Jurkat76 cell lines), APCs, and other cell lines were collected, washed with FACS buffer, and stained with the corresponding antibodies as per manufacturer's instructions. Cells were washed twice with FACS buffer prior to analysis. |
|---|---|

| | |
|---|---|
| Instrument | Cells were analyzed at the UCSF Helen Diller Comprehensive Cancer Center using the ATTUNE2 or ATTUNE NxT Flow Cytometer. Flow sorting was performed on the BD Biosciences FACSAria Flow Cytometer. |
| Software | Data was collected using the BD Biosciences FACSDiva software v6.1.3 or the Attune Cytometric software. Analysis was performed using FlowJo v10.7.2. |
| Cell population abundance | At least 1e6 cells were collected for analysis and 5e4 cells were analyzed. |
| Gating strategy | Using the FSC/SSC gating, debris was removed, and the single alive cells were gated with the additional live/dead staining. Each population was gated based on the surface or intracellular markers as described in the manuscript. Details are provided in the main text and the extended data. |

☒ Tick this box to confirm that a figure exemplifying the gating strategy is provided in the Supplementary Information.

