## [Peer Review File · Nature]

Tumor-wide RNA splicing aberrations generate therapeutically actionable public neoantigens

Corresponding Author: Professor Hideho Okada

Version 0:

Reviewer comments:

Referee #1

(Remarks to the Author)

In the present work, Kwok and colleagues have established a robust pipeline for the identification of public non-canonical neoantigens arising from aberrant splicing, and more specifically neo-junctions (NEJ), across distinct tumor histotypes as well as across different spatial locations within a subset of tumors with available specimens. The Authors then proceeded to validate a subset of such neoantigens for: (i) expression at the proteome and immunoproteome level; and (ii) ability to elicit CD8 T cell responses from healthy donors. Finally, the Authors take a dive into possible molecular mechanisms underlying the generation of NEJ-derived neoantigens in GBM, suggesting an association with components of the splicing machinery in subsets of GBM patients.

Although the systematic analysis of non-canonical neoantigens arising from aberrant splicing is not novel, the analytic approach herein presented is very thorough and solid, representing a more compelling alternative to other similar pipelines already published. We also commend the Authors for the rigorous and comprehensive functional validation of the NEJ-derived neoantigens.

Our major criticisms are related to two main aspects:

- 1) The claims on intratumoral heterogeneity (ITH) are overstretched. We recognize the importance of identifying clonally- rather than subclonally-represented targets for immunotherapy; similarly, we acknowledge the value of sampling multiple spatially distinct specimens from the same tumor. However, it must be also recognized that comprehensive assessment of intratumor clonality is hampered by several limitations. Those include: i) inevitable biases in sampling and ii) limits in the (bulk) sequencing sensitivity. Besides these (inevitable) technical limitations, ITH is not the sole "therapeutic barrier" to finding ideal targets for immunotherapy. Indeed, even if spatially resolved, this in-depth analysis remains a snapshot of tumors taken at a single time point. As such, it does not take into consideration evolution across time, space (in the sense of primary vs. metastatic locations) and therapeutic bottlenecks.
- 2) The attempt at identifying mechanisms underlying the tumor-specific aberrant splicing in their present form are not sufficiently compelling. In the last figure of the manuscript the Authors try to infer molecular mechanisms associated with aberrant splicing by comparing and contrasting different GBM subtypes. The Authors show that siRNA knockdown of some defined splicing factors could impact the abundance of NEJ transcripts. Given the much-emphasized pan-cancer nature of the aberrant splicing events, it is hard to find exhaustive an association analysis performed only on GBM. Extension of the association studies between expression/mutational status of splicing factors to the same initial set of TCGA tumors would definitely strengthen the results.

Minor comments:

- "MHC" should be replaced throughout the manuscript with "HLA" as the study is focused on human samples.
- In line 190 the Authors incorrectly use the term "HLA haplotypes" instead of "HLA alleles", as it would be difficult to have more than one HLA-A allele on a single haplotype.

Referee #2

(Remarks to the Author)

In the manuscript entitled "Tumor-wide RNA splicing aberrations generate immunogenic public neoantigens", Kwok and colleagues posit that intratumoral heterogeneity is a major roadblock to immunotherapy-based cancer control and eradication in the clinic and that uncovering neoantigens that are ubiquitously present within tumors will offer a work around this problem. This hypothesis is reasonable and if successful, will offer new targets for future therapeutic development.

To identify neoantigens, the authors leverage publicly available large RNAseq datasets from TCGA to uncover aberrantly spliced transcripts that form cancer neoantigens (termed neojunctions) in multiple cancer types. Using an impressive array of bioinformatic tools in an extensive pipeline, they were able to sift through TCGA datasets and identify a short list of neojunctions that were homogeneously expressed in many cancers. Although the concept of neojunction is not new, uncovering targets that are broadly present in cancer is novel.

They then determined the degree of intratumoral heterogeneity by analyzing RNAseq data from spatially distinct core biopsies (multi-site sampling) for the presence of neojunction sequences in several cancer types. In the case of gliomas (GBM and LGG, IDH1 WT and mut), the authors utilized their own source material of 10 spatially resolved cores/tumor to determine the ubiquitousness of neojunction sequences. In doing so, the authors identified neojunctions that putatively represent public targets for all cancers.

Focusing on two candidates, the authors demonstrated that endogenous neojunction-derived peptides from these aberrantly spliced genes are produced in cells. Using in vitro systems, they then showed engagement of TCR in CD8+ with downstream T cell activation from these neojunction-derived neoantigen peptides when presented by MHC1 cells. Furthermore, the authors identified and isolated TCR clonotypes for these peptides. Finally, they demonstrated significant neojunction differences in GBM vs IDH1mut-A and IDH1mut-O tumors.

The work presented here is significant. The authors uncovered public ubiquitous neojunctions that can be leveraged for immunotherapies. However, given the lack of response for all immunotherapies applied to GBM so far, it is difficult to imagine how any neoantigen-directed immunotherapy using neojunctions as targets would be any different. Overall, this is an excellent study that can be improved by addressing the comments below.

Major Comments:

- 1- The authors should address whether the neojunctions identified in other studies are also observed in their datasets and whether they are more or less ubiquitously expressed than their 2 candidates (how many cancers? How extensive within same tumor ?etc).
- 2- The authors need to include data from cancer patients, preferably GBM patients, showing that they carry CD8+ T cells that are recognizing one or both RPL22 and GNAS neojunctions. I recognize that this is a tall order to fulfill but not having that data substantially diminish the impact of the study.
- 3- The authors mention that a 10-core sampling per tumor is "better" than 3 or 5 core sampling. However there's no metric provided to back up this claim. This needs to be addressed with statistical figures to determine what would the optimal number of core samples are necessary to capture ubiquitous expression in each tumor. Maybe using the 10 core datasets and analyzing less cores (eg. 7, 5, 3 etc), would one obtain the same results?
- 4- Additional GBM PDX cell lines from the Mayo panel need to be tested for primed cytotoxic CD8+ T cell killing.
- 5- Orthogonal output readouts of CD8+ T cell cytotoxicity activity on GBM115 and others need to be included (in addition to the automated cell counting) and the mechanism(s) by which the CD8+ T cells kill the GBM cells.
- 6- The last figure is interesting but also sort of out of place with the public, pan cancer topic of the entire manuscript. In Fig. 7, the authors focus on differences in expression of neojunctions in IDH1 wt GBM and IDH1 mut LGGs. This switch the focus of the manuscript from ubiquitous neojunctions to cancer specific ones.

Minor Comments:

- 1- Part of Figure 1 is repeated in Figure 2.

Referee #3

(Remarks to the Author)

In the work entitled "Tumor-wide RNA splicing aberrations generate immunogenic public neoantigens" Kwok et al studied the conservation of cancer specific splicing-derived neojunction across tumors and patients, with the aim of discovering public, tumor-wide, tumor specific antigens (TSAs). They developed a pipeline (mostly composed of previously described methods) identifying and characterizing neojunction from several different intra-tumor sites and regions, from a high number of patients with a variety of cancers. A sizable number of previously not annotated neojunctions was identified and a big proportion of them shown to be expressed as proteins. Some example neojunction-derived peptides were also shown to be processed endogenously and presented by the HLA class I. Experimental evidence that such peptides can be recognized by CD8 T cells and that in vitro generated CD8 TCR-T cells specific for two, public neojunction-derived TSAs can kill tumor

cell lines is also provided. This indicates a potential use in "off-the-shelf" immunotherapies. The manuscript is well written (with only small errors in figures and text), experiments well-conceived and the results very interesting, potentially providing a new class of relatively common TSAs for immune-oncological interventions. However, further clarification of some results presented in the manuscript, as well as showing real in-vivo immunogenicity is required before this study can be considered for publication.

Major, specific comments:

1. The authors start by identifying nonannotated junction reads across different cancers in TCGA data. When neojunctions present in more than 10% of the samples were considered and designated as public, an average of 373 neojunctions across all tumor types was found. Even if the increased presence of splicing neojunctions in tumors was previously reported and one can debate the 10% cutoff, the rate of 54% is high and very interesting. A detailed analysis of the junctions is showed in figure 2. According to a study by the PCAWG Transcriptome Core Group (which should be referenced in the manuscript, see below), 96% of somatic mutations have a negative effect on splicing and figure 2 shows the expected bias for loss versus gain of the 3' junction. Surprisingly, there is a lack of such bias at the 5' junction, which was not discussed by the authors. Is there an explanation for this apparent positive selection for 5' junction gain?

2. Next the authors studied the intratumor heterogeneity (ITH) of the neojunctions. A subset of them was found to be expressed across many/all biopsies and in a high proportion of patients, representing interesting candidates for therapeutic interventions. To this end, the authors focused on gliomas, which are known to have a particularly high ITH. The analysis of 10 maximally distanced biopsies from 56 tumors by whole-exom and RNA-seq showed neojunctions present across multiple patients. Perhaps not entirely surprising, more neojunctions were found in gliomas carrying IDH1 mutations, but they were particularly enriched in oligodendroglioma patients.

Still, quite some ITH was observed, so in general a combination of peptides derived from many neojunctions would be required for a therapy covering the whole tumor. Even if a proportion of the neojunctions was expressed in the entire tumor in at least one case, and some in at least two or more cases, the authors should provide calculations indicating the minimal and average numbers of neojunctions required to cover all the tumor cells, in a certain fraction of patients (e.g. 50%, 75%, 100%). This is important to judge the feasibility of personalized approaches. What were the most conserved and most prevalent neojunctions? What was the highest prevalence among patients? Some data is shown in figure 3F, but the colors are difficult to read and shown is only the fraction of cases with a particular neojunction. The fraction of patients with defined numbers of tumor-wide neojunctions, conserved high, mid and so on, should be also shown, and example numbers mentioned in the text. This would allow to see how many patients had at least one neojunctions present in the entire tumor, how many two, and how many had no tumor-wide neojunctions.

3. A big fraction of the public neojunctions expressed in gliomas was found at the mRNA level in cell line transcriptomic data, peptides derived from 38% of the public neojunctions were found by proteomics. However, protein expression of neojunctions was clearly lower than the canonical junctions (figure 4C) and should be mentioned as such.

Overall, 192(24%) of the neojunctions were detected by both mRNA and proteomics. Prediction algorithms were used to assess the proteasomal cleavage and presentation of tumor-wide neojunctions by MHC class I. 832 peptides were in the top 10% of both algorithms and were further analyzed. 81 were also in the 192 neojunctions previously characterized by mRNA and proteomics expression. The rest of the analysis was done with 32 HLA-A*02:01 binders. High intra-tumoral conservation was observed for most of these and several were chosen for further analysis.

These experiments are well done, however care should be taken that correct figures are referenced, as for example Figure 3G could not be found.

4. Next, the authors show that CD8 T cells can recognize neojunction-derived peptides, by testing 4 of the 32 HLA-A*02:01-binding candidates in a in-vitro sensitization assay with peptide-loaded DCs and INF γ readout. Reactivity against two of these peptides was found. 7 TCRs for NeoARPL22 and one 1 TCR for NeoAGNAS were discovered. The reactivity of one TCR per antigen was verified by cloning and transduction into a TCR-negative Jurkat cell line carrying TCR signaling reporters, and coculture with peptide pulsed APCs. Reactivity was observed at relatively low (1nM) peptide concentrations. Similar data was obtained by using PBMC-derived CD8 T cells and measurement of INF γ , TNF α and degranulation markers. MHC specificity was shown by antibody blocking experiments and endogenous peptide processing and presentation, by TCR recognition of peptides derived from transfected mRNA and HLA ligandomics.

These elegant experiments prove natural peptide processing and the TCR-epitope reactivity, but contrary to the authors conclusions, do not prove immunogenicity, nor high TCR affinity. Even if the higher immunogenicity is likely, because of the "foreigners" of the neojunction-derived peptides, it is dependent on many processes and can only be tested in-vivo. Showing that patient activated/memory T cells (preferably TILs) contain reactivities against such peptides by tetramer staining, ELISPOT or in vitro re-call, would show real immunogenicity and enhance the manuscript greatly. In-vitro experiments show only precursor frequency.

Furthermore, T cell responses in peptide titration experiments are dependent on two factors: the binding affinity of the peptide to the MHC, and the binding affinity of the TCR to the pMHC. Therefore, no conclusion about TCR affinity can be made and instead the term "functional avidity" should be used.

Still, these results clearly indicate the antigen specificity of the TCRs and their possible use for therapy. This aspect is highlighted by showing that TCR-T cells can kill glioma cells in-vitro. Together, the results show that the neojunction-derived peptides can be endogenously processed and presented by the tumor cells in sufficient amounts to allow in vitro TCR-T cell killing. However, even if neojunction-derived peptides are more commonly expressed in tumors, there is nothing particular about them which would make them more likely to be processed and presented by the tumor cells in general. Therefore, each of the neojunctions-derived peptides needs to be tested in a similar way like any other therapeutic candidate peptide. This should be mentioned in the manuscript.

5. Finally, to characterize splicing-related genes which would trigger higher neojunction generation, it was discovered that loss of SNRPD2 and SF3A3 may lead to the increased expression of neojunctions in IDH1 mutant gliomas. For example, almost 50% of the 789 neojunctions increased with decreased SNRPD2 expression.
Nice and clear experiments.

Minor issues

- The study "Genomic basis for RNA alterations in cancer" by PCAWG Transcriptome Core Group et al. should be mentioned in the manuscript (<https://doi.org/10.1038/s41586-020-1970-0>).
- Figure 3B has no yellow annotations.
- Figure 3G could not be found.
- Line 270: alanine scanning is done on epitopes not TCRs
- Flow cytometry data in Figure 6E appear to be missing.
- Text in many main and supplementary figures is way too small to be legible.
- Suppl Figure 6 is not legible so clustering of the genes cannot be assessed.

Referee #4

(Remarks to the Author)

The manuscript "Tumor-wide RNA splicing aberrations generate immunogenic public neoantigens" by Kwok et al. describes the identification of tumor-wide and public neoantigens by datamining different RNASeq sources to land in identifying aberrant splicing in GNAS and RPL22 and other mRNAs associated to IDHmut tumors. Interestingly, they present data that supports that these splicing changes (only presented for GNAS and RPL22) potentially act as neoantigens for cytotoxic T-cells and can be developed into cellular immunotherapies. In addition, the authors present preliminary data on potential splicing factors that may regulate splicing changes associated to IDHmut tumors. This manuscript has a big effort in in silico analyses across and within tumors, however, major experiments are needed to functionally validate the findings, as described below.

Major comments:

1. The thresholds that authors use to decide on relevant NEJ throughout the study should be justified. Also consider presenting more stringent criteria in the analyses in the first 3 figures. Seems that the number of hits are very high based on previous literature.
2. While the authors focus on NEJ/derived peptides that bind to HLA-A*02:01, would be important to present: 1) what NEJs/peptides were identified for other HLAs? Is the identification of NEJs related to differences in the prevalence of HLAs among the cohorts studied? If so, how is this related to prevalence of the HLA and the capacity of HLA binding and immunoeediting of tumors? Is the prevalence of HLA affecting skewing the NEJ predicting?
3. Re: Fig 4 comment should be made on IDH mut vs WT NEJs. Were the IDHmut NEJ found in other IDH mut tumors (i.e. LIHC IDHmut)? I know this is covered in Fig 7 but let the reader know that more on this finding is studied in Fig 7.
4. Consider bringing figure 7 before all immune studies. This will keep all the NEJ identification and mechanism together before shifting gears towards to immune-triggering response.
5. The authors go through detailed analyses of multiple data sets to land in 32 NEJs, and finally 2 functional NEJ (GNAS and RPL22), however, there is no detailed description on what are the exact NEJs for these genes (A3', A5', SE etc). Are these NEJs impacting the function of the encoded proteins? Producing functional/non-functional proteins?
6. Importantly, there is no PCR validation of the computational identified NEJs or WB of the encoded proteins and identified splicing factors between tumors and after manipulation with siRNA.
7. Would be nice if authors can use the database "MAJIQlopedia: an encyclopedia of RNA splicing variations in human tissues and cancer" (PMID: 37953365) to re-validate that GNAS, RPL22 and IDHmut-NEJs are cancer specific and potentially conserved across other tumors.
8. Re: Fig 7: The authors should also rule-in or rule-out the role of the listed splicing factors in GNAS and RPL22 NEJs.
9. Re: Fig 7: Only loss-of-function studies were performed and using only single -siRNAs. Orthogonal validation studies are required.
10. Can immune studies be performed on the IDHmut NEJs?

Minor comments:

11. What is public neoantigens? Best to define the first time used. You only understand as you read more into the manuscript.
12. Fig.3E is not clear how clustering is associated to IDH1 status.
13. Grammatical error in line 140.
14. NEJ acronym was not introduced and had to be deducted from text.
15. Re: Fig 4: A comment should be added on potential reason(s) why TCF12 NEJ did not elicit a response. Is the encoded antigen not strong enough?
16. Fig 6 has panels that are hard to interpret due to the small font-size.

Version 1:

Reviewer comments:

Referee #1

(Remarks to the Author):

I've gone through the author's answers to Reviewer#1's comments and they appear to me to be satisfactory. On both major comments, the authors performed a number of additional computational analyses and included the results (figures) in the manuscript.

On the heterogeneity and sampling, the authors recognized the technical limitations however, their 10+ biopsies performed is rather amazing. The bottom line is that one would never be able to get all targets when sampling is involved. In my opinion, the authors have adequately answered comment 1. Note that Extended Data Figure 2 doesn't contain red points in their graph as they pointed out.

The issue with comment #2 pertained to expanding the analysis to other tumor types and their subtypes to demonstrate an association of splicing factor mutations and NJs. The authors have expanded their analyses to now include other tumor types and show strong association.

Referee #2

(Remarks to the Author)

In this extensively revised version of their manuscript, Kwok and colleagues beautifully strengthened their study of public neojunctions by clarifying text and adding copious analyses. This current version of their work significantly solidifies their previous conclusions. The authors have addressed all of my comments exquisitely.

Referee #3

(Remarks to the Author)

The revised manuscript is substantially improved, and I appreciate the efforts that the authors have made. However, while the authors addressed most of my concerns, they did not really address the one concerning immunogenicity. Given that it is claimed in the title, convincing results showing in-vivo T cell responses should be provided.

In an attempt to show immunogenicity of the NEJ-derived peptides, the authors analyzed the frequency of peptide-specific CD8+ T cells in patient PBMCs (n=3) after in-vitro stimulation with peptide-loaded APCs (IVS). They see positive cells in one patient only, which could represent de-novo priming, especially since the description of the IVS in the methods mentions the isolation of naive T cells. To control for de-novo priming the authors provide results from PBMCs of one healthy donor. Given the low frequency of responses in patients (1/3), this is way too little.

As mentioned in the original review, I see two possibilities to address this:

1. (preferred) by showing that in-vivo activated or memory T cells (preferably TILs) from patients contain reactivities against NEJ-derived peptides by tetramer staining, ELISPOT, in vitro re-call etc., or
2. by IVS experiments showing a clear, statistically significant difference, between a substantial number of patients and healthy donors.

Unfortunately, neither was provided so far.

I am still supportive of the publication, but to claim immunogenicity, more patients and controls must be analyzed and/or ex-vivo activation/phenotypic analysis needs to be provided.

Minor issue:

The gating of the flow cytometry data in Extended Data Figure 8B looks strange for the APC-A parameter (it cuts the whole population in half). Was the gate set according to a FMO ("fluorescence minus one") control? If yes, it should be shown, if not, it should be done, and the data re-analyzed.

Referee #5

(Remarks to the Author)

Comments on revised manuscript Kwok et al.

Comments: After reading the revised manuscript, the comments of Referee 4 and the authors' rebuttal letter we conclude that two major points that were raised by Referee 4, as they pertain to splicing, were not addressed satisfactorily. We feel that these are major points and need to be addressed if the authors want to claim that there is involvement of splicing factors in generating neojunctions.

1. The authors do not provide data validating that the splicing events creating neojunctions are regulated by knockdown of the splicing factors. Referee 4 point 6 asked for PCR validation of the computational identified NEJs or WB of the encoded proteins and identified splicing factors between tumors and after manipulation with siRNA. The authors respond in their rebuttal letter "We have performed PCR validation of these NEJs in Supplementary Figure 4H of the original manuscript via amplicon sequencing, which can now be found in Extended Data Figure 6J of the revised manuscript." Extended Data Figure 6J shows "Read frequency of reads spanning neojunctions in RPL22 and GNAS compared to the canonical junction spanning reads in glioma cell lines (n=1)". This does not validate the splicing events pre and post knockdown of splicing factors.

The authors also provide Extended Figure 5A, B. Extended Figure 5A, B shows "RNA-sequencing-derived (A) and qPCR (B) TPM expression of CELF2, SNRPD2, and SF3A3 following siRNA knockdown." This is validation of splicing factor expression from pre and post treated cell lines, but the authors do not supply PCR validation of the computational identified NEJs. The authors respond in their rebuttal letter that performing WB to validate NEJ-encoded proteins is not feasible. They should make some effort to validate by PCR.

2. We agree with Referee 4's request for orthogonal validation studies (Referee 4 point 9). The authors should provide this either by experiments with overexpression of the splicing factors or rescue experiments. Extended Figure 5C, D does not satisfactorily address Referee 4 point 9. The authors supply RNA-seq data analysis of further siRNA knockdown experiments. From the rebuttal letter: "RNA-sequencing analysis demonstrated an associated decrease in the expression of 19 (8.60%) and 28 (12.67%) IDH 1mut-specific neojunctions, respectively, in SF10417 and SF10602 when treated with CELF2 siRNA compared to their non-treated controls (now included in Extended Data Figure 6C of the revised manuscript)." (I assume that the authors are referring to Extended Data Figure 5C of the revised manuscript). Again, the authors do not validate any specific neojunction (a similar comment to Referee 4 point 6). An example of orthogonal validation would be knockout by a different method than siRNA, such as CRISPR knockout, or overexpression or rescue experiments.

Other comments:

- Line 169-170 Figure 3J "... and observed a trend of decreased expression of the associated neojunction across both lines (Figure 3J). The graphs in Figure 3J show that this decrease is non-significant.

- Line 191-192 "Similarly, with increasing levels of SF3A3 expression, 178 (22.6%) neojunctions tended to increase in expression, and 127 (16.1%) neojunctions tended to decrease in expression." This is in direct conflict with the conclusion of the next sentence and what is shown in the figure "Notably, siRNA knockdown of either SNRPD2 or SF3A3 in the GBM115 cell line (Extended Data Figures 5A-5B) led to a significant increase in the expression levels of their associated neojunctions". This statement needs to be changed.

- Line 192-195 "Notably, siRNA knockdown of either SNRPD2 or SF3A3 in the GBM115 cell line (Extended Data Figures 5A-5B) ..." The fourth panel of Extended Figure 5B shows an increase in SF3A3 expression with siSF3A3, where one would expect a decrease.

- It is not clear why the authors chose to validate neojunction ACAP2 (Figure 3J, 3L) and not one of the neojunctions that are the focus of the manuscript (NEJ-RPL22 and NEJ-GNAS). The figure legend for Figure 3L reads: "Expression of NJACAP2 in GBM115 treated with control siRNA or siSN RPD2 (left) or siSF3A3 (right)". The Y axis of the right panel of Figure 3L is labelled PEA15NJ, and not ACAP2NJ.

- In response to Referee 4 point 9 the authors write "Similarly, we were able to observe an overall decrease in in NJACAP2 expression in 1 of 2 mutant IDH1 glioma cell lines, which corresponds to the results in the original manuscript's Figure 7J. These additional findings shown below are included in Extended Data Figure 6E of the revised manuscript." (I assume the authors are referring to Extended Figure 5E in the revised manuscript.) Also, it is not clear what is the left-hand panel in the rebuttal letter referring to (it is not in the revised manuscript).

- Line 233-234 "When considering both RNA-seq and MS confirmation of glioma-specific neojunctions, we validated the presence of 192 (24.3%) public neojunctions expressed across all patient-derived samples (Figure 4D)." Figure 4D is a schematic demonstrating the selection of high-confidence neojunctions for downstream analysis and not validation.

- Line 267-269 "When ITH of these 32 neojunctions was investigated in the data set from spatially mapped samples, high intratumoral conservation was observed for most of these NEJs, particularly the two nucleotide A3 loss-encoding neojunction located within GNAS (NJGNAS) (Figure 4M)." The authors do not comment on the NEJ-RPL22, one of the NEJs that is the focus of the manuscript.

- Line 378 -379 "The higher expression level of the canonical GNAS allele over RPL22 may contribute to the prevalence of NJGNAS detected across all analyses." Not clear what is meant by higher expression of GNAS allele over RPL22?

- Figure legend for Extended figure 6N is missing.

Version 2:

Reviewer comments:

Referee #3

(Remarks to the Author)

I thank the authors for carefully considering my comments. The issue related to the FACS analysis was addressed satisfactorily. However, the issue of immunogenicity is still controversial. I fully appreciate the reasoning of the authors and agree that a thorough analysis of true in-vivo immunogenicity is very difficult due to the complex biology, and often not possible due to practical constraints. However, if something cannot be convincingly shown, it cannot be claimed. Therefore, considering all the above, and following the authors' response to my comment on page 2 "... the central conclusion of our manuscript – that a subset of recurrent NEJs undergo physiologic processing and H LA presentation to generate therapeutically actionable shared neoantigens – remains unchanged", I suggest changing the title of the manuscript to "Tumor-wide RNA splicing aberrations generate therapeutically actionable shared neoantigens" or similar. In this case, immunogenicity is not claimed as the main finding, and the interesting results of this study can be made available to the public.

Furthermore, following the same logic, I would avoid the use of the word "immunogenic" in the manuscript as much as possible and suggest the following changes:

1. Delete "immunogenic" from the sentence in line 61: "We identified /immunogenic/ neojunction-derived TSAs that were proteolytically-processed and presented on a prevalent human leukocyte antigen (H LA) molecule."
2. Change "immunogenicity" to "responses" in line 274: "We next sought to determine whether NEJ-derived neopeptides can drive T-cell /immunogenicity/ responses."
3. Delete "immunogenicity" from the sentence in line 281: "...CD8+ conditions revealed neoantigen-reactivity /immunogenicity/ in two out of four..."
4. Change "immunogenicity" to "reactivity" in line 274: "allowed the demonstration of neoantigen-specific /immunogenicity/ reactivity and tumor-specific killing"
5. Delete "immunogenic" from the sentence in line 988: "/Immunogenic/ Cytokine assays were performed..."
6. Change "immunogenicity" to "reactivity" in line 1064: "maintained for downstream co-culture and /immunogenicity/ reactivity assays"; and line 1191: "In dose-dependent /immunogenicity/ reactivity assays,"

Taken together, provided these changes are made, I support the publication of this manuscript.

Referee #5

(Remarks to the Author)

The second revision of the manuscript has positively addressed our comments raised previously. The authors have addressed our concern on the need for orthogonal validation studies to assess the relationship between the splicing factors and NJ expression. They have now performed CRISPR-mediated knockout of the splicing factors to determine its effect on the expression of the different NJs (Fig. 3J, L) and moved the siRNA knockout to the extended data (Extended Data 5C-5D). However, the effect on the expression of NJACAP2 upon CELF2 knockout with CRISPR is not very strong as with the other splicing factors. This is of particular importance as the authors state that 'NJACAP2 was most positively correlated with the expression of CELF2'. However, in cell line SF10417, where the siRNA mediated knockout of CELF2 was quite strong, the effect on NJACAP2 expression was not significant (Extended Data Fig. 5C-5D). While in the same cell line where CRISPR-mediated knockdown of CELF2 was approximately 50% (Extended Data 5A), the effect on NJACAP2 expression was significant for guide 2 (Fig. 3J). Can the authors provide an explanation for this? The data for the other splicing factors is very significant and promising.

Owing to our comments "It is not clear why the authors chose to validate neojunction ACAP2 (Figure 3J, 3L) and not one of the neojunctions that are the focus of the manuscript (NEJ-RPL22 and NEJ-GNAS)", the authors have provided clarity on the matter by stating that the manuscript has a mechanism aspect and an immunotherapeutic aspect. Although this is now clearer, we feel that there is a lack of continuity between the two sections of the manuscript as no clear evidence on the immunotherapeutic effects of NJACAP2 and NJPEA15 was presented. The clinical aspect of the manuscript titled 'Tumor-wide RNA splicing aberrations generate immunogenic public neoantigens' may be confusing to the readers.

The authors satisfactorily responded to all the other comments of this reviewer

Dear Editor and Reviewers:

We appreciated the reviewers' constructive comments on our revised manuscript "Tumor-wide RNA splicing aberrations generate immunogenic public neoantigens." We have addressed all of the reviewers' comments and responded in a point-to-point manner below.

Referees' comments:

Referee #1 (Remarks to the Author):

In the present work, Kwok and colleagues have established a robust pipeline for the identification of public non-canonical neoantigens arising from aberrant splicing, and more specifically neo-junctions (NEJ), across distinct tumor histotypes as well as across different spatial locations within a subset of tumors with available specimens. The Authors then proceeded to validate a subset of such neoantigens for: (i) expression at the proteome and immunoproteome level; and (ii) ability to elicit CD8 T cell responses from healthy donors. Finally, the Authors take a dive into possible molecular mechanisms underlying the generation of NEJ-derived neoantigens in GBM, suggesting an association with components of the splicing machinery in subsets of GBM patients.

Although the systematic analysis of non-canonical neoantigens arising from aberrant splicing is not novel, the analytic approach herein presented is very thorough and solid, representing a more compelling alternative to other similar pipelines already published. We also commend the Authors for the rigorous and comprehensive functional validation of the NEJ-derived neoantigens.

Our major criticisms are related to two main aspects:

1) The claims on intratumoral heterogeneity (ITH) are overstretched. We recognize the importance of identifying clonally- rather than subclonally-represented targets for immunotherapy; similarly, we acknowledge the value of sampling multiple spatially distinct specimens from the same tumor. However, it must be also recognized that comprehensive assessment of intratumor clonality is hampered by several limitations. Those include: i) inevitable biases in sampling and ii) limits in the (bulk) sequencing sensitivity. Besides these (inevitable) technical limitations, ITH is not the sole "therapeutic barrier" to finding ideal targets for immunotherapy. Indeed, even if spatially resolved, this in-depth analysis remains a snapshot of tumors taken at a single time point. As such, it does not take into consideration evolution across time, space (in the sense of primary vs. metastatic locations) and therapeutic bottlenecks.

We thank the reviewer for their insightful comments, and we have correspondingly looked into the evolution of neojunction expression over space, time, and therapeutic bottlenecks and added our findings into **Extended Data Figure 2C, 2I-2K**. The reviewer has additionally provided an insightful comment on the characterization of "ideal" immunotherapy targets and the importance of scaling back emphasis on intratumor heterogeneity as the sole therapeutic barrier. As such, we updated our text in the revision to reflect that intratumor heterogeneity is one of multiple important factors to consider when evaluating a neoantigen's candidacy as an effective therapeutic target in **Lines 93-103**. Regarding the inevitable biases in biopsies, we hoped to minimize this challenge in our multi-core sampling approach where all 10+ biopsied regions of gliomas were maximally-distanced based on spatial mapping. We note that this is well beyond the standard in the field of single samples without spatial information. Due to the large sum of samples collected from this extensive approach, it was not feasible to conduct single-cell sequencing analyses on all 500+ samples; however, we agree with the reviewer that higher cell resolution beyond bulk sequencing would bring an additional level of sensitivity to the study. We note that an important technical limitation of single cell analyses is the sparse data (very few reads per transcript) that prohibits the detection of the vast majority of individual NJs, leading to unacceptably high false negative rates.

We appreciate the reviewer's recognition of our study's capability to spatially resolve neojunction expression across various cancers. We agree that while in-depth, our initial analyses did not consider heterogeneity

across time (primary vs. recurrence) and space (primary vs. metastatic). To address this feedback, we first re-analyzed the pan-cancer TCGA RNA-sequencing data to characterize neojunctions in paired samples obtained from primary and recurrent cancers. Within the subset of cancers that we investigated, matching primary and recurrence data could be identified in COAD ($n=1$), GBM ($n=3$), LGG ($n=12$), LIHC ($n=2$), and LUAD ($n=2$). A substantial proportion of neojunctions persisted in many matched pairs (medians across these cancer types ranged between 32.5% and 57.6%), and we now report this finding in **Extended Data Figure 2J**. We characterized a subset of neojunctions that were found tumor-wide across all spatially-mapped samples and conserved from primary to recurrence in LIHC (3 out of 173 (1.7%) neojunctions) and COAD (7 out of 265 (2.4%) neojunctions).

In further consideration of the reviewer's suggestion to investigate neojunction persistence between primary and metastases, there are no such brain tumor metastases to evaluate, and thus we cannot assess this perspective in LGG and GBM. Instead, we analyzed the expression of neojunctions across TCGA tumor types that had patient-matched primary and metastasis samples: PRAD ($n=1$), COAD ($n=1$) and SKCM ($n=2$). Although the number of cases is limited, we observed a substantial number of neojunctions conserved between primary and metastatic samples (medians across all investigated cancer types range between 43.8% and 72.5%), and we report this finding in **Extended Data Figure 2I**.

Given the modest number of paired samples available in TCGA, we extended our pan-cancer analyses of SKCM primary vs. metastasis using RNA-sequencing data from the PEACE multi-region sampling study of SKCM metastatic sites conducted by Spain L. et al. (*Cancer Discovery*, 2023). Raw.fastq files from this study were analyzed using our SSNIP pipeline. We initially identified 353 putative neojunctions expressed in TCGA SKCM RNA-sequencing data. The inclusion of SKCM neojunctions in our pan-cancer study was included in **Figure 1** and **Extended Data Figures 1B-1F** of the revised manuscript. We detected the expression of 135

(38.2%) in the PEACE multi-region study, and notably, found 13 (9.6%) neojunctions that were detected in every metastatic site in at least one patient. This exciting addition to our pan-cancer analysis further illustrates the presence of neojunctions that are ubiquitously expressed across metastases and the primary tumor. (Extended Data Figure 2C)

Finally, we sought to address the reviewer's question regarding the conservation of neojunctions following therapeutic bottlenecks. To that end, we investigated patient-matched primary and recurrent samples following temozolomide treatment in our glioma dataset. We observed respective averages of 79.2% and 82.3% neojunctions conserved in hypermutated and non-hypermutated glioma recurrences. Interestingly, while the large majority of neojunctions persisted from primary to the first recurrence, only an average of 16.2% of neojunctions identified at primary were identified at the second recurrence, indicated by red points in the following plot (Extended Data Figure 2). Altogether, these findings demonstrate the persistence of neojunctions across space, time, and therapeutic bottlenecks. These figures have now been added to Extended Data Figure 2.

2) The attempt at identifying mechanisms underlying the tumor-specific aberrant splicing in their present form are not sufficiently compelling. In the last figure of the manuscript the Authors try to infer molecular mechanisms associated with aberrant splicing by comparing and contrasting different GBM subtypes. The Authors show that siRNA knockdown of some defined splicing factors could impact the abundance of NEJ transcripts. Given the much-emphasized pan-cancer nature of the aberrant splicing events, it is hard to find exhaustive an association analysis performed only on GBM. Extension of the

association studies between expression/mutational status of splicing factors to the same initial set of TCGA tumors would definitely strengthen the results.

We thank the reviewers for their comment and agree that performing differential splicing gene expression analysis of the remaining tumor types could suggest a shared pan-cancer mechanism driving neojunction expression. To address their question in our revised manuscript, we utilized TCGA-associated publications for each tumor type to attain the proper molecular or histological subtypes within each cancer as follows:

- KICH¹ was classified as either Eosinophilic or Classic.
- KIRP² was classified by cluster-of-cluster analysis as C1, C2a, C2b, or C2cCIMP.
- LIHC³ was classified by iCluster as C1, C2, or C3.
- LUAD⁴ was classified by iCluster as C1, C2, C3, C4, C5, or C6.
- PRAD⁵ was classified by mutations/fusion in ERG, ETV1, ETV4, FLI1, SPOP, FOXA1, IDH1, other.
- SKCM⁶ was classified by mutations in BRAF, RAS, NF1, or by triple wild-type.

Similar to our analysis between glioma subtypes, we evaluated the expression of neojunctions across each tumor subtype and performed independent Wilcoxon Rank Sum tests between each subtype. The iCluster C3 and C6 subtype in LIHC and LUAD, respectively, demonstrated significantly different levels of neojunction expression (p -value < 0.05) when compared to all other subtypes within the same cancer. In our revised manuscript, we include this analysis as **Figures 3M-3N and Extended Data Figures 5F-5I** shown below and updated the text with the inclusion of **Lines 203-212**.

We next performed differential gene expression (DESeq2) and gene set enrichment analysis (GSEA) on these subtypes against the other cancer-specific subtypes. While no significant changes in the expression of splicing-related gene sets were observed between LIHC iCluster1 and iCluster3, 23 splicing-related gene sets were shown to be significantly upregulated (p -value < 0.05, NES > 1.5) in iCluster3 relative to iCluster2. Notably, GSEA of LUAD iCluster6 demonstrated that this subtype had significantly downregulated expression (p -value < 0.05, NES < -1.5) of splicing-related gene sets when compared to all other five subtypes: iCluster1 ($n=35$), iCluster2 ($n=34$), iCluster3 ($n=56$), iCluster4 ($n=40$), iCluster5 ($n=27$). Twenty-three splicing-related genes were downregulated across all comparisons. These findings suggest that dysregulated expression of splicing-related

genes is associated with differential neojunction expression across different cancer subtypes, supporting the hypothesis of a pan-cancer mechanism underlying aberrant splicing. We have incorporated these additional analyses into our revised manuscript to strengthen our results (**Supplementary Table 2**).

Minor comments:

- “MHC” should be replaced throughout the manuscript with “HLA” as the study is focused on human samples.

We thank the Reviewer for noting that our study focused exclusively on human samples. All instances of “MHC” are now replaced with “HLA”.

- In line 190 the Authors incorrectly use the term “HLA haplotypes” instead of “HLA alleles”, as it would be difficult to have more than one HLA-A allele on a single haplotype.

We thank the Reviewer for their and now use “HLA alleles” in place of “HLA haplotypes”.

Referee #2 (Remarks to the Author):

Comments to Authors:

In the manuscript entitled “Tumor-wide RNA splicing aberrations generate immunogenic public neoantigens”, Kwok and colleagues posit that intratumoral heterogeneity is a major roadblock to immunotherapy-based cancer control and eradication in the clinic and that uncovering neoantigens that are ubiquitously present within tumors will offer a work around this problem. This hypothesis is reasonable and if successful, will offer new targets for future therapeutic development.

To identify neoantigens, the authors leverage publicly available large RNAseq datasets from TCGA to uncover aberrantly spliced transcripts that form cancer neoantigens (termed neojunctions) in multiple cancer types. Using an impressive array of bioinformatic tools in an extensive pipeline, they were able to sift through TCGA datasets and identify a short list of neojunctions that were homogeneously expressed in many cancers. Although the concept of neojunction is not new, uncovering targets that are broadly present in cancer is novel.

They then determined the degree of intratumoral heterogeneity by analyzing RNAseq data from spatially distinct core biopsies (multi-site sampling) for the presence of neojunction sequences in several cancer types. In the case of gliomas (GBM and LGG, IDH1 WT and mut), the authors utilized their own source material of 10 spatially resolved cores/tumor to determine the ubiquitousness of neojunction sequences. In doing so, the authors identified neojunctions that putatively represent public targets for all cancers.

Focusing on two candidates, the authors demonstrated that endogenous neojunction-derived peptides from these aberrantly spliced genes are produced in cells. Using in vitro systems, they then showed engagement of TCR in CD8+ with downstream T cell activation from these neojunction-derived neoantigen peptides when presented by MHC1 cells. Furthermore, the authors identified and isolated TCR clonotypes for these peptides. Finally, they demonstrated significant neojunctions differences in GBM vs IDH1mut-A and IDH1mut-O tumors.

The work presented here is significant. The authors uncovered public ubiquitous neojunctions that can be leveraged for immunotherapies. However, given the lack of response for all immunotherapies applied to GBM so far, it is difficult to imagine how any neoantigen-directed immunotherapy using neojunctions as targets would be any different. Overall, this is an excellent study that can be improved by addressing the comments below.

Major Comments:

1- The authors should address whether the neojunctions identified in other studies are also observed

in their datasets and whether they are more or less ubiquitously expressed than their 2 candidates (how many cancers? How extensive within same tumor ? etc).

We thank the reviewer for their suggestion. At their recommendation, we have investigated cancer-specific RNA splicing-derived neoantigens described in three recent and relevant publications (PMID: 33811047, PMID: 37192158, PMID: 37953365). A major limitation of using these studies in addressing this comment, however, is that the specific splicing junction sites that generate the corresponding tested neoantigens were not disclosed. This made it difficult to determine whether their validated neojunctions were ubiquitously expressed in our datasets. Bigot et al. (2021, *Cancer Discovery*) provided a list of unique neoantigen n-mer candidates that were identified, but without the nucleic acid sequence and the splice site, it was not feasible to map back the final peptide candidate to the neojunction. Similarly, since the neojunctions in all three studies are barcoded, we are unable to evaluate whether the characterized neojunctions in their studies are more or less ubiquitously expressed than our candidates.

Nevertheless, to address this critique point, we performed the inverse study to determine whether the neojunctions identified in our study could be identified in the neojunctions identified in the remaining two studies. It is important to note that appreciable discrepancies will manifest in the early junction calling steps across different pipelines based on various variables (e.g. STAR versions, gene annotation files, etc.). Nevertheless, we still observed a significant proportion of neojunction identified in our pipeline detected within those of others. On average, we observed 14.6% and 3.9% of the neojunctions we identified through the SSNIP platform detected in IRIS (left, PMID: 37192158) and MAJIQlopedia (right, PMID: 37953365), respectively. We have included these new results into the new manuscript as **Extended Data Figures 1E-1F**. These findings demonstrate that while shared neojunctions can be identified between pipelines, our pipeline has captured unique neojunctions that are demonstrated to be immunogenic and ubiquitously expressed.

2- The authors need to include data from cancer patients, preferably GBM patients, showing that they carry CD8+ T cells that are recognizing one or both RPL22 and GNAS neojunctions. I recognize that this is a tall order to fulfill but not having that data substantially diminish the impact of the study.

We thank the reviewer for their insightful comment. At their request, we performed immune monitoring studies on archived PBMC samples obtained from $n=3$ HLA-A*02:01⁺ glioma patients featured in our initial submission. To detect rare circulating T cell populations, we performed in vitro stimulation (IVS) using HLA-A*02:01⁺ dendritic cells pulsed with the mass-spec identified NeoA_{GNAS} peptide. As a control for de novo priming under these conditions, we contemporaneously performed IVS using PBMC from an HLA-A*02:01⁺ healthy donor. Immune monitoring was performed using dual dextramer staining to enhance the sensitivity of detecting rare circulating T cell populations. We detected an immunogenic response in 1 of 3 glioma patients but not the HLA-A*02:01⁺ healthy donor control. The reactive T cell population was specific for the NeoA_{GNAS} epitope as they did not bind to dextramers loaded with an irrelevant HLA-A*02-restricted 9-mer neoantigen control. These findings further support the immunogenicity and potential clinical application of targeting NEJ-derived neoantigens and are now included in our revised manuscript as **Figure 5J**. The updated text included in this revised manuscript can be found in **Lines 326-331**.

3- The authors mention that a 10-core sampling per tumor is “better” than 3 or 5 core sampling. However there’s no metric provided to back up this claim. This needs to be addressed with statistical figures to determine what would the optimal number of core samples are necessary to capture ubiquitous expression in each tumor. Maybe using the 10 core datasets and analyzing less cores (eg. 7, 5, 3 etc), would one obtain the same results?

We agree with the reviewer that a statistical metric would be beneficial in capturing the importance of increasing the number of core samples in the accurate characterization of neojunction intratumor heterogeneity. As such, we performed further extensive analyses on our spatially-mapped cohort to determine whether more core samplings help capture ubiquitous expression in tumors. By reinterpreting our spatially-mapped GBM/LGG RNA-seq data, we can demonstrate that selectively fewer neojunctions are found to be expressed across progressively greater spatially-mapped samples per tumor (**Extended Data Figure 2F-2G**). Furthermore, we performed paired t-tests between the number of neojunctions expressed between 10-core sampling and 1- through 9-core sampling and demonstrate that the number of neojunctions found across all 10 samples is significantly less than those found in fewer core samples (**Extended Data Figure 2H**). The significance in the difference of neojunction expression between samples exponentially diminishes when approaching 10 samples. Of course, increasing the number of intratumor samples will undoubtedly strengthen the confidence, but we believe that 10 samples in this study is reasonable in demonstrating the importance of increased sampling compared to the current standard of analysis. This finding further solidifies the novelty of our study in underscoring the importance of characterizing intratumor heterogeneity of neojunctions using larger numbers of spatially distanced biopsies.

4- Additional GBM PDX cell lines from the Mayo panel need to be tested for primed cytotoxic CD8+ T cell killing.

To further demonstrate the potential broad applicability of the shared NJ we identified and at the recommendation of the reviewer, we expanded our cytotoxicity assays to include an additional Mayo Clinic GBM PDX cell line (GBM102). In addition, we have also included a panel of melanoma cell lines (WM-266-4 and RPMI-7951). Like our cell killing assays with GBM115, we experimentally measured recognition of the GNAS NJ-derived neoantigen in these additional cell lines. We have included these xCELLigence output data in **Extended Data Figure 8A** of the revised manuscript.

We performed an additional experiment to validate that tumor cell killing occurs in an HLA-A2 dependent manner. We utilized a Mayo Clinic GBM cell line with detectable expression of NEJ_{GNAS} but lacking the expression of HLA-A2 (GBM39) and either left this cell line untreated or transduced with HLA-A2. This allowed us to have a non-HLA-A2-expressing and HLA-A2-expressing version of the same cell line. By performing xCELLigence on a co-culture of non-transduced and TCR-transduced CD8⁺ T-cells against both of these cell lines, we demonstrate robust killing of the NEJ_{GNAS}-derived neoantigen in only in the GBM39 cell line with HLA-A2 transduced. We have now added these new findings as **Figure 6H** in the revised manuscript. This experiment provides orthogonal validation of results using a pan-HLA blocking antibody and demonstrate that tumor-killing reported in our original manuscript is mediated by HLA-A2 presentation of the neoantigen.

5- Orthogonal output readouts of CD8+ T cell cytotoxicity activity on GBM115 and others need to be included (in addition to the automated cell counting) and the mechanism(s) by which the CD8+ T cells kill the GBM cells.

In direct response to the Reviewer's query, we performed orthogonal assays to validate neoantigen-specific immune-recognition and cytotoxicity. CD107a and CD137 (also known as 4-1BB) are markers of T cell degranulation and activation, respectively. CD107a is expressed on CD8+ T-cell surfaces when cytotoxic granules are released which indicates that the corresponding T-cell has been activated and engaged in cytotoxic response.⁷ CD137 is a costimulatory receptor that is selectively expressed on the surface of T-cells following TCR ligation. Engagement of CD137 enhances the proliferation, cytokine production, and cytotoxic activity of T cells.⁸ Similar to the experiments above, we performed a co-culture of tumor cells with either non-transduced or TCR-transduced CD8+ T-cells at a 1:1 E:T ratio for 24 hours. We then collected the CD8+ T-cells from the co-culture and measured the surface expression of both CD107a and CD137 by flow cytometry analysis. As shown below, we now demonstrate that both CD107a and CD137 surface expression is significantly increased when TCR-transduced CD8+ T-cells are co-cultured with tumor cells compared to non-transduced CD8+ T-cells under the same conditions. We included this data into our revised manuscript's **Figure 6I** and **Extended Data Figure 8B** and incorporated the corresponding text in **Lines 367-369**.

6- The last figure is interesting but also sort of out of place with the public, pan cancer topic of the entire manuscript. In Fig. 7, the authors focus on differences in expression of neojunctions in IDH1 wt GBM and IDH1 mut LGGs. This switch the focus of the manuscript from ubiquitous neojunctions to cancer specific ones.

We appreciate the reviewer's advice regarding the narrative flow of the figures and addressed this by shifting the order of which the figures are presented in the manuscript. We moved Figure 7 of the original manuscript to the present **Figure 3** in the revision so that now the mechanistic studies are introduced after the discussion of NEJ intratumor heterogeneity but before the immune studies. Additionally, we moved all mechanism-related supplementary data figures earlier in the revised manuscript to **Extended Data Figures 3-6**. We believe that this change significantly improves the readability and logic of the manuscript.

To broaden the scope beyond neojunctions differentially expressed between IDH1wt and IDH1mut gliomas, we first utilized TCGA-associated publications for each tumor type to attain the proper molecular or histological subtypes within each cancer:

- KICH¹ was classified as either Eosinophilic or Classic.
- KIRP² was classified by cluster-of-cluster analysis as C1, C2a, C2b, or C2cCIMP.
- LIHC³ was classified by iCluster as C1, C2, or C3.
- LUAD⁴ was classified by iCluster as C1, C2, C3, C4, C5, or C6.
- PRAD⁵ was classified by mutations/fusion in ERG, ETV1, ETV4, FLI1, SPOP, FOXA1, IDH1, other.
- SKCM⁶ was classified by mutations in BRAF, RAS, NF1, or by triple wild-type.

Similar to our glioma subtype analyses, we evaluated the expression of neojunctions across each tumor subtype and performed independent Wilcoxon Rank Sum tests between each subtype. The iCluster C3 and C6 subtype in LIHC and LUAD, respectively, demonstrated significantly different levels of neojunction expression (p -value < 0.05) when compared to all other subtypes within the same cancer.

We then performed differential gene expression (DESeq2) and gene set enrichment analysis (GSEA) on these subtypes against the other cancer-specific subtypes. While no significant changes in the expression of splicing-related gene sets were observed between LIHC iCluster1 and iCluster3, 23 splicing-related gene sets were shown to be significantly upregulated (p -value < 0.05, NES > 1.5) in iCluster3 relative to iCluster2. Notably, GSEA of LUAD iCluster6 demonstrated that this subtype consistently had significantly downregulated expression (p -value < 0.05, NES < -1.5) of splicing-related gene sets when compared to all other five subtypes: iCluster1 ($n=35$), iCluster2 ($n=34$), iCluster3 ($n=56$), iCluster4 ($n=40$), iCluster5 ($n=27$). Twenty-three splicing-related genes were consistently downregulated across all comparisons. These findings suggest that dysregulated expression of splicing-related genes is associated with differential neoantigen expression across different cancer subtypes, supporting the hypothesis of a pan-cancer mechanism underlying aberrant splicing. We have incorporated these additional analyses into our revised manuscript to strengthen our results (Supplementary Table 2).

Minor Comments:

1- Part of Figure 1 is repeated in Figure 2.

To address the repeated part of the figure Figure 1 in the original manuscript's Figure 2A we moved Figure 1 to Extended Data Figure 1A and removed Figure 2A.

Referee #3 (Remarks to the Author):

Remarks to authors

In the work entitled "Tumor-wide RNA splicing aberrations generate immunogenic public neoantigens" Kwok et al studied the conservation of cancer specific splicing-derived neoantigen across tumors and patients, with the aim of discovering public, tumor-wide, tumor specific antigens (TSAs). They developed a pipeline (mostly composed of previously described methods) identifying and characterizing neoantigen from several different

intra-tumor sites and regions, from a high number of patients with a variety of cancers. A sizable number of previously not annotated neojunctions was identified and a big proportion of them shown to be expressed as proteins. Some example neojunction-derived peptides were also shown to be processed endogenously and presented by the HLA class I. Experimental evidence that such peptides can be recognized by CD8 T cells and that in vitro generated CD8 TCR-T cells specific for two, public neojunction-derived TSAs can kill tumor cell lines is also provided. This indicates a potential use in “off-the-shelf” immunotherapies. The manuscript is well written (with only small errors in figures and text), experiments well-conceived and the results very interesting, potentially providing a new class of relatively common TSAs for immune-oncological interventions. However, further clarification of some results presented in the manuscript, as well as showing real in-vivo immunogenicity is required before this study can be considered for publication.

Major, specific comments:

1. The authors start by identifying nonannotated junction reads across different cancers in TCGA data. When neojunctions present in more than 10% of the samples were considered and designated as public, an average of 373 neojunctions across all tumor types was found. Even if the increased presence of splicing neojunctions in tumors was previously reported and one can debate the 10% cutoff, the rate of 54% is high and very interesting. A detailed analysis of the junctions is showed in figure 2. According to a study by the PCAWG Transcriptome Core Group (which should be referenced in the manuscript, see below), 96% of somatic mutations have a negative effect on splicing and figure 2 shows the expected bias for loss versus gain of the 3' junction. Surprisingly, there is a lack of such bias at the 5' junction, which was not discussed by the authors. Is there an explanation for this apparent positive selection for 5' junction gain?

We apologize that **Figure 2G** in our original manuscript may have been too imprecise and easily misinterpreted. In our original manuscript, “A3 loss” does not mean the loss of splicing junctions at the 3' end. Rather, neojunctions characterized as “A3 loss” are those that have lost portions of the canonical 3' exon, and “A3 gain” represents neojunctions that lead to the gain of portions of the intron. We recognize that how these neojunctions were characterized and defined is not entirely clear. We therefore have updated the figure legend text of **Figure 1F** legend, **Lines 539-541**, to avoid the ambiguity in interpretation and better define A3 loss/gain and A5 loss/gain. Based on these definitions, it is true that the majority of the investigated TCGA tumor types have a predominance in A5 gain neojunctions (meaning that the neojunction on the 5' end leads to an inclusion of the adjacent intron). We have not yet investigated this observation in-depth. We hope the reviewer will agree that downstream *in vitro* analyses of these neojunctions and the splicing machinery would be beyond the scope of this paper.

2. Next the authors studied the intratumor heterogeneity (ITH) of the neojunctions. A subset of them was found to be expressed across many/all biopsies and in a high proportion of patients, representing interesting candidates for therapeutic interventions. To this end, the authors focused on gliomas, which are known to have a particularly high ITH. The analysis of 10 maximally distanced biopsies from 56 tumors by whole-exome and RNA-seq showed neojunctions present across multiple patients. Perhaps not entirely surprising, more neojunctions were found in gliomas carrying IDH1 mutations, but they were particularly enriched in oligodendroglioma patients.

Still, quite some ITH was observed, so in general a combination of peptides derived from many neojunctions would be required for a therapy covering the whole tumor. Even if a proportion of the neojunctions was expressed in the entire tumor in at least one case, and some in at least two or more cases, the authors should provide calculations indicating the minimal and average numbers of neojunctions required to cover all the tumor cells, in a certain fraction of patients (e.g. 50%, 75%, 100%). This is important to judge the feasibility of personalized approaches. What were the most conserved and most prevalent neojunctions? What was the highest prevalence among patients? Some data is shown in figure 3F, but the colors are difficult to read and shown is only the fraction of cases with a particular neojunction. The fraction of patients with defined numbers of tumor-wide neojunctions, conserved high, mid and so on, should be also shown, and example numbers mentioned in the text. This would allow to see how many patients had at least one neojunctions present in the entire tumor, how many two, and how many had no tumor-wide neojunctions.

We are glad the reviewer finds our spatially-mapped sampling approach to be useful in identifying neojunctions that demonstrate high ITH. We also thank the reviewer for their advice on improving the presentation of Figure 3F in our original manuscript. At their recommendation, we now revised the figure to illustrate the data from the perspective of each patient sample ($n=51$) rather than each GBM/LGG neojunction ($n=789$). **Figures 2F-2G** and **Extended Data Figure 2D** in our updated manuscript now illustrate the distribution of neojunctions based on their conservation across each patient's tumor. We believe that this restructure has significantly improved the readability of the figure. Additionally, text has been included in **Lines 124-128** to indicate the subset of neojunctions ($n=37$) that were found across all biopsy cores in more than 10% of the study cohort. We hope that this figure clearly illustrates the fraction of tumor-wide, highly-, moderately-, and lowly-conserved neojunctions per patient. Based on our definition of highly-conserved neojunctions in the manuscript (expressed in at least 70% of spatially-mapped samples within a tumor), we have identified 15 neojunctions that, on average, are expressed in $\geq 70\%$ of spatially-mapped samples across all tumors. Importantly, the GNAS NEJ is within this group.

3. A big fraction of the public neojunctions expressed in gliomas was found at the mRNA level in cell line transcriptomic data, peptides derived from 38% of the public neojunctions were found by proteomics. However, protein expression of neojunctions was clearly lower than the canonical junctions (figure 4C) and should be mentioned as such.

Overall, 192(24%) of the neojunctions were detected by both mRNA and proteomics. Prediction algorithms were used to assess the proteasomal cleavage and presentation of tumor-wide neojunctions by MHC class I. 832 peptides were in the top 10% of both algorithms and were further analyzed. 81 were also in the 192 neojunctions previously characterized by mRNA and proteomics expression. The rest of the analysis was done with 32 HLA-A*02:01 binders. High intra-tumoral conservation was observed for most of these and several were chosen for further analysis. These experiments are well done, however care should be taken that correct figures are referenced, as for example Figure 3G could not be found.

We appreciate of the reviewer's kind remarks and are glad that they find our experiments were well performed. Regarding the original manuscript's text, the reviewer is correct in identifying the absence of Figure 3G. As previously addressed in our response to the reviewer's Comment 2, we now include **Figures 2F-2G** and **Extended Data Figure 2D-2E** that interprets the heterogeneity of neojunctions through the perspective of each set of spatially-mapped samples. As such, we have replaced the above text with "50 out of 56 (89.3%) of the patients revealed at least one neojunction detected across all intratumorally-mapped samples, and this suggests the strong possibility of targeting these clonally-conserved NEJs for personalized immunotherapy applications."

4. Next, the authors show that CD8 T cells can recognize neojunction-derived peptides, by testing 4 of the 32 HLA-A*02:01-binding candidates in a in-vitro sensitization assay with peptide-loaded DCs and INF γ readout. Reactivity against two of these peptides was found. 7 TCRs for NeoARPL22 and one 1 TCR for NeoAGNAS were discovered. The reactivity of one TCR per antigen was verified by cloning and transduction into a TCR-negative Jurkat cell line carrying TCR signaling reporters, and coculture with peptide pulsed APCs. Reactivity was observed at relatively low (1nM) peptide concentrations. Similar data was obtained by using PBMC-derived CD8 T cells and measurement of INF γ , TNF α and

degranulation markers. MHC specificity was shown by antibody blocking experiments and endogenous peptide processing and presentation, by TCR recognition of peptides derived from transfected mRNA and HLA ligandomics.

These elegant experiments prove natural peptide processing and the TCR-epitope reactivity, but contrary to the authors conclusions, do not prove immunogenicity, nor high TCR affinity. Even if the higher immunogenicity is likely, because of the “foreigners” of the neojunction-derived peptides, it is dependent on many processes and can only be tested in-vivo. Showing that patient activated/memory T cells (preferably TILs) contain reactivities against such peptides by tetramer staining, ELISPOT or in vitro re-call, would show real immunogenicity and enhance the manuscript greatly. In-vitro experiments show only precursor frequency.

Furthermore, T cell responses in peptide titration experiments are dependent on two factors: the binding affinity of the peptide to the MHC, and the binding affinity of the TCR to the pMHC. Therefore, no conclusion about TCR affinity can be made and instead the term “functional avidity” should be used.

Still, these results clearly indicate the antigen specificity of the TCRs and their possible use for therapy. This aspect is highlighted by showing that TCR-T cells can kill glioma cells in-vitro. Together, the results show that the neojunction-derived peptides can be endogenously processed and presented by the tumor cells in sufficient amounts to allow in vitro TCR-T cell killing. However, even if neojunction-derived peptides are more commonly expressed in tumors, there is nothing particular about them which would make them more likely to be processed and presented by the tumor cells in general. Therefore, each of the neojunctions-derived peptides needs to be tested in a similar way like any other therapeutic candidate peptide. This should be mentioned in the manuscript.

The reviewer presented many insightful comments regarding the terminology and presentation used by our original interpretation of the immune studies. We agree that the experiments utilizing peptide pulsed-APCs co-cultured with TCR-transduced T-cells measure functional avidity rather than TCR affinity or immunogenicity. We have adjusted the main text of the manuscript to reflect the reviewer’s feedback. We agree that functional avidity assays alone are insufficient to demonstrate the therapeutic potential of the candidate peptides, hence our downstream cytotoxicity assays utilizing tumor cell lines. We agree with the reviewer’s final comment regarding the non-correlation between a neojunctions’ expression level and the final presentation of their corresponding neoantigen by HLA. The purpose of the SSNIP approach was to curate a high-confidence list of neoantigen candidates that are not only ubiquitously expressed, but also predicted to be proteolytically cleaved and HLA-bound using two independent prediction algorithms that specialized in both characteristics. From our final candidate list of 32 HLA-A2-bound neoantigens, we selected four candidates for IVS and TCR identification (NeoA_{S100A6}, NeoA_{RPL22}, NeoA_{GNAS}, NeoA_{TCF12}). Interestingly, based on this pilot study and **Figure 4** of our original manuscript, we technically achieved a 100% discovery rate of reactive CD8⁺ T-cell populations for all prioritized NEJs that we investigated and undersold the accuracy of this approach in the original manuscript. This further demonstrates that our SSNIP approach is a remarkably accurate method for identifying neoantigen candidates that are properly processed and presented.

5. Finally, to characterize splicing-related genes which would trigger higher neojunction generation, it was discovered that loss of SNRPD2 and SF3A3 may lead to the increased expression of neojunctions in IDH1 mutant gliomas. For example, almost 50% of the 789 neojunctions increased with decreased SNRPD2 expression.

Nice and clear experiments.

We thank the Reviewer for their positive feedback on our experiments to uncover specific splicing factors that modulate neojunction expression in gliomas.

Minor issues

• The study “Genomic basis for RNA alterations in cancer” by PCAWG Transcriptome Core Group et al. should be mentioned in the manuscript (<https://doi.org/10.1038/s41586-020-1970-0>).

We have included this citation in our updated text in *Line 145*.

- **Figure 3B has no yellow annotations.**

We have now ensured the visibility of the yellow annotations that pertain to clusters of tumor-wide neojunctions.

- **Figure 3G could not be found.**

We apologize for the omission of a G panel in this figure and have fixed the issue as described in reviewer Comment 3.

- **Line 270: alanine scanning is done on epitopes not TCRs**

The text in the updated manuscript now reads, “Alanine scanning mutagenesis of NeoA_{RPL22} (top) and NeoA_{GNAS}-pulsed T2 cells cultured with reactive (bottom) TCR-transduced triple-reporter Jurkat76/CD8 cells co-cultured with alanine-substituted neoantigen-pulsed T2 cells, neoantigen-pulsed T2 cells, or non-pulsed T2 cells.”

- **Flow cytometry data in Figure 6E appear to be missing.**

The reviewer keenly noted that in reference to our statement, “As expected, flow analysis following co-culture showed tumor-specific immune reactivity mounted by both TCR-transduced CD8⁺ T-cell lines against the glioma cell lines (**Figure 6E**).” We did not include the corresponding flow analysis data in the original submission and have removed this line in the updated manuscript.

- **Text in many main and supplementary figures is way too small to be legible.**

We apologize for the recurring legibility issue. We correspondingly adjusted the size of the fonts and modified the figures to make them more legible.

- **Suppl Figure 6 is not legible so clustering of the genes cannot be assessed.**

We made the appropriate changes to address this issue by increasing both the resolution of the figure and clarifying the clustering of the genes in the heatmap.

Referee #4 (Remarks to the Author):

The manuscript “Tumor-wide RNA splicing aberrations generate immunogenic public neoantigens” by Kwok et al. describes the identification of tumor-wide and public neoantigens by datamining different RNASeq sources to land in identifying aberrant splicing in GNAS and RPL22 and other mRNAs associated to IDHmut tumors. Interestingly, they present data that supports that these splicing changes (only presented for GNAS and RPL22) potentially act as neoantigens for cytotoxic T-cells and can be developed into cellular immunotherapies. In addition, the authors present preliminary data on potential splicing factors that may regulate splicing changes associated to IDHmut tumors. This manuscript has a big effort in in silico analyses across and within tumors, however, major experiments are needed to functionally validate the findings, as described below.

Major comments:

1. The thresholds that authors use to decide on relevant NEJ throughout the study should be justified. Also consider presenting more stringent criteria in the analyses in the first 3 figures. Seems that the number of hits are very high based on previous literature.

The Reviewer inquires about the thresholds used to define NEJ expression and how this compares to those used in prior studies. The criteria for our analyses were established based on similar thresholds and filters

employed in a previous study that investigated cancer-specific splicing isoforms in TCGA (Kahles A. et al., 2018). We recognized an oversight in the original depiction of the total TCGA neojunctions. Specifically, Figure 2 of the original manuscript included neojunctions with only one read count and double-negative neojunctions not found in both GTEx and TCGA. Following a careful review and correction of the code, we have now properly visualized our original stringent criteria referenced in the manuscript and ensured a more accurate representation of putative neojunction expression across all TCGA samples in **Figure 1** of our updated manuscript. Importantly, this change does not affect the conclusions of our studies. We sincerely appreciate the reviewer's keen eye in identifying this issue, and we believe that these adjustments strengthen the robustness and reliability of our findings. The revised figures in our manuscript now accurately reflect the refined criteria applied to our analyses.

2. While the authors focus on NEJ/derived peptides that bind to HLA-A*02:01, would be important to present: 1) what NEJs/peptides were identified for other HLAs? Is the identification of NEJs related to differences in the prevalence of HLAs among the cohorts studied? If so, how is this related to prevalence of the HLA and the capacity of HLA binding and immunoediting of tumors? Is the prevalence of HLA affecting skewing the NEJ predicting?

We appreciate the feedback provided by the reviewer. In the original manuscript, we investigated NEJ-derived candidates predicted to bind strongly to five HLA-A types that are expressed across a wide range of demographics⁹ (HLA-A*01:01, HLA-A*02:01, HLA-A*03:01, HLA-A*11:01, HLA-A*24:02) (**Figure 4F, 4K** of the original manuscript). Although the identification of NEJ-neoepitopes is not dependent on the prevalence of any specific HLA since we, ourselves, select the HLA we want to focus on, we agree that the inclusion of more HLA-alleles in our analyses would strengthen the study. Based on the same citation and the Allele Frequency

Net Database, we sought to investigate the 45 most prevalent HLA-A alleles shown in the table below. However, 8 of these alleles (**shown below in gray**) were unavailable for analysis in either HL Athena or MHCFlurry 2.0, and thus we characterized NEJ-derived neoantigens predicted to be presented by the other 37 HLA-A alleles.

01:01	02:01	02:02	02:03	02:04	02:05	02:06	02:07	02:11	02:12
02:19	02:24	02:264	02:52	03:01	03:02	03:27	11:01	11:02	11:06
23:01	24:02	24:06	24:07	24:41	25:01	26:01	26:03	29:02	29:10
29:25	29:50	30:01	30:02	31:01	31:03	31:08	31:29	32:01	33:03
34:01	68:01	68:02	68:03	74:01					

We then re-performed our SSNIP analysis on all 789 glioma-specific neojunctions – this time with the inclusion of all 37 HLA-A alleles. Similar to the analysis performed in **Figure 4** of the original manuscript, we inputted all cancer-specific n-mer candidates against these HLA-A alleles through HL Athena and MHCFlurry 2.0. We characterized all high-confidence candidates (top 1%tile of both HL Athena and MHCFlurry 2.0 scores) based on their combined presentation scores against each HLA-A allele. Despite the presumed predisposition for the HLA-A*02:01 alleles due to the TCGA cohort demographic, no inherent bias for HLA-A*02:01 alleles was observed in our analyses. HLA-A alleles have previously been reported to favor the binding and presentation of shorter length peptides, and our data supports this notion: the large majority of candidates within the top 1%tile have peptide lengths of 8 or 9 amino acids. These findings were included into the updated manuscript as **Extended Data Figures 6F-6G**. As we demonstrate here, we do not see a direct association between the identification of NEJs and the prevalence of any HLA. We are confident that SNIPP can be used to investigate NEJ-derived neoantigens that are strong binders to any of these remaining HLA alleles based on this *in silico* validation. While it would be interesting to investigate further *in vitro*, IVS and 10x V(D)J sequencing of a new cohort of HLA-A allele-bound neoantigens would not be feasible for us to the length of time required and amount of funds.

Furthermore, we looked to investigate whether the neoantigen candidates derived from NEJ_{RPL22} or NEJ_{GNAS} were predicted to be bound and presented by any HLA allele besides HLA-A*02:01. Interestingly, Neo_{GNAS} was predicted to be a strong binder to only a subset of HLA-A2 sub-alleles by both algorithms, specifically HLA-A*02:01, HLA-A*02:04, and HLA-A*02:11. It had drastically reduced predicted binding to any HLA alleles beyond the HLA-A2 family. Similarly, Neo_{RPL22} had strong predicted binding preferences to HLA-A*02:01, HLA-A*02:07, HLA-A*02:12, HLA-A*02:19, and HLA-A*02:24 and significantly reduced affinity to alleles external to the HLA-A2 family. These findings were included in **Extended Data Figures 8C-8D**. Here, we show that our neoantigens are predicted to strongly bind to multiple different HLA-A2 alleles with limited neoantigen presentation capacity across the HLA loci beyond that of HLA-A2.

3. Re: Fig 4 comment should be made on IDH mut vs WT NEJs. Were the IDHmut NEJ found in other IDH mut tumors (i.e LIHC IDHmut)? I know this is covered in Fig 7 but let the reader know that more on this finding is studied in Fig 7.

We appreciate the valuable feedback from the reviewer regarding Figure 7 in our original manuscript and the need for clarification on the presence of IDH-mutated (IDHmut) NJs in other tumor types. To address this inquiry, we performed an analysis on IDH1mut NJs identified in our spatially-mapped glioma samples in **Figure 2E** of the revised manuscript. Specifically, we defined IDH1mut-specific NJs as those showing significant upregulation in IDH1mut samples compared to IDH1wt samples (p -value < 0.05 , \log_2 fold change > 1.5). This stringent analysis led to the identification of 224 NEJs (28.39% of total identified glioma NEJs) as IDH1mut-specific. In response to the reviewer's suggestion, we extended our investigation to include IDH1mut tumor samples from LIHC ($n=4$, TCGA et al., Cell 2017) and PRAD ($n=3$, TCGA et al., Cell 2015). The analysis revealed a substantial number of IDH1mut-specific NEJs that were expressed in these cases. Specifically, 36 (16.07%) and 30 (13.39%) IDH1mut-specific NEJs were found to be expressed in IDH1mut LIHC and PRAD cases, respectively. This significant observation emphasizes the existence of a class of IDH1mut NJs that are conserved across different tumor types with similar IDH1 mutations. We believe that this finding enriches the understanding of the broader implications of the IDH1mut NJ expression patterns and contributes to the overall context of our study. The figure below has been added to the updated manuscript as **Extended Data Figure**

5K, and the corresponding text is found in **Lines 171-172**. We sincerely thank the reviewer for guiding us to enhance the clarity and depth of our analysis.

4. Consider bringing figure 7 before all immune studies. This will keep all the NEJ identification and mechanism together before shifting gears towards to immune-triggering response.

We appreciate the reviewer's advice regarding the narrative flow of the figures and agree with shifting the order of which the figures are presented in the manuscript. To reflect these changes, we have moved Figure 7 of the original manuscript to the present **Figure 3** such that the mechanistic studies are introduced after the discussion of NEJ intratumor heterogeneity but before all immune studies. We believe that this change significantly improves the readability and logic of the manuscript.

5. The authors go through detailed analyses of multiple data sets to land in 32 NEJs, and finally 2 functional NEJ (GNAS and RPL22), however, there is no detailed description on what are the exact NEJs for these genes (A3', A5', SE etc). Are these NEJs impacting the function of the encoded proteins? Producing functional/non-functional proteins?

We appreciate the insightful observation made by the reviewer concerning the lack of detailed information on the exact nature of the identified NEJs in our study, particularly those related to the GNAS and RPL22 genes. We acknowledge the importance of elucidating the impact of the NEJs on the encoded proteins and their potential functional consequences. To address this concern, we provide a detailed characterization of the NEJs in both GNAS and RPL22, shedding light on their specific classifications and the resulting effects on protein structure and function in **Lines 281-283**. For GNAS, the NEJ leads to an aberrant loss of nucleotides at the 3' end, resulting in a frame-shift mutation and a premature stop codon. This combination strongly suggests the production of a non-functional mutant GNAS protein though co-expressed with wild type GNAS. In the case of RPL22, the NEJ induces an in-frame loss of one amino acid, impacting the alpha helix structure. To comprehensively assess the effect on protein integrity, we employed AlphaFold2 to model the structure of both wild-type and mutant RPL22 sequences. As illustrated in **Extended Data Figure 6N** of the revised manuscript, the RPL22 NEJ results in the deletion of one residue in the alpha helix (highlighted in red). Importantly, the overall integrity of the protein structure remains consistent between the wild-type and mutant isoforms. These findings provide a detailed insight into the functional consequences of the NEJs, demonstrating how they are still endogenously processed and presented in **Figure 6** of the revised manuscript despite generating frame-shift mutations, premature stop codons, and alterations in protein structure. We believe that this additional information strengthens the significance of our study by connecting the identified NEJs to potential functional impacts on the encoded proteins. We sincerely thank the reviewer for their valuable input.

6. Importantly, there is no PCR validation of the computational identified NEJs or WB of the encoded proteins and identified splicing factors between tumors and after manipulation with siRNA.

We have performed PCR validation of these NEJs in **Supplementary Figure 4H** of the original manuscript via amplicon sequencing, which can now be found in **Extended Data Figure 6J** of the revised manuscript.

Addressing the reviewer's request for PCR and western blot assessment of the selected splicing factors pre- and post-siRNA-mediated knockdown, we have utilized qPCR (**Extended Data Figure 5B**) in the original manuscript, and for an orthogonal readout, we performed RNA-sequencing (**shown below**) on pre- and post-treated cell lines with biological replicates ($n=3$). RNA-sequencing validates our qPCR quantification of knockdown by which candidate splicing factor expression was reduced by 60% or greater across all cell lines.

Regarding western blotting of the NEJ-encoded proteins, we thank the reviewer for this comment, however, we do not believe this is feasible given the minimal difference in molecular mass of the proteins encoded by the two NEJs of interest and the canonical junction. The greater expression level of the canonical junction compared to the novel junction would also be problematic for detection on a Western blot.

7. Would be nice if authors can use the database “MAJIQlopedia: an encyclopedia of RNA splicing variations in human tissues and cancer” (PMID: 37953365) to re-validate that GNAS, RPL22 and IDHmut-NEJs are cancer specific and potentially conserved across other tumors.

We highly appreciate the thoughtful suggestion from the reviewer to utilize the MAJIQlopedia database for the re-validation of our identified NEJs, particularly those associated with GNAS, RPL22, and IDH mutations. We performed a comprehensive comparison with MAJIQlopedia normal and cancer data, while also acknowledging potential discrepancies introduced by variations in the read aligner of choice, and aligner

versions, as well as the gene annotation files. Despite these challenges, our analysis revealed that a sizeable number of our neojunctions identified through our pipeline were indeed detectable in MAJIQlopedia, including the GNAS NEJ. The number of identified neojunctions ranged from 14 to 54 across all studied tumor. This cross-validation with MAJIQlopedia not only reaffirms the cancer-specific expression of the identified NEJs but also suggests their potential conservation across a spectrum of tumor types. We are grateful to the reviewer for suggesting the integration of MAJIQlopedia into our validation process, enriching the comprehensiveness and reliability of our study. We included this new figure into the revised manuscript as **Extended Figure 1F** and cited PMID: 37953365 in **Line 85**.

8. Re: Fig 7: The authors should also rule-in or rule-out the role of the listed splicing factors in GNAS and RPL22 NEJs.

The reviewer brings up an excellent recommendation for further investigating the effects of the listed splicing factors on the expression of our validated NEJs. In the original manuscript, we used DESeq2 and GSEA to identify two sets of splicing factors that were either upregulated in mutant *IDH1* glioma cases or downregulated due to co-deletion of chromosomes 1p and 19q in *IDH1*-mutant oligodendrogliomas. Across all three glioma subtypes, we performed a three-way Pearson correlation of these upregulated/downregulated splicing genes against all 789 identified neojunctions (**original manuscript's Figure 7**). Per the reviewer's suggestion, we tightened the focus of the analysis to only the NEJs found in *GNAS* and *RPL22*. Pearson correlation analysis illustrated no significant correlation between chromosome 1p/19q splicing genes and NEJs. Rather, it revealed a significant increase in the overall correlation of *IDH1*-mutant-associated splicing factors with NEJ_{RPL22} compared to NEJ_{GNAS} . (**below, Extended Data Figure 5J** and in text as **Lines 175-176**) This robust finding is supported by our identification of NEJ_{RPL22} as an *IDH1*-mutant-specific NEJ based on our characterization in response to the reviewer's Comment 3. This suggests a potential direct or indirect role of the expression of these splicing factors on *IDH1*-mutant-specific NEJs, such as NEJ_{RPL22} .

9. Re: Fig 7: Only loss-of-function studies were performed and using only single -siRNAs. Orthogonal validation studies are required.

We apologize for the lack of clarity in our description for our siRNA knockdown experimental design. We used multiple pooled siRNAs ($n=4$) per splicing factor in our knockdown procedure for each cell line. We have revised the manuscript to reflect these changes.

We also thank the reviewer for their insightful suggestion to include additional orthogonal validation studies. To that end, we recently performed *SNRPD2* and *SF3A3* siRNA knockdown on biological replicates ($n=3$) of GBM115 and *CELF2* siRNA knockdown on biological replicates ($n=3$) of SF10417 and SF10602 in preparation for RNA-sequencing analyses to investigate its effect on the expression of neojunctions. Bulk RNA-sequencing allows us to analyze siRNA knockdown-mediated changes in all NEJs in a high throughput manner at the expense of sensitivity compared to our results from qPCR analyses. We characterized 244 *IDH1*mut-specific neojunctions as those significantly upregulated in mutant *IDH1* glioma cases compared to their wild-type counterpart (\log_2 fold change > 1.5 , p -value < 0.05). RNA-sequencing analysis demonstrated an associated decrease in the expression of 19 (8.60%) and 28 (12.67%) *IDH1*mut-specific neojunctions, respectively, in SF10417 and SF10602 when treated with *CELF2* siRNA compared to their non-treated controls (now included in **Extended Data Figure 6C** of the revised manuscript). We also characterized 52 *IDH1*mut-O-specific neojunctions as those were significantly upregulated in mutant *IDH1*-O glioma cases compared to their mutant *IDH1*-A and wild-type counterparts (\log_2 fold change > 1.5 , p -value < 0.05). RNA-sequencing analysis demonstrated an associated increase in the expression of 7 (13.46 %) and 4 (7.69%) *IDH1*mut-O-specific neojunctions in GBM115 cells treated with *SF3A3* and *SNRPD2* siRNA, respectively (**Extended Data Figure 6D** in the revised manuscript). These findings demonstrate that the correlation assessment performed from TCGA samples is supported by the results of RNA-sequencing of siRNA knockdown of corresponding tumor cell lines.

While validating our qPCR findings corresponding to siRNA knockdown of *CELF2*, *SNRPD2*, and *SF3A3* and their effects on NJ_{ACAP2} expression, we observe a consistent trend across 3 out of 4 of our cell line conditions: In both *SNRPD2* and *SF3A3* knockdowns in GBM115, we observe an overall increase in the expression of NJ_{ACAP2} which supports the initial findings in Figure 7L of the original manuscript); Similarly, we were able to observe an overall decrease in NJ_{ACAP2} expression in 1 of 2 mutant *IDH1* glioma cell lines, which corresponds to the results in the original manuscript's Figure 7J. These additional findings shown below are included in **Extended Data Figure 6E** of the revised manuscript. Given all of the data we have provided, we believe that this should be sufficient as an orthogonal approach.

10. Can immune studies be performed on the IDHmut NEJs?

As the reviewer noted in Comment 3, we have defined IDH1mut-specific NJs as those showing significant upregulation in IDH1mut samples compared to IDH1wt samples (p-value < 0.05, log2 fold change > 1.5). As previously discussed in Comment 8, NEJ_{RPL22} is considered an IDH1mut-specific neojunction, and as such, we have performed the corresponding immunogenicity and cytotoxicity analyses on this NEJ in Figure 5 and Figure 6, respectively.

Minor comments:

11. What is public neoantigens? Best to define the first time used. You only understand as you read more into the manuscript.

We agree that “public” was not clearly defined early in the manuscript, and we have addressed this at the first instance that this term was introduced in the revised manuscript.

12. Fig.3E is not clear how clustering is associated to IDH1 status.

Originally, the legend presented the different glioma subtypes as GBM (blue), Astro (yellow), and Oligo (red). However, we recognize that many readers are not familiar with the *IDH1* mutation status associated with each of those glioma subtypes. We changed the legend to *IDH1*wt (blue), *IDH1*mut-A (yellow), and *IDH1*mut-O (red) to more definitively define clustering based on *IDH1* mutational status.

13. Grammatical error in line 140.

We thank the reviewer’s attention to detail and have made the appropriate corrections.

14. NEJ acronym was not introduced and had to be deduced from text.

We thank the reviewer for this comment and have made the appropriate corrections. The “NEJ” acronym was introduced in the original manuscript under the section titled “Tumor-wide neojunctions encode neoantigens predicted to be processed and presented by MHC-I” (Line 24). However, in the original text, NEJ was the acronym for “neopeptide-encoding neojunctions” which is now corrected to “neopeptide-encoding junctions” to more accurately reflect the acronym.

15. Re: Fig 4: A comment should be added on potential reason(s) why TCF12 NEJ did not elicit a response. Is the encoded antigen not strong enough?

The reviewer brings up an excellent point regarding the lack of reactivity from the NEJ_{TCF12}-derived neoantigen in our IVS study. In general, lack of response is related to the duality of antigen and TCR properties that lead to immune activation, not antigen strength alone. From the antigen perspective, our SNIPP approach controls for

high-confidence neoantigens by predicting candidates that have a strong propensity to be properly processed by the proteasome and presented by HLA. By controlling for this half of the equation for immune reactivity, we increase the chance of identifying specific antigen-specific TCRs in peripheral blood-derived PBMCs. The lack of NeoA_{TCF12}-specific TCRs in **Figure 4** is therefore likely due to the lack of reactive CD8+ T-cell clones in the PBMC rather than the strength of the neoantigen itself. To further support this statement, we have attached a preliminary data figure (**shown below**) of a pilot study IVS performed on our healthy **Donor 0**. We did not include this figure in the original manuscript as we did not have the 10x V(D)J sequencing resources available at the time of the pilot study, and were not able to retrieve the TCR sequences reactive to the NEJs derived from both S100A6 and TCF12. Interestingly, based on this pilot study and **Figure 4** of our original manuscript, we technically achieved a 100% discovery rate of reactive CD8+ T-cell population of all four NEJs that we selected and investigated. This further demonstrates that our SNIPP approach is reasonably accurate in identifying neoantigen candidates that are properly processed and presented in a high throughput manner.

16. Fig 6 has panels that are hard to interpret due to the small font-size.

We have made the appropriate changes to the fonts in **Figure 6**.

This email has been sent through the Springer Nature Manuscript Tracking System NY-610A-SN&MTS

1. Davis, C. F. *et al.* The somatic genomic landscape of chromophobe renal cell carcinoma. *Cancer Cell* **26**, 319–330 (2014).
2. null null. Comprehensive Molecular Characterization of Papillary Renal-Cell Carcinoma. *N. Engl. J. Med.* **374**, 135–145.
3. Cancer Genome Atlas Research Network. Electronic address: wheeler@bcm.edu & Cancer Genome Atlas Research Network. Comprehensive and Integrative Genomic Characterization of Hepatocellular Carcinoma. *Cell* **169**, 1327-1341.e23 (2017).
4. Cancer Genome Atlas Research Network. Comprehensive molecular profiling of lung adenocarcinoma. *Nature* **511**, 543–550 (2014).

5. Cancer Genome Atlas Research Network. The Molecular Taxonomy of Primary Prostate Cancer. *Cell* **163**, 1011–1025 (2015).
6. Cancer Genome Atlas Network. Genomic Classification of Cutaneous Melanoma. *Cell* **161**, 1681–1696 (2015).
7. Alter, G., Malenfant, J. M. & Altfeld, M. CD107a as a functional marker for the identification of natural killer cell activity. *J. Immunol. Methods* **294**, 15–22 (2004).
8. Otano, I. *et al.* CD137 (4-1BB) costimulation of CD8+ T cells is more potent when provided in cis than in trans with respect to CD3-TCR stimulation. *Nat. Commun.* **12**, 7296 (2021).
9. González-Galarza, F. F. *et al.* Allele frequency net 2015 update: new features for HLA epitopes, KIR and disease and HLA adverse drug reaction associations. *Nucleic Acids Res.* **43**, D784-8 (2015).
10. Liu, L. *et al.* IDH1 fine-tunes cap-dependent translation initiation. *J. Mol. Cell Biol.* **11**, 816–828 (2019).

Dear Editor and Reviewers:

We appreciated the reviewers' constructive comments on our the first revision of our manuscript titled "Tumor-wide RNA splicing aberrations generate immunogenic public neoantigens." We have addressed all of the reviewers' comments and responded in a point-to-point manner below.

Referees' comments:

Referee #1 (Remarks to the Author):

I've gone through the author's answers to Reviewer#1's comments and they appear to me to be satisfactory. On both major comments, the authors performed a number of additional computational analyses and included the results (figures) in the manuscript.

On the heterogeneity and sampling, the authors recognized the technical limitations however, their 10+ biopsies performed is rather amazing. The bottom line is that one would never be able to get all targets when sampling is involved. In my opinion, the authors have adequately answered comment 1. Note that Extended Data Figure 2 doesn't contain red points in their graph as they pointed out.

The issue with comment #2 pertained to expanding the analysis to other tumor types and their subtypes to demonstrate an association of splicing factor mutations and NJs. The authors have expanded their analyses to now include other tumor types and show strong association.

[Our response] We appreciate the kind remarks of the reviewer and thank them for their careful observation regarding **Extended Data Figure 2K**. In the most recent version of the manuscript, the figure legend and **Lines 128-129** now accurately reflect the figure.

Referee #2 (Remarks to the Author):

In this extensively revised version of their manuscript, Kwok and colleagues beautifully strengthened their study of public neojunctions by clarifying text and adding copious analyses. This current version of their work significantly solidifies their previous conclusions. The authors have addressed all of my comments exquisitely.

[Our response] Thank you!

Referee #3 (Remarks to the Author):

The revised manuscript is substantially improved, and I appreciate the efforts that the authors have made. However, while the authors addressed most of my concerns, they did not really address the one concerning immunogenicity. Given that it is claimed in the title, convincing results showing in-vivo T-cell responses should be provided.

In an attempt to show immunogenicity of the NEJ-derived peptides, the authors analyzed the frequency of peptide-specific CD8+ T-cells in patient PBMCs (n=3) after in-vitro stimulation with peptide-loaded APCs (IVS). They see positive cells in one patient only, which could represent de-novo priming, especially since the description of the IVS in the methods mentions the isolation of naive T-cells. To control for de-novo priming the authors provide results from PBMCs of one healthy donor. Given the low frequency of responses in patients (1/3), this is way too little.

As mentioned in the original review, I see two possibilities to address this:

- 1. (preferred) by showing that in-vivo activated or memory T-cells (preferably TILs) from patients contain reactivities against NEJ-derived peptides by tetramer staining, ELISPOT, in vitro re-call etc., or*
- 2. by IVS experiments showing a clear, statistically significant difference, between a substantial number of patients and healthy donors. Unfortunately, neither was provided so far.*

I am still supportive of the publication, but to claim immunogenicity, more patients and controls must be analyzed and/or ex-vivo activation/phenotypic analysis needs to be provided.

[Our response] We thank the Reviewer for finding that our revised manuscript has been substantially improved and are grateful they are supportive of publication. We wish to clarify that the IVS experiments using patient-derived PBMCs were performed with a bulk population of CD8⁺ T-cells that included memory T-cells and not isolated naïve cells. We have added language to our results and methods sections to state this experimental detail more clearly.

The immune monitoring data featured in our revision made use of archived biospecimens obtained from glioma patients which satisfied the following criteria: 1) expressed at least one copy of *HLA-A*02:01*, 2) had tumors with documented *GNAS* NEJ expression by RNA-seq, and 3) had cryopreserved PBMC samples obtained when the subject was on minimal to no exogenous steroids. No other subjects in our existing biorepositories satisfied all three criteria. We attempted to perform immune monitoring on *n*=10 additional *HLA-A*02:01*⁺ subjects who were on intermediate to high doses of steroids at the time of collection; however, the viability of the thawed specimens was sufficiently poor to preclude reliable testing.

We previously performed immune monitoring on a cohort of patients with diverse solid cancers that express an *HLA-A*03:01* restricted public neoantigen resulting from a *PIK3CA* hotspot mutation (Nat Med. 2022, PMID: 35484264). Despite all patients expressing both the restricting HLA allele and mutant *PIK3CA* allele, only 30% of subjects had evidence of a “spontaneous” T-cell response to the neoantigen. A similar response frequency was observed in a second cohort of patients with extra-cranial solid cancers that express an *HLA-A*01:01* restricted public neoantigen resulting from a recurrent *NRAS* Q61 hotspot mutation (Etxeberria and Klebanoff, personal communication). We anticipate that the frequency of patients with circulating neoantigen-specific T-cells will be significantly less than 30% in subjects with gliomas. This is because antigens derived from the CNS parenchyma may not effectively prime circulating T-cells and treatments, such as chemotherapy, radiation therapy, and corticosteroids, can promote immune suppression and mask antigenicity (Nat Immunol. PMID: 28092374).

Given these limitations, it is not feasible for us to prospectively increase our sample size to enable statistical comparison between patients and healthy donors within the time frame required for a revision. We note that several recent studies have concluded that the capacity to generate T-cell responses to neoantigens using the “outsourced” repertoire of HLA-matched healthy donors, a strategy we also employed, is a more reliable measure of immunogenicity and actionability than the ability to measure T-cell responses in patients. For example, Strønen et al. found that a significant proportion (82%) of endogenously processed and presented neoantigens are neglected by autologous tumor-infiltrating lymphocytes in melanoma patients (Science 2016, PMID: 27198675). However, polyclonal T-cells transduced with TCRs cloned from healthy donors following *in vitro* stimulation with candidate neoantigens could recognize patient-derived tumor cells. Similarly, Bassani-Sternberg et al. discovered that the majority (63%) of mass spectrometry-identified neoantigens failed to generate a detectable T-cell response in melanoma patients (Nat Commun. 2016, PMID: 27869121). The authors discovered that IVS of T-cells from HLA-matched donors using mass-spec identified peptides could generate neoantigen-specific T-cells, establishing the immunogenic potential of these epitopes. Thus, despite the limited number of glioma subjects analyzed for “spontaneous” immunogenicity in our study, the central conclusion of our manuscript – that a subset of recurrent NEJs undergo physiologic processing and HLA presentation to generate therapeutically actionable shared neoantigens – remains unchanged. We briefly summarized these points in the Discussion **Lines 388-390**.

Minor issue:

The gating of the flow cytometry data in Extended Data Figure 8B looks strange for the APC-A parameter (it cuts the whole population in half). Was the gate set according to a FMO (“fluorescence minus one”) control? If yes, it should be shown, if not, it should be done, and the data re-analyzed.

[Our response] We thank the reviewer for their attention to this figure. We re-reviewed the flow gating of this analysis. Per the reviewer’s suggestion and singling out the control with independent fluorescence, the independent APC-A channel exhibited lower read-out and thus greater variance compared to the PE-A channel. We therefore removed the APC-A condition for this specific experiment and reanalyzed the data to re-gate for the PE-A channel alone to have clearer population gating specifically for CD137⁺ CD8⁺ T-cells. CD137 alone is already significant in demonstrating TCR priming and T-cell effector functions. (Otano I. et al., *Nature*

Communications 2021; PMID: 3491197) We remade the CD137+ surface expression plot and moved it to **Extended Data Figure 8C**. The flow gating is updated in **Extended Data Figure 8B** shown below.

In place of the CD137 surface stain, we performed ELISA assays on the supernatant samples from the same experiment as an extension of our response to the reviewer's original request for orthogonal methods of characterizing the cytotoxicity. Notably, our ELISA illustrates significantly elevated levels of granzyme B excreted from NJ_{GNAS} CD8+ TCR-specific T-cells compared to non-transduced T-cells co-cultured with various cancer cell lines. We replaced the previous CD137 surface staining results with these results in updated **Figure 6I** (shown below).

In addition, we performed 3 additional orthogonal ELISA assays for secreted IFN γ , IL-2, and TNF α . Significant increases in the expression of all three of these cytokines were shown in co-cultures with TCR-transduced CD8+ T-cells compared to their non-transduced counterparts. In the revised manuscript, we now include these additional findings as **Extended Data Figures 8D-8F** (see figure panels below). We believe that the inclusion of these assays further confirms the mechanism of both neoantigen-specific CD8+ T-cell cytotoxicity and TCR activation.

Referee #5 (Remarks to the Author):

Comments on revised manuscript Kwok et al.

Comments: After reading the revised manuscript, the comments of Referee 4 and the authors' rebuttal letter we conclude that two major points that were raised by Referee 4, as they pertain to splicing, were not addressed satisfactorily. We feel that these are major points and need to be addressed if the authors want to claim that there is involvement of splicing factors in generating neojunctions.

1. *The authors do not provide data validating that the splicing events creating neojunctions are regulated by knockdown of the splicing factors. Referee 4 point 6 asked for PCR validation of the computational identified NEJs or WB of the encoded proteins and identified splicing factors between tumors and after manipulation with siRNA. The authors respond in their rebuttal letter "We have performed PCR validation of these NEJs in Supplementary Figure 4H of the original manuscript via amplicon sequencing, which can now be found in Extended Data Figure 6J of the revised manuscript." Extended Data Figure 6J shows "Read frequency of reads spanning neojunctions in RPL22 and GNAS compared to the canonical junction spanning reads in glioma cell lines (n=1)". This does not validate the splicing events pre and post knockdown of splicing factors. The authors also provide Extended Figure 5A, B. Extended Figure 5A, B shows "RNA-sequencing-derived (A) and qPCR (B) TPM expression of CELF2, SNRPD2, and SF3A3 following siRNA knockdown." This is validation of splicing factor expression from pre and post treated cell lines, but the authors do not supply PCR validation of the computational identified NEJs. The authors respond in their rebuttal letter that performing WB to validate NEJ-encoded proteins is not feasible. They should make some effort to validate by PCR.*

[Our response] In direct response to the reviewers comments, we present both the existing siRNA effects, and the new CRISPRi effects on NJ production. Reviewer 4's Point #6 stated, "there is no PCR validation of the computational identified NJs or WB of the encoded proteins and identified splicing factors between tumors and after manipulation with siRNA". **Extended Data Figure 5D** (originally **Figures 3J and 3L**) are the PCR validations of the computationally identified NJ_{ACAP2} and NJ_{PEA15} before and after our initial manipulation with siRNA against splicing factors CELF2 (**Figure 3J**), SNRPD2 (**Figure 3L, left**), SF3A3 (**Figure 3L, right**). As discussed in our response to the next comment, we also have applied an orthogonal knockdown approach, CRISPRi (**Figure 3J, 3L, Extended Data Figure 5A**), and show a significant impact on the splicing factor gene target of the CRISPRi, and the NJ level after knockdown. Together the analysis before and after siRNA, and before and after CRISPRi targeting splicing factors (see below), show substantial effects on NJ production.

In summary, **Extended Data Figure 5D** in the second revision (**Figures 3J and 3L in the first revision manuscript**) illustrate qPCRs of NJ_{ACAP2} and NJ_{PEA15} before and after siRNA knockdown of *CELF2*, *SNRPD2*, or *SF3A3*. **Extended Data Figures 5A and 5B** confirm significant knockdown of the splicing factors that are targeted to generate the effect. We explain our selection of NJ_{ACAP2} and NJ_{PEA15} in response to the reviewer's following comments below.

2. *We agree with Referee 4's request for orthogonal validation studies (Referee 4 point 9). The authors should provide this either by experiments with overexpression of the splicing factors or rescue experiments. Extended Figure 5C, D does not satisfactorily address Referee 4 point 9. The authors supply RNA-seq data analysis of further siRNA knockdown experiments. From the rebuttal letter: "RNA-sequencing analysis demonstrated an associated decrease in the expression of 19 (8.60%) and 28 (12.67%) IDH1mut-specific neojunctions, respectively, in SF10417 and SF10602 when treated with CELF2 siRNA compared to their non-treated controls (now included in Extended Data Figure 6C of the revised manuscript)." (I assume that the authors are referring to Extended Data Figure 5C of the revised manuscript). Again, the authors do not validate any specific neojunction (a similar comment to Referee 4 point 6). An example of orthogonal validation would be knockout by a different method than siRNA, such as CRISPR knockout, or overexpression or rescue experiments.*

[Our response] As recommended by the reviewer, we have taken an orthogonal approach, CRISPR interference (CRISPRi) experiments to assess the relationship of *CELF2*, *SNRPD2*, and *SF3A3* with specific neojunctions. Knockdown of the *IDHmut*-associated *CELF2*, located on chromosome 10, was performed on *IDHmut* glioma cell lines to determine whether the *CELF2* loss was associated with a respective decrease in the expression of NJ_{ACAP2} (below figure – top left). Knockdown of *SNRPD2* and *SF3A3*, located respectively on chromosomes 19q and 1p, was performed on GBM115 which contains both copies of chromosomes 1p/19q (below figure – top middle and right), and we would expect to see a respective increase in the expression of

NJ_{ACAP2} and NJ_{PEA15}. These results have confirmed the correlations we have identified *in silico* and show equivalent or greater statistical significance compared to the siRNA experiments. Please find below our NJ measurements, before and after siRNA (below figure – middle row), and before and after the new CRISPRi (below figure – bottom row). Multiple sgRNAs were tested to identify those that achieve a strong knockdown compared to controls. We have now moved the prior siRNA data to **Extended Data Figures 5C-5D** and the new CRISPRi data is now in **Figures 3J and 3L**. The target gene knockdown data is now in **Extended Data Figure 5A**.

Other comments:

- Line 169-170 Figure 3J "... and observed a trend of decreased expression of the associated neojunction across both lines (Figure 3J). The graphs in Figure 3J show that this decrease is non-significant.

[Our response] We thank the reviewer for their feedback. In response to the reviewer critique point #2 for orthogonal validation, we have conducted CRISPRi experiments. With these recent experiments, we observed a significant decrease in NJ_{ACAP2} upon knockdown of *CELF2*. This was observed with sgCELF2_2, which obtained a stronger suppression of *CELF2* than sgCELF2_1, suggesting a dose-dependent inverse relationship between *CELF2* and NJ_{ACAP2} level, which we hypothesized from the correlative data. This data set is now in the main figures as **Figure 3J** (shown below). Significance values: * $P < 0.05$, ** $P < 0.01$, *** $P < 0.001$, **** $P < 0.0001$, ns, no significance.

- Line 191-192 "Similarly, with increasing levels of SF3A3 expression, 178 (22.6%) neojunctions tended to increase in expression, and 127 (16.1%) neojunctions tended to decrease in expression." This is in direct conflict with the conclusion of the next sentence and what is shown in the figure "Notably, siRNA knockdown of either SNRPD2 or SF3A3 in the GBM115 cell line (Extended Data Figures 5A-5B) led to a significant increase in the expression levels of their associated neojunctions". This statement needs to be changed.

[Our response] We sincerely apologize for the confusion created by our miswording of this sentence. We corrected **Lines 189-190** to the following: "Similarly, with decreasing levels of SF3A3 expression, 178 (22.6%) neojunctions tended to increase in expression, and 127 (16.1%) neojunctions tended to decrease in expression."

- Line 192-195 "Notably, siRNA knockdown of either SNRPD2 or SF3A3 in the GBM115 cell line (Extended Data Figures 5A-5B) ..." The fourth panel of Extended Figure 5B shows an increase in SF3A3 expression with siSF3A3, where one would expect a decrease.

[Our response] We apologize for the confusion, and we thank the reviewer for pointing this out. Upon reviewing, we found issues with the export of the file that had gone unnoticed by us, and we've correctly replaced **Extended Data Figure 5C's** panel with the corrected one below:

- It is not clear why the authors chose to validate neojunction ACAP2 (Figure 3J, 3L) and not one of the neojunctions that are the focus of the manuscript (NEJ-RPL22 and NEJ-GNAS). The figure legend for Figure 3L reads: "Expression of NJACAP2 in GBM115 treated with control siRNA or siSNRPD2 (left) or siSF3A3 (right)". The Y axis of the right panel of Figure 3L is labelled PEA15NJ, and not ACAP2NJ.

[Our response] We have included rationales for our choices in **Lines 160-165** for the *CELF2* knockdown and in **Lines 189-192** for the *SNRPD2* and *SF3A3* knockdowns. *NJ_{ACAP2}* and *NJ_{PEA15}* are two NJs that directly correlate or inversely correlate, respectively with expression of those splicing factors in tumor tissue, which led us to the mechanism portions of this manuscript. In contrast, *NJ_{GNAS}* and *NJ_{RPL22}* are investigated as candidate NJs for the **immunotherapeutic aspect** of the manuscript. Our correlative analyses did not identify a candidate splicing factor correlating specifically with *GNAS* and *RPL22* NEJs. We illustrate the logic flow below, separating the immunotherapeutic aspects from the mechanism of NJ production aspects:

To further clarify, our decision to assess *NJ_{ACAP2}* and *NJ_{PEA15}* is based on the three-way Pearson correlation where we sought to identify NJs that were the most negatively or positively correlated with the expression of our selected splicing factors, *CELF2*, *SNRPD2*, or *SF3A3*. *NJ_{ACAP2}* was most positively correlated with the expression of *CELF2* and most negatively correlated with the expression of *SNRPD2*, and thus we selected the *NJ_{ACAP2}* as the candidate NJ that would likely have the greatest likelihood of being modulated by dysregulated expression of *CELF2* and *SNRPD2*. Likewise, *NJ_{PEA15}* was shown to be most inversely correlated with the expression of *SF3A3* across all three glioma subtypes. We understand that the details for this decision were not made entirely

clear in the figure, and to address that, we boxed and labeled the corresponding correlated NJs in **Figures 3I and 3K** to highlight the chosen candidate NJs for the mechanistic study.

For the *CELF2* knockdown, the updated text now reads “We performed CRISPRi- and siRNA-mediated knockdown of *CELF2* in patient-derived *IDH* mutant cell lines⁶⁵ (**Extended Data Figures 5A-5B**), and investigated the change in expression of the highest correlating neojunction (NJ_{ACAP2}) (**Figure 3I**) associated with this splicing-related gene. With CRISPRi-mediated knockdown of *CELF2*, we observed a significant decrease in the expression of NJ_{ACAP2} (**Figure 3J**), and siRNA-mediated knockdown demonstrated a similar trend of reduced NJ_{ACAP2} expression (**Extended Data Figure 5D**).”

For the *SNRPD2* and *SF3A3* knockdowns, the updated text now reads “We investigated whether the two NJs that showed the strongest inverse correlations, NJ_{ACAP2} and NJ_{PEA15} (**Figure 3K**), against *SNRPD2* or *SF3A3* expression might be causally linked to the expression of these splicing factors. Notably, both CRISPRi and siRNA knockdown of either *SNRPD2* or *SF3A3* in the GBM115 cell line (**Extended Data Figures 5A-5C**) significantly increase expression of NJ_{ACAP2} or NJ_{PEA15} (**Figure 3L, Extended Data Figure 5D**).”

- In response to Referee 4 point 9 the authors write “Similarly, we were able to observe an overall decrease in in NJACAP2 expression in 1 of 2 mutant IDH1 glioma cell lines, which corresponds to the results in the original manuscript’s Figure 7J. These additional findings shown below are included in Extended Data Figure 6E of the revised manuscript.” (I assume the authors are referring to Extended Figure 5E in the revised manuscript.) Also, it is not clear what is the left-hand panel in the rebuttal letter referring to (it is not in the revised manuscript).

[Our response] The reviewer is correct in noting that we intended to refer to **Extended Data Figure 5E** in the initial revision manuscript (**Extended Data Figure 5G** in the current version). The left-hand panel in the previous rebuttal letter was for rebuttal purposes only, and therefore was not included in the manuscript.

- Line 233-234 “When considering both RNA-seq and MS confirmation of glioma-specific neojunctions, we validated the presence of 192 (24.3%) public neojunctions expressed across all patient-derived samples (Figure 4D).” Figure 4D is a schematic demonstrating the selection of high-confidence neojunctions for downstream analysis and not validation.

[Our response] Thank you. We corrected **Lines 233-235** to reflect that **Figure 4D** is a schematic illustrating the selection of high-confidence neojunctions rather than a validation as the reviewer points out. We adjusted the text to now read as “When considering both RNA-seq and MS confirmation of glioma-specific neojunctions, we selected 192 (24.3%) public neojunctions expressed across all patient-derived samples for subsequent investigations (**Figure 4D**).”

- Line 267-269 “ When ITH of these 32 neojunctions was investigated in the data set from spatially mapped samples, high intratumoral conservation was observed for most of these NEJs, particularly the two nucleotide A3 loss-encoding neojunction located within GNAS (NJGNAS) (Figure 4M).” The authors do not comment on the NEJ-RPL22, one of the NEJs that is the focus of the manuscript.

[Our response] We initially did not include details of NEJ_{RPL22} due to the word limit. However, in the revised manuscript, we supply a brief descriptive sentence detailing NEJ_{RPL22} in **Lines 283-286** per the reviewer’s suggestion. The text reads, “Neo_{GNAS} results in an A3 loss of 2 nucleotides that generates a frame-shift and premature stop codon. Neo_{RPL22} encodes for an in-frame A3 loss of 6 nucleotides, resulting in a loss of two amino acids in an alpha helix (**Extended Data Figure 6N**).”

- Line 378 -379 “ The higher expression level of the canonical GNAS allele over RPL22 may contribute to the prevalence of NJGNAS detected across all analyses.” Not clear what is meant by higher expression of GNAS allele over RPL22?

[Our response] We appreciate the opportunity to clarify the specified text. *GNAS* is transcribed at much higher levels than *RPL22* (3-fold greater expression considering the FPKM of glioma RNA-sequencing data). Therefore, if the detection of NJ_{GNAS} and NJ_{RPL22} is constant, the detection of NJ_{GNAS} will theoretically be much more likely.

To clarify, we adjusted the main text in **Lines 383-386** to “The higher neojunction levels detected for NJ_{GNAS} compared to NJ_{RPL22} might be explained by the fact that *GNAS* typically exhibits higher transcript expression levels than *RPL22* in tumor tissue. This differential expression could contribute to the greater prevalence of NJ_{GNAS} detected in our analyses, thereby enhancing immunogenicity and tumor-specific killing by TCR_{G4.1} **(Figures 6F-6G).**”

- *Figure legend for Extended figure 6N is missing.*

[Our response] Thank you for catching this omission. The figure legend text for Extended Figure 6N has now been included.

Dear Editor and Reviewers:

We appreciated the reviewers' constructive comments and other comments from the editorial office on our revised manuscript "Tumor-wide RNA splicing aberrations generate therapeutically actionable public neoantigens" (the original title was "Tumor-wide RNA splicing aberrations generate immunogenic public neoantigens"). We have addressed all of the reviewers' comments and responded in a point-to-point manner below.

1. The number of main text references should be 60 in total or less - currently there are 120.

Reference total has been reduced to 60.

2. Please ensure that the methods references are continuously numbered.

Methods references are included with the main text references, all of which are continuously numbered and within the 60 references limit.

3. Please re-supply the Extended data figures individually in EPS, JPEG or TIF format.

Extended Data Figures have been individually resubmitted in JPEG format.

4. Please reduce subheadings to 40 characters (with spaces) or less.

Subheadings have been reduced in length to as close to 40 characters as possible, ensuring clarity and maintaining the scientific message.

5. The text is currently 4919 words. Please try to reduce to nearer 4500 words if possible.

Main text has been reduced to under 4500 words.

6. Please shift one of the Main figures to Extended Data. I suggest Fig. 5 as it's huge and it will be a struggle to fit this on a single page of final print text while retaining legibility.

Select figure panels in Figure 5 have been moved to the Extended Data Figures in order to reduce the overall figure size. Extended Data Figure 9 has been introduced and substantial reorganization of both Main Figures and Extended Data figures have been made. These changes are summarized in the following table.

Previous Figure #	New Figure #
Main Figure 5E	Extended Data Figure 7A
Main Figure 5F	Extended Data Figure 7B
Main Figure 5G	Main Figure 5E
Main Figure 5H	Main Figure 5F
Main Figure 5I	Main Figure 5G
Main Figure 5J	Main Figure 5H
Extended Data Figure 7A	Extended Data Figure 7C
Extended Data Figure 7B	Extended Data Figure 7D
Extended Data Figure 7C	Extended Data Figure 8A
Extended Data Figure 7E	Extended Data Figure 8B

Extended Data Figure 7D	Extended Data Figure 8C
Extended Data Figure 8	Extended Data Figure 9

7. Please provide a supplementary information guide.

The Supplementary Information Guide has now been made and included.

8. The supplementary videos do not have titles/legends. Please add.

Supplementary Video 1 now has both a title and legend. This is shown also in the Supplementary Information Guide.

8. There are potential third party rights issues in the figures - please check sources or if permissions are needed for the human diagram, instruments, petriplate, cells, brain, antigen, antibody, brain, PBMC isolation tube illustrations in the figures.

Third-party rights form was completed and submitted with this most recent revision.

9. There are potential third party rights issues in the figures and the TPR table has not been provided.

Third-party rights form was completed and submitted with this most recent revision.

10. Please ensure that the text size in all figures is at least 5 pt Arial.

Figure text has been updated to reflect increased font size for visibility.

11. Please ensure all main figure legends are 300 words or less (Fig 2: 346, Fig 3: 404, Fig 5: 357, Fig 6: 396).

All figure legends have now been updated to contain 300 words or less. Due to the word count limitations, additional statistical information has been moved to **Supplementary Table 3**.

12. Unfortunately our formatting doesn't allow tables in the Methods section. Please remove the tabular formatting for the CRISPR interference section - it's fine to have the guides as a list.

The CRISPR interference table has been removed. In place of it is a bulleted list of all CRISPR interference oligonucleotide names and sequences ordered from IDT.

13. The Discussion currently has a subheading ("Limitations.."). Please remove and make this continuous text.

The subheading "Limitations of the Study" has been removed from the manuscript text.

14. In the Methods please have separate "Data Availability" and "Code Availability" sections. Ensure that any accession codes are provided and released.

The Methods section has been updated with both a Data Availability and Code Availability section. All relevant accession codes are provided.

15. Please double check all your Extended Data Figs. as some of them are of poor resolution making it difficult to read - e.g. Ext Data 7.

Extended Data Figure 7D has been replaced with an updated figure with higher resolution.

16. Please address any of the outstanding minor points of the Referees.

All minor points from Referees #3 and #5 have been addressed in the manuscript, and our responses to remaining points are shown below:

I thank the authors for carefully considering my comments. The issue related to the FACS analysis was addressed satisfactorily. However, the issue of immunogenicity is still controversial. I fully appreciate the reasoning of the authors and agree that a thorough analysis of true in-vivo immunogenicity is very difficult due to the complex biology, and often not possible due to practical constraints. However, if something cannot be convincingly shown, it cannot be claimed.

Therefore, considering all the above, and following the authors response to my comment on page 2 "... the central conclusion of our manuscript – that a subset of recurrent NEJs undergo physiologic processing and HLA presentation to generate therapeutically actionable shared neoantigens – remains unchanged", I suggest changing the title of the manuscript to "Tumor-wide RNA splicing aberrations generate therapeutically actionable shared neoantigens" or similar. In this case, immunogenicity is not claimed as the main finding, and the interesting results of this study can be made available to the public.

Furthermore, following the same logic, I would avoid the use of the word "immunogenic" in the manuscript as much as possible and suggest the following changes:

1. Delete "immunogenic" from the sentence in line 61: "We identified /immunogenic/ neojunction-derived TSAs that were proteolytically-processed and presented on a prevalent human leukocyte antigen (HLA) molecule."
2. Change "immunogenicity" to "responses" in line 274: "We next sought to determine whether NEJ-derived neopeptides can drive T-cell /immunogenicity/ responses."
3. Delete "immunogenicity" from the sentence in line 281: "...CD8+ conditions revealed neoantigen-reactivity /immunogenicity/ in two out of four..."
4. Change "immunogenicity" to "reactivity" in line 274: "allowed the demonstration of neoantigen-specific /immunogenicity/ reactivity and tumor-specific killing"
5. Delete "immunogenic" from the sentence in line 988: "/Immunogenic/ Cytokine assays were performed..."
6. Change "immunogenicity" to "reactivity" in line 1064: "maintained for downstream co-culture and /immunogenicity/ reactivity assays"; and line 1191: "In dose-dependent /immunogenicity/ reactivity assays,"

Taken together, provided these changes are made, I support the publication of this manuscript.

All changes requested from Reviewer #3 were implemented in the updated manuscript.

Referee #5 (Remarks to the Author):

The second revision of the manuscript has positively addressed our comments raised previously. The authors have addressed our concern on the need for orthogonal validation studies to assess the relationship between the splicing factors and NJ expression. They have now performed CRISPR-mediated knockout of the splicing factors to determine its effect on the expression of the different NJs (Fig. 3J, L) and moved the siRNA knockout to the extended data (Extended Data 5C-5D). However, the effect on the expression of NJACAP2 upon CELF2 knockout with CRISPR is not very strong as with the other splicing factors. This is of particular importance as the authors state that 'NJACAP2 was most positively correlated with the expression of CELF2'. However, in cell line SF10417, where the siRNA mediated knockout of CELF2 was quite strong, the effect on NJACAP2 expression was not significant (Extended Data Fig. 5C-5D). While in the same cell line where CRISPR-mediated knockdown of CELF2 was approximately 50% (Extended Data 5A), the effect on NJACAP2 expression was significant for

guide 2 (Fig. 3J). Can the authors provide an explanation for this? The data for the other splicing factors is very significant and promising.

We thank the reviewer for their careful consideration of our new data. We too noticed this interesting trend and believe that this is the result of splicing changes becoming more evident and stabilized with the longer experimental time frame in the CRISPRi experiment compared to siRNA experiments. The siRNA experiments were conducted via transient transfection and the peak target knockdown was reached at 72 hours, and RNA was collected and target expression and splice patterns were assessed at this time point. In contrast, the CRISPRi knockdown was conducted via transduction (24 hours), which was followed by puromycin selection (96 hours) and recovery and expansion of the cells (7 days) and then RNA collection. Therefore, the time point for the assessment of target expression and splice patterns was performed at around day 12. Therefore, it is likely that these changes in splicing were overall more evident in the CRISPRi experiments than the siRNA experiments as target RBP knockdown was sustained for a longer period, which we suspect allowed more time for splice changes to take place and stabilize within the cell population. In support of this, the CRISPRi data were more significant for all RBP targets and neojunctions compared to the siRNA experiments.

Owing to our comments “It is not clear why the authors chose to validate neojunction ACAP2 (Figure 3J, 3L) and not one of the neojunctions that are the focus of the manuscript (NEJ-RPL22 and NEJ-GNAS)”, the authors have provided clarity on the matter by stating that the manuscript has a mechanism aspect and an immunotherapeutic aspect. Although this is now clearer, we feel that there is a lack of continuity between the two sections of the manuscript as no clear evidence on the immunotherapeutic effects of NJACAP2 and NJPEA15 was presented. The clinical aspect of the manuscript titled ‘Tumor-wide RNA splicing aberrations generate immunogenic public neoantigens’ may be confusing to the readers.

Thank you for the observation. The exploration of mechanism-driven changes in NJ expression began when we observed significant differences in NJ expression between tumor subtypes. In the case of **Figures 3A-3B** and **3G**, we investigated upregulated splicing genes that were associated with increased NJ recognition in the mutant *IDH* cases compared with the wild-type cases. NEJ_{GNAS} was a NJ that was readily expressed across both subtypes (and therefore not *IDH*mut-specific), however NEJ_{RPL22} was validated to be expressed predominantly in *IDH*mut cases. In the case of NEJ_{RPL22}, only 5 splicing-related genes were shown to be correlated to increased NEJ_{RPL22} expression: *PABPC1*, *RBM19*, *RNU6ATAC*, *HSPA1A*, *TSEN2*. Of these 5 splicing-related genes, only *PABPC1* was characterized as an *IDH*mut-specific splicing gene in **Figure 3G**. At the time of analysis, limited prior studies investigating *PABPC1*-mediated splicing deterred us from using this gene as our initial launching point for studying the mechanism driving NJ expression.

Therefore, of the *IDH*mut-specific (**Figure 3G**) and chromosome 1p/19q-specific splicing-related genes (**Figure 3H**), we selected for those that were well studied, with splicing events that were quantifiably differentiated due to splicing factor dysregulation. This led to the selection of *CELF2*, *SNRPD2*, and *SF3A3*, and the subsequent selection of NJs correlated with their expression: NJ_{ACAP2} and NEJ_{PEA15}. While NEJ_{GNAS} and NEJ_{RPL22} were selected with high confidence for their potential immunologic response, NJ_{ACAP2} and NEJ_{PEA15} were selected with high confidence for their mechanistic relationship with the chosen splicing-related genes. In our approach, both proof-of-concept paths required appropriate candidates with the highest likelihood of success.

Regardless, many NJs have the potential to be therapeutic targets, as we have demonstrated to completion with only 2 NJs in this study. We originally made the effort to organize all bioinformatic NJ characterization to Figures 1-3 and clinical/immunologic properties of the neojunction-derived neoantigens to **Figures 4-6** to ensure a logical transition. In response to the reviewer's suggestion, we have now also included a sentence in the discussion (**Lines 943-944**) to reflect this point: *“Future studies identifying and targeting splicing-related genes associated with NEJ_{GNAS} and NEJ_{RPL22} expression will potentially bolster their expression for improved therapeutic response.”*